# Early coordination of cell migration and cardiac fate determination during mammalian gastrulation

Shayma Abukar [1,2], Peter A Embacher[3], Alessandro Ciccarelli [4], Sunita Varsani-Brown [4], Isabel G W North[1], Jamie A Dean [3,5], James Briscoe [4] & Kenzo Ivanovitch [1]✉

## Abstract

During gastrulation, mesodermal cells derived from distinct regions are destined to acquire specific cardiac fates after undergoing complex migratory movements. Here, we used light-sheet imaging of live mouse embryos between gastrulation and heart tube formation to track mesodermal cells and to reconstruct lineage trees and 3D migration paths for up to five cell divisions. We found independent progenitors emerging at specific times, contributing exclusively to left ventricle/atrioventricular canal (LV/AVC) or atrial myocytes. LV/AVC progenitors differentiated early to form the cardiac crescent, while atrial progenitors later generated the heart tube's Nr2f2+ inflow tract during morphogenesis. We also identified short-lived multipotent progenitors with broad potential, illustrating early developmental plasticity. Descendants of multipotent progenitors displayed greater dispersion and more diverse migratory trajectories within the anterior mesoderm than the progeny of uni-fated progenitors. Progenitors contributing to extraembryonic mesoderm (ExEm) exhibited the fastest and most dispersed migrations. In contrast, those giving rise to endocardial, LV/AVC, and pericardial cells showed a more gradual divergence, with late-stage behavioural shifts: endocardial cells increased in speed, while pericardial cells slowed down in comparison to LV/AVC cells. Together, these data reveal patterns of individual cell directionality and cardiac fate allocation within the seemingly unorganised migratory pattern of mesoderm cells.

**Keywords** Cardiac Progenitors; Gastrulation; Light Sheet Microscopy; Heart Fields; Live imaging
**Subject Categories** Cardiovascular System; Development

## Introduction

How cell fate specification and morphogenesis are coordinated in time and space to generate tissues and organs of unique forms and functions is central to developmental biology. This phenomenon is evident during gastrulation when mesodermal cells acquire diverse cardiac fates and engage in complex cell movements to generate spatial patterns, including a cohesive cardiac crescent, which transforms into a primitive heart tube.

In the gastrulating embryo, tracking the derivatives of single progenitors by clonal analysis led to the finding that progenitors are assigned to specific anatomical locations in the heart prior to the formation of the heart fields (Wei and Mikawa, 2000; Zhang et al, 2021; Devine et al, 2014; Lescroart et al, 2014). Uni-fated *Mesp1+* progenitors solely destined for the left ventricle (LV) and atria myocardium can be distinguished from uni-fated endocardium progenitors (Lescroart et al, 2014). Additional clonal analysis of *Hand1+* progenitors located at the embryonic/extraembryonic boundary in the early gastrulating embryo identified bipotent and tripotent progenitors. These progenitors generated LV/AVC myocytes in addition to pericardium, epicardium, and extraembryonic tissues (Zhang et al, 2021).

One limitation of clonal analysis, however, is that the history of the cells is *deduced* by analysing descendants at the endpoint. It does not allow the identification of the progenitors' initial locations or subsequent migratory paths in the embryo. Single-cell tracking in live imaging is needed for this and is the most rigorous approach to reconstituting cell lineages, identifying when cardiac progenitors become lineage-restricted during gastrulation and enabling migration analysis (Wolff et al, 2018).

A recent live-imaging analysis uncovered the dynamics of mesodermal cell migration during mouse gastrulation (Dominguez et al, 2023). This analysis revealed that cells dispersed extensively in the embryo, with clearly separate movements of daughter cells, suggesting cell identity may not be fixed but instead influenced by the position of the cells at the end of the migration period. However, as Dominguez et al discussed, the motility of mesodermal cells is unlikely to be completely random. There may be some regulation of directionality of individual cell migration to ensure progeny migrate to their correct locations and establish spatial patterns, including the cardiac crescent and distinct LV/AVC and atrial progenitor domains (Bardot et al, 2017; Gonzalez et al, 2022; Ivanovitch et al, 2021). Indeed, a previous migration analysis showed mesodermal cell migrate with directionality during mouse

[1]Developmental Biology and Cancer Department, Institute of Child Health, University College London, 30 Guilford Street, London WC1N 1EH, UK. [2]Institute of Cardiovascular Science, University College London, Gower Street, London WC1E 6BT, UK. [3]Department of Medical Physics and Biomedical Engineering, University College London, Gower St, London WC1E 6BT, UK. [4]The Francis Crick Institute, 1 Midland Road, London NW1 1AT, UK. [5]Institute for the Physics of Living Systems, University College London, London WC1E 6BT, UK. ✉E-mail: k.ivanovitch@ucl.ac.uk

gastrulation (Saykali et al, 2019). Thus, early mammalian mesoderm migration may exhibit some degree of determinism. This phenomenon is reminiscent of an evolutionarily distinct species, the ascidian, in which a small number of genealogically related and determined heart progenitors migrate with predetermined directionally (Christiaen et al, 2008). However, it is not known whether progenitors adopt more stereotypical migratory trajectories, once committed to specific cardiac fates in the context of mammalian gastrulation.

Here, through long-term live imaging and single-cell tracking in mouse embryos, spanning from the initiation of gastrulation to the stages of heart tube formation, our goal was to reconstitute the lineage tree of cells and assess how the migratory paths of cells relate to their eventual cardiac fate within the seemingly unorganised migration pattern.

# Results

## Development and characterisation of *cTnnT-2a-eGFP* mice

To track cardiomyocytes in vivo, we developed a knock-in mouse reporter line *cTnnT-2a-eGFP* where the *eGFP* sequence is inserted downstream of the endogenous *cardiac troponin T* (*cTnnT*) loci. A virus-derived 2a self-cleaving peptide inserted between the *cTnnT* and *eGFP* coding sequence ensures co-expression of both cTnnT and eGFP proteins (Fig. 1A) (Szymczak et al, 2004). The *cTnnT-2a-eGFP* line was maintained as homozygotes. Animals are viable and indistinguishable from heterozygotes. Whole-mount immunostaining for cTnnT confirmed specific eGFP expression in cTnnT+ cardiomyocytes at E8 -heart tube stage- and E12.5 (Fig. 1B,C).

We first analysed cardiac differentiation dynamics in real time using multiphoton live imaging and the cardiomyocyte *cTnnT-2a-eGPF* reporter line (Fig. 1D). We combined the *cTnnT-2a-eGFP* reporter with the *Bre:H2BCerulean* BMP reporter (Ivanovitch et al, 2021), which expressed cerulean in the lateral plate mesoderm. We found initial GFP-positive cells appeared within the Bre-Cerulean positive lateral plate mesoderm at E7.5, consistent with initial sparse cTnnT protein distribution found in the lateral plate mesoderm (Fig. 1D, arrows) (Ivanovitch et al, 2017). This event was followed by the establishment of the cardiac crescent, an epithelium-like structure reminiscent of a mesenchymal-epithelial transition and preceding the formation of the heart tube (Cui et al, 2009; Dominguez et al, 2023; Ivanovitch et al, 2017) (Fig. 1D). We conclude that the *cTnnT-2a-eGFP* reporter faithfully identifies cardiomyocytes among a population of lateral plate mesodermal cell derivatives.

## Lineage analysis using a tamoxifen-inducible T/Bra reporter

Previous lineage tracing analysis revealed a temporal order in which different cardiac lineages arise within the mesoderm (Ivanovitch et al, 2021; Lescroart et al, 2014). To further assess the contribution of the temporal distinct mesodermal populations to the cardiac crescent and heart tube cell populations, we performed similar genetic tracing using an inducible $T^{nGFP-CreERT2/+}$ mouse, expressing *CreERT2* and nuclear localised GFP (nGFP) downstream of the endogenous *T* (Imuta et al, 2013) in

combination with the $R26R^{tdTomato/+}$ reporter and our novel cardiomyocyte *cTnnT-2a-eGPF* reporter ($T^{nEGP-CreERT2/+}$; $R26R^{tdTomato/+}$; *cTnnT-2a-eGFP*).

At ~E7.75, the mesoderm is partitioned into three progeny compartments within the cardiac crescent: prospective endocardium, prospective myocardium, and prospective pericardium (DeRuiter et al, 1992; Dominguez et al, 2023). At later somite stages (E8.25), the cardiac crescent has generated the heart tube. The heart tube has an inverted Y shape, and the two arms of the Y -or inflows-, positioned inferiorly, are fated to become the atria, with the stem of the Y becoming the left ventricle (LV) and atrioventricular canal (AVC) (Moorman et al, 2003).

Doses of tamoxifen (0.02 mg/bw) were administered at successive gastrulation stages (Fig. 2A) (Ivanovitch et al, 2021). Early administration at E6 + 7 h resulted in sparse tdTomato+ labelling of myocytes, endocardial, and pericardial cells in the cardiac crescent (n = 3/3 embryos, Fig. 2B,C), as well as in the endoderm. Later administration at E6 + 21 h showed similar sparse labelling in most cases (n = 7/10 embryos) (Fig. 2Bii), while in some cases (n = 3/10 embryos), no contribution to GFP+ myocytes cells was observed (Fig. 2Biii). As previously noted, variability in embryonic stages within litters at E6 + 21 h likely explains these differences, with some embryos at more advanced stages having already downregulated *T/Bra* expression in cardiac crescent progenitors (Ivanovitch et al, 2021).

For both E6 + 7 h and E6 + 21 h tamoxifen regimens, a higher proportion of tdTomato+ myocytes were labelled compared to endocardial and pericardial progenitors (Fig. 2C,D). This disparity suggests that the initial T/Bra-expressing mesoderm contains a greater number of myocyte progenitors. This finding likely explains the higher number of myocytes in the cardiac crescent relative to endocardial and pericardial cells. In some cases (n = 5/13 embryos), we observed a large number of tdTomato-labelled GFP+ myocytes (between 57 and 127 cells) with rare or no corresponding tdTomato-labelled endocardial cells (between 0 and 8 cells) (Fig. 2Ciii). This observation supports the existence of independent progenitors that specifically contribute to myocytes in the cardiac crescent, but not to the endocardium (Wei and Mikawa, 2000; Lescroart et al, 2014; Sendra et al, 2024, preprint). These results are consistent with a previous Foxa2 lineage tracing study, which identified independent *Foxa2*+ progenitors in the *T/Bra*+ primitive streak contributing to ventricular myocytes, pericardium, and epicardium, but not the endocardium using a similar tamoxifen regimen (Ivanovitch et al, 2021).

When tamoxifen was administered at E7 + 7 h, we observed a rare contribution of tdTomato+ cells to myocytes in the cardiac crescent (n = 3/8 embryos, mean 10 tdTomato+ myocytes per embryo) and no contribution to endocardial or pericardial progenitors for all embryos analysed (Fig. 2Ciii). At the heart tube stage, however, there was a strong contribution of tdTomato+ cells to inflow myocytes (>100 cells in all embryos analysed, n = 4/4 embryos), and inflow endocardial cells (39 cells on average, n = 4/4 embryos) which are destined to form the atria during later development (Fig. 2Ei,ii). This finding suggests an independent source of T/Bra-positive late mesodermal cells contributing to inflow myocytes and endocardium, as indicated in previous lineage tracing studies (Ivanovitch et al, 2021; Zamir et al, 2017). Determining the clonal relationship between these cell populations will require tracing individual progenitors from their mesodermal origins to their final positions in the heart tube. As previously noted in a

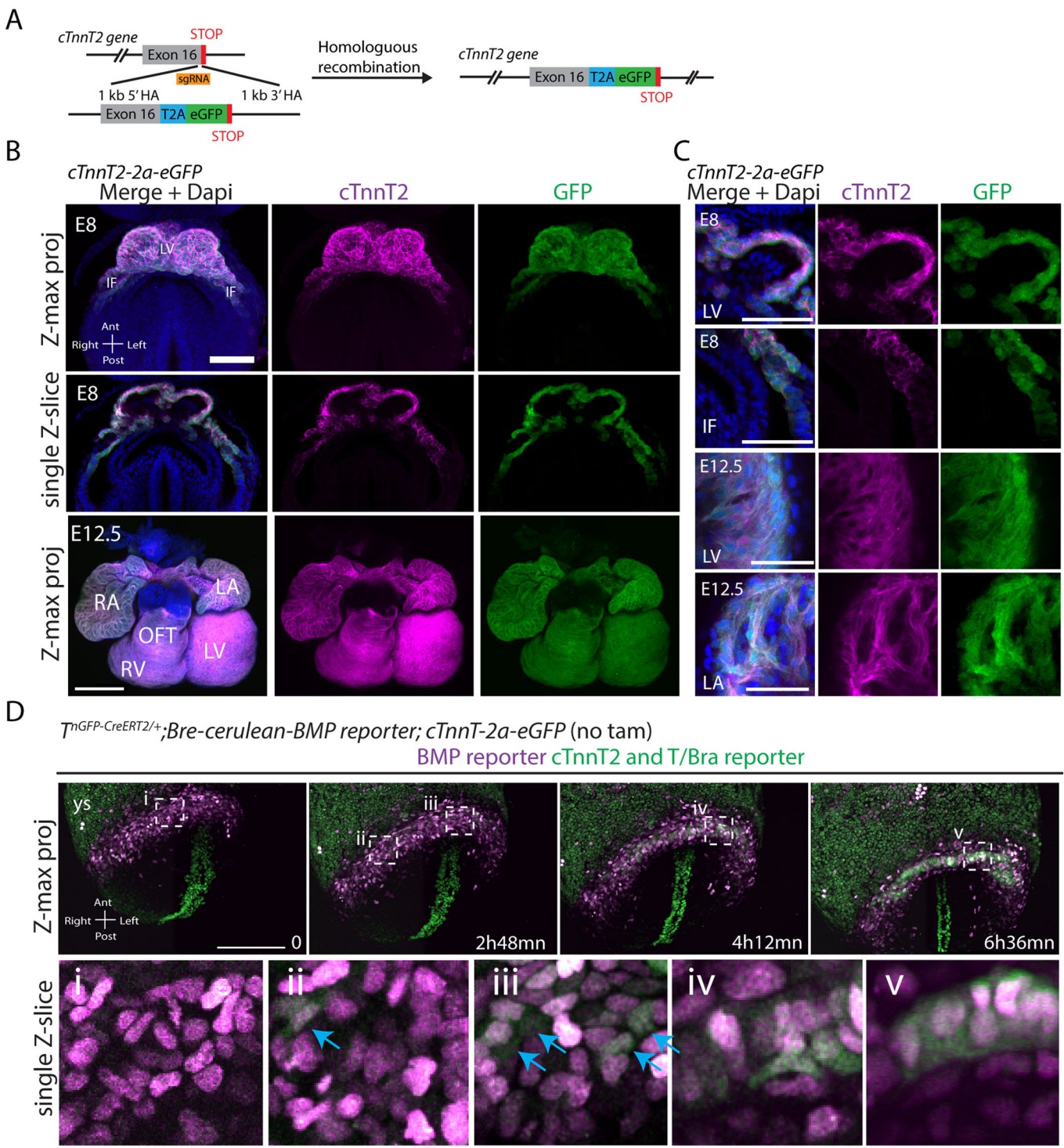

**Figure 1. Characterization of the cTnnT-2a-GFP line.**

(**A**) CRISPR-Cas9 strategy to insert a T2a-eGFP cassette into the cTnnT2 gene. (**B**, **C**) Immunofluorescence for cTnnT2 in *cTnnT-2a-eGFP* hearts at E8 (scale: 100 μm) and E12.5 (scale: 200 μm). (**D**) Multiphoton time-lapse sequence of a *T^{nGPF-CreERT2/+}*; *Bre-cerulean; cTnnT-2a-eGFP* embryo (scale: 100 μm). Blue arrows point to *cTnnT-2a-eGFP+* cells. IF inflows, HT heart tube, OFT outflow tract, LV left ventricle, RV right ventricle, LA left atrium, RA right atrium, YS yolk sac. Source data are available online for this figure.

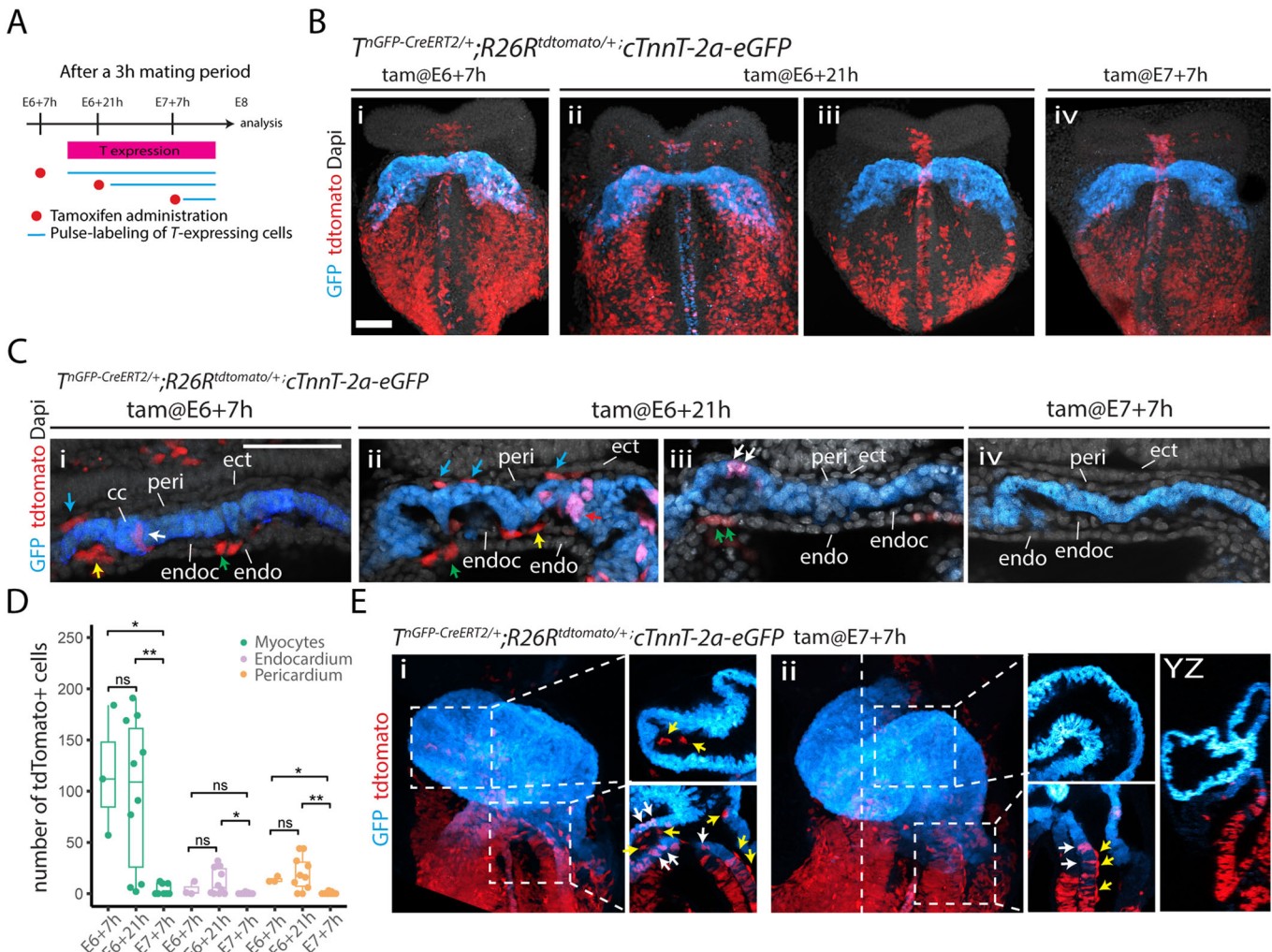

**Figure 2. Lineage tracing of early cardiac mesoderm.**

(A) Tamoxifen administration strategy. (B) $T^{nGPF-CreERT2/+};R26R^{tdTomato/+}; cTnnT-2a-eGFP$ embryos labelled at E6 + 7 h, E6 + 21 h, and E7 + 7 h, analysed at E8. Scale: 100 μm. (C) Single optical sections from $T^{nGPF-CreERT2/+};R26R^{tdTomato/+}; cTnnT-2a-eGFP$ embryos labelled at E6 + 7 h, E6 + 21 h, and E7 + 7 h, analysed at E8. Scale: 100 μm. (D) Quantification of tdTomato+ cells (myocardial, endocardial, pericardial) in embryos labelled at E6 + 7 h ($n = 3$), E6 + 21 h ($n = 10$), and E7 + 7 h ($n = 8$). Myocardium: E6 + 7 h vs E6 + 21 h, $P$ value = 0.81. Myocardium: E6 + 21 h vs E7 + 7 h, $P$ value = 0.0061. Myocardium: E6 + 7 h vs E7 + 7 h, $P$ value = 0.014. Endocardium: E6 + 7 h vs E6 + 21 h, $P$ value = 0.55. Endocardium: E6 + 21 h vs E7 + 7 h, $P$ value = 0.029. Endocardium: E6 + 7 h vs E7 + 7 h, $P$ value = 0.24. Pericardium: E6 + 7 h vs E6 + 21 h, $P$ value = 0.8. Pericardium: E6 + 21 h vs E7 + 7 h, $P$ value = 0.0061. Pericardium: E6 + 7 h vs E7 + 7 h, $P$ value = 0.01. Each data points are one embryo. The box boundaries represent the 25th (lower quartile) and 75th (Upper quartile) percentiles, with the centre line indicating the median. The whiskers extend to the minimum and maximum values in the dataset. (E) Embryos labelled at E7 + 7 h, analysed at E8.5. Arrows in (C) and (E) indicate cell types: blue for pericardium, yellow for endocardium, green for endoderm, and red for myocytes. CC cardiac crescent, peri pericardium, ect ectoderm, endoc endocardium, endo endoderm. *$P \leq 0.05$, **$P \leq 0.01$, ***$P \leq 0.001$, ****$P \leq 0.0001$. All statistical analyses were performed using the Mann–Whitney $U$ test. Source data are available online for this figure.

similar experiment (Ivanovitch et al, 2021), we also observed cases with sparse tdTomato+ endocardial cells in the ventricular regions, without any tdTomato+ cells in the ventricular myocyte GFP+ layer ($n = 2/4$ embryos) (Fig. 2Eii). This observation further supports the idea that independent sources exist for myocyte and endocardial progenitors.

## Establishing long-term light-sheet live microscopy for cardiac lineage analysis

We employed live imaging and single-cell tracking, using the cardiomyocyte-specific cTnnT-2a-eGFP reporter, to reconstruct the

lineage trees of mesodermal cells. Our objective was to identify the initial mesodermal progenitors contributing to the distinct cell populations within the heart tube of the gastrulating mouse embryo (Fig. 3A). This approach required the culture and imaging of early-stage mouse embryos, spanning from the onset of gastrulation to the heart tube formation stage, a period of ~25–40 h.

We found the Viventis LS1 open-top light-sheet microscope allowed the culture of early mouse embryos over long periods of embryonic development (>24 h) (Moos et al, 2024). Incubation media was stable and could be exchanged during acquisition. A large media volume (~1 ml) improved embryonic viability for long-term imaging. Embryos cultured from E6.5 and for up to 40 h

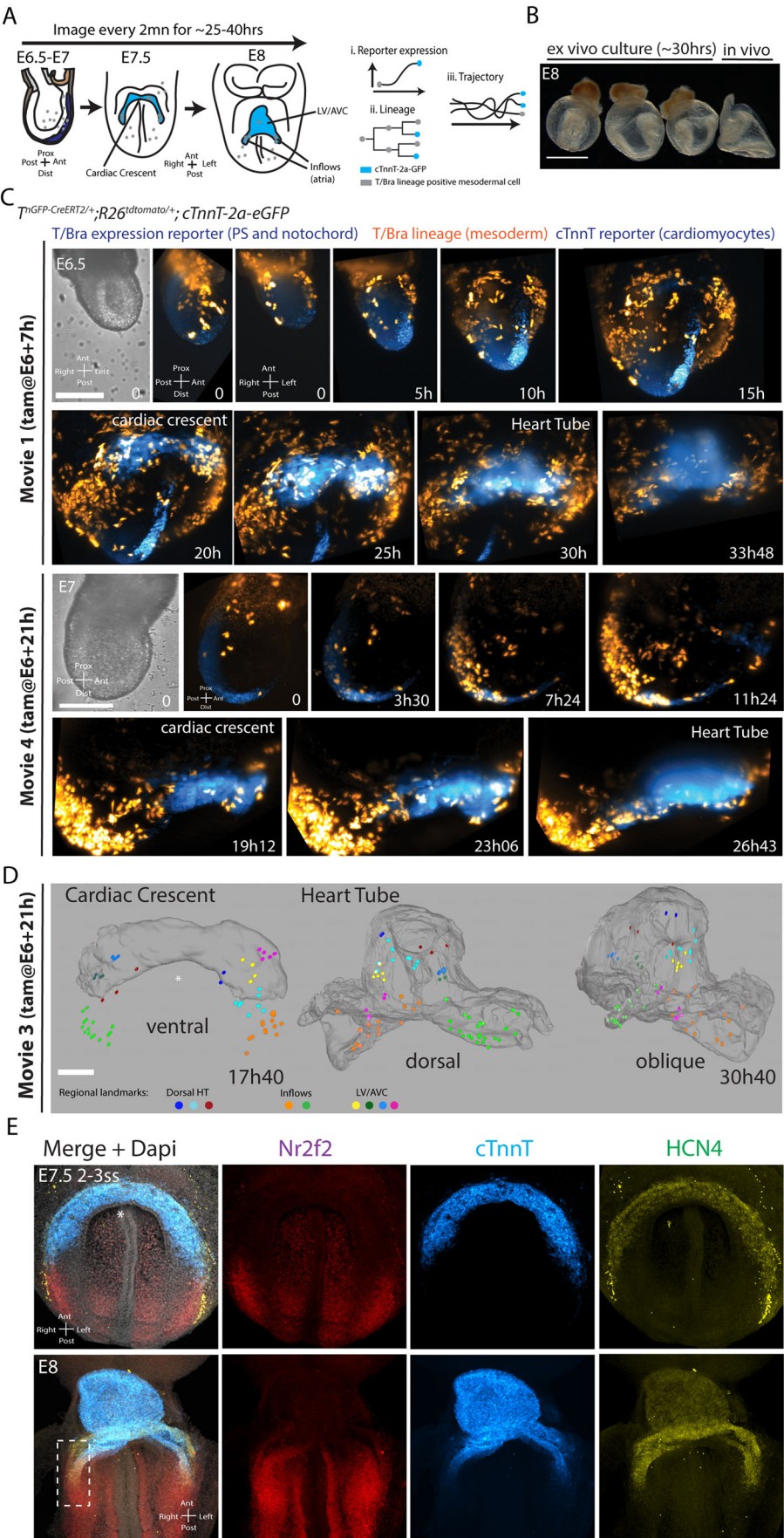

**Figure 3. Live Analysis of cardiogenesis from gastrulation to heart tube formation.**

(A) Schematic of cell tracking showing reporter expression, lineage relationships, and cell trajectories. (B) Example of embryos cultured ex vivo for 30 h. (C) Time-lapse sequences of $T^{nGPF-CreERT2/+}$;$R26R^{tdTomato/+}$; $cTnnT-2a-eGFP$ embryos after tamoxifen administration (0.02 mg/g) at the indicated times. Scale bar: 100 µm. (D) Fate mapping of the cardiac crescent. Cells are colour-coded by their final position in the heart tube. Scale bar: 100 µm. (E) HCR in situ hybridization in E7.5-E8 embryos showing expression of HCN4, Nr2f2, and cTnnT, with the inflow region highlighted at E8. Dotted box highlights a Nr2f2/HCN4+ domain. ss somites. Source data are available online for this figure.

developed normally; a cardiac crescent formed and generated a heart tube corresponding to E8 embryos (Fig. 3B; Appendix Fig. S2A,B).

To permanently label mesodermal cells and their progeny at a density suitable for live cell tracking, we used an inducible $T^{2a-cre/ERT2}$ mouse combined with $R26R^{tdomato}$ reporter and administered intermediate doses of tamoxifen (0.02 mg/bw) at E5, E6 + 7 h and E6 + 21 h (Ivanovitch et al, 2021). We first administered tamoxifen earlier (at E5) and cultured embryos in tamoxifen-free culture media, from before the start of the gastrulation period and onset of T/Bra expression in the primitive streak and mesoderm (at E6). In these conditions, no tdTomato-expressing cells could be identified in the intra-embryonic mesoderm over ~11 h of live-imaging acquisition (Appendix Fig. S2C). This finding confirms that creERT2 activity in $T^{2a-cre/ERT2}$ embryos requires T/Bra expression (Ivanovitch et al, 2021). In the absence of tamoxifen, rare tdTomato-positive cells were identified in only one embryo (Appendix Fig. S1, out of 5 embryos), confirming that tdTomato widespread mesodermal expression in $T^{2a-cre/ERT2}$; $R26^{tdTomato}$ embryos requires tamoxifen.

We generated five light-sheet live-imaging datasets spanning 23 to 41 h of mouse embryonic development from gastrulation—E6.5-E7 (Fig. EV1)—to heart tube stage (Fig. 3C; Appendix Fig. S3; Movies EV1 and 2). Embryos were imaged at 2 min intervals. Raw data amounts to 5–7 terabytes per experiment, representing up to half a million images. To correct for drift during acquisition, BigStitcher (Horl et al, 2019) was used to register the datasets in 4D as previously described (Dominguez et al, 2023).

Embryos in Movies EV3–5 are imaged from a later stage (OB-LB stages) than those imaged in Movies EV1 and 2 (MS to LS stages) (Table EV1). At T0, embryos in Movies EV3–5 have a longer PS (Fig. EV1A), an exposed node located at the distal tip (Fig. EV1A'), and higher PS nGFP expression (Fig. EV1B,C; Appendix Fig. S6A–A'), reflecting increased T/Bra TF expression in the PS at these later stages, as previously shown (Ivanovitch et al, 2021). Moreover, the notochord plate begins to extend (see red dotted lines in Fig. EV1A').

Each movie contains up to ~1200 time points, and a small percentage (<1%) of linkage inaccuracy between cells could lead to lineage misinterpretation that propagates over the course of the movie. Although automated cell tracking methods have seen major advances (Malin-Mayor et al, 2023; Sugawara et al, 2022; Bragantini et al, 2024), it is still the case that achieving the level of precision necessary to reconstitute cell lineages over protracted periods remains challenging. To obtain accurate cell lineages, we manually tracked single T/Bra lineage-positive cells by visualising them at successive time points from the beginning to the end of the movie using Massive Muti-view Tracker (MaMut) (Wolff et al, 2018). We interrupted a track when it was impossible to identify the same cell across two successive time points unequivocally, which resulted in

the presence of short tracks. A total of 227 mother cells were tracked for up to 5 generations resulting in 1299 descendants (Movies EV3 and 4).

We determined the identity of the final daughters based on their location in the heart tube: within an inner CD31+ endocardial layer ensuring the presence of a circulatory system (Dominguez et al, 2023; Ivanovitch et al, 2017), a myocardial layer formed by cTnnT-2a-eGFP+ cardiomyocytes and derived from the splanchnic mesoderm, and an outer layer derived from the Hand1+ somatic mesoderm called the pericardium (Movies EV5–7; Appendix Fig. S4A,B) (Zhang et al, 2021). We could discriminate these three cell types in our live-imaging datasets. Moreover, cardiomyocytes were further distinguished by their higher levels of cTnnT-2a-eGFP reporter expression (Appendix Fig. S5A–F and "Methods"). A substantial number of mother cells ($n = 111$) produced at least one progeny whose fate could not be determined (see explanations in "Methods"). In these cases, we could not determine if the mother cells were uni-fated or multipotent. Finally, the locations of the myocyte cTnnT-2a-eGFP+ cells within the LV/AVC and inflow regions of the heart tube indicated their fates. In what follows, we describe the lineage trees and timing for mesodermal progenitors' specification into distinct cardiac lineages.

## Heart tube morphogenesis

Initial analysis during the stages that the heart tube develops reveals a progressive propagation of myocyte differentiation and tissue folding along the anterior–posterior axis (Fig. 3D). Myocyte differentiation is first triggered in the anterior regions that generate the cardiac crescent, and as tissue folding proceeds following epithelialization, the most anterior cells reposition to assume a more ventral position within the LV/AVC components of the heart tube. As differentiation and folding continue posteriorly, posterior cells contribute to the dorsal closure and inflow regions of the heart tube (see colour-coded regional landmarks in Fig. 3D; Movie EV8). Thus, heart tube morphogenesis occurs through a coordinated process of differentiation, epithelialization, and tissue folding, with the cardiac crescent initially forming the ventral aspect of the heart tube, while the inflow and dorsal aspects are progressively recruited to complete its formation.

In situ HCR analysis further showed that the cardiac crescent and posterior mesoderm, destined to form the LV/AVC and inflow components of the heart tube respectively, exhibit distinct gene expression profiles (Fig. 3E). HCN4, a first heart field marker (Liang et al, 2013; Spater et al, 2013), is expressed in both the prospective LV/AVC and inflow components of the heart tube, while Nr2f2, a marker required for atrial lineage specification (Wu et al, 2013), has its expression restricted to the posterior mesoderm and inflows myocardium in the heart tube (E7.5 and E8.5 in Fig. 3E).

## Independant LV/AVC and Atria progenitors contribute to distinct regions of the heart tube

We next addressed the timing of LV/AVC and inflow myocyte lineage segregation. From our five datasets, we identified 91 progenitors contributing to at least one cTnnT-2a-eGFP+ myocyte (Fig. 4; Table EV2). Of these 91 progenitors, 61 contributed to the cardiac crescent, giving rise to 272 descendants, with 47 of these 61 progenitors traceable all the way to the heart tube. In addition, 30 progenitors contributed to atrial cells, generating 84 descendants.

Previous clonal labelling of single mesodermal progenitors demonstrated the existence of precursors restricted to specific anatomical locations within the heart (Devine et al, 2014; Lescroart et al, 2014; Zhang et al, 2021, Meilhac et al, 2004a), while clonal induction at earlier embryonic stages typically generated larger clones that span multiple heart compartments, contributing to both heart fields (Meilhac et al, 2004a). Consistent with these findings, our cell tracing at the mesoderm stage showed that most traced cells exhibit limited dispersion within the heart tube. Analysing the progeny locations within the heart tube, we identified a clonal boundary at the junction between the LV/AVC and inflow myocyte compartments, suggesting that atrial and LV/AVC progenitors have distinct mesodermal origins (Figs. 5Ai–iii and EV2A–D). None of the clones contributed to both the LV/AVC and inflow myocyte compartments.

In most cases, GFP+ clones were confined to specific dorsoventral (DV) and anteroposterior (AP) positions, with 35 out of 44 clones spreading less than 20% along the longest AP axis of the LV/AVC heart tube region, and 32 out of 44 clones spreading less than 20% along the longest DV axis (Fig. 5B–E; Movie EV9). In rare instances, we observed clones spreading across both the ventral and dorsal regions, as well as the anterior and posterior regions, particularly when tracked from earlier stages of gastrulation (e.g. Movies EV1 and 2). These clones dispersed across 40% to 60% of the entire AP ($n = 1/44$) and DV ($n = 4/44$) axis lengths (Fig. 5B,D–E). In even rarer cases ($n = 1/44$), we observed cells crossing the midline, with a clone spanning both the left and right sides of the cardiac crescent and heart tube (Figs. 5F and 6A, indicated by white arrows). The later the embryonic stage at the start of culture, the less dispersed along the anterior-posterior (AP) and dorsal–ventral (DV) axes of the heart tube the clones were (Fig. EV1A–C). This is consistent with the notion that cell dispersion was more prominent during the earliest phases of migration taking place in the earlier embryos, consistent with the results from Dominguez et al (Dominguez et al, 2023).

Analysis of distances between sister cells shows that, on average, GFP+ clones do not spread significantly further during the stages of heart tube formation (Fig. 5G,H). This finding indicates that most clonal dispersion occurs during earlier cell migration, whereas the heart tube formation is primarily driven by tissue-scale processes like epithelial deformation and folding, rather than individual cell movements or intercalation (Cui et al, 2009; Dominguez et al, 2023; Ivanovitch et al, 2017). Consequently, clonal spreading is minimal during heart tube formation.

LV/AVC progenitors are born first and differentiate into *cTnnT-2a-eGFP+* myocytes before other cardiomyocytes. This establishes the initial cardiac crescent within a ~15-h period with the first LV/ AVC progeny differentiating at 16 h and the last at 31 h. Atrial progenitors are born later, differentiate later, from 24 h to 42 h, and

are recruited to posterior regions during the folding of the cardiac crescent into the heart tube. This event establishes the inflows (Figs. 5Aiii and 6A–D).

Myocytes developed concurrently within each lineage (Fig. 6E). The time intervals between the first and the last daughter to transition into cTnnT-2a-eGFP+ myocytes were similar on average between the LV/AVC and atrial lineages (means of 2.0 h and 2.1 h for LV/AVC and atrial lineages respectively). Notably, the differentiation timing varied among lineages, with some displaying greater synchrony than others. For instance, in 5 out of 55 lineages, the mother cell generated LV/AVC cTnnT-2a-eGFP+ myocyte daughters in more than 5 h. In contrast, in 25 out of 55 lineages, all daughters transitioned into cTnnT-2a-eGFP+ myocytes in less than 1 h.

Together, the live-imaging analysis shows that the heart tube is established by at least two sets of independent LV/AVC and inflow myocyte progenitors generated from early and late mesoderm, respectively, and differentiating into myocytes at different embryonic stages. This observation is in line with the previous hypothesis that atrial and LV/AVC compartments have distinct spatial and temporal origins during gastrulation in the mouse (Bardot et al, 2017; Gonzalez et al, 2022; Ivanovitch et al, 2021).

## Identification of uni-fated, bipotent and tripotent mesodermal progenitors

We next investigated whether the mesoderm is predominantly composed of multipotent cells, which have the potential to differentiate into various cell types, or a heterogeneous mixture of cells that are already committed to specific fates (Psychoyos and Stern, 1996).

Among the five embryos analysed, we found that the majority of progenitors tracked within the early proximal mesoderm contributed to the LV/AVC ($n = 61$), whereas endocardial ($n = 18$) and pericardial ($n = 13$) progenitors were less common in the mesoderm (Fig. 4; Table EV2 and Fig. 7A,B). This disparity, in contribution to the three cell types, also shown by lineage tracing experiments in a larger cohort of embryos (Fig. 2D), likely explains the higher proportion of myocytes compared to endocardial and pericardial cells within the cardiac crescent, indicating that a larger number of LV/AVC progenitors is necessary to account for these differences.

A substantial number of mother cells produced at least one progeny whose fate could not be determined ($n = 111$) (Table EV2 and see Fig. 7C, green arrows, for an example and explanations in "Methods"). In these cases, we were unable to determine whether the mother cells were uni-fated or multipotent. Focusing on progenitors for which we could determine potency, our analysis revealed a predominance of uni-fated progenitors ($n = 98$, generating 728 descendants) compared to bipotent/tripotent progenitors ($n = 18$, generating 302 descendants). We identified 29 uni-fated mother cells generating only LV/AVC myocytes and 12 bipotent/ tripotent progenitors contributing to at least one LV/AVC myocyte descendant. Thus, approximately a third (12/41) of the mother cells contributing to LV/AVC myocytes, for which we could determine potency, were bipotent or tripotent.

We could not determine whether certain types of bipotent progenitors were more prevalent than others (Table EV2). Instead, we observed a diversity of bipotent progenitor cell types, including

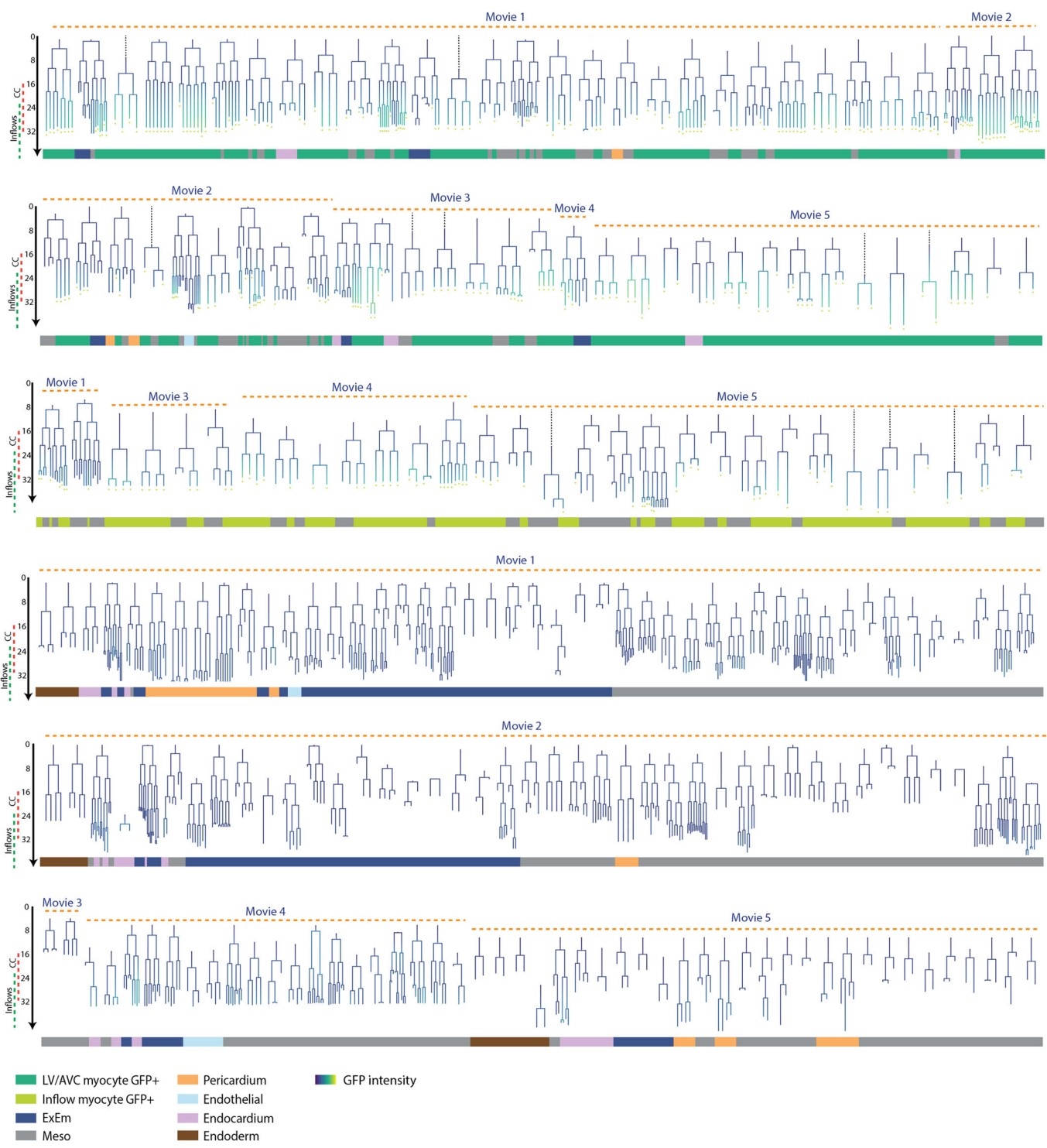

**Figure 4. Reconstruction of the early mesodermal lineage tree.**

Lineage trees of tracked progenitors. Cell types are shown at the endpoints, with GFP-positive descendants indicated by a yellow plus sign. Lineages are colour-coded based on normalized GFP intensities. Dashed lines indicate mother cells that could not be tracked back to their birth. Progenitors not contributing to any LV/AVC myocytes are shown in the bottom panel. cc cardiac crescent. Source data are available online for this figure.

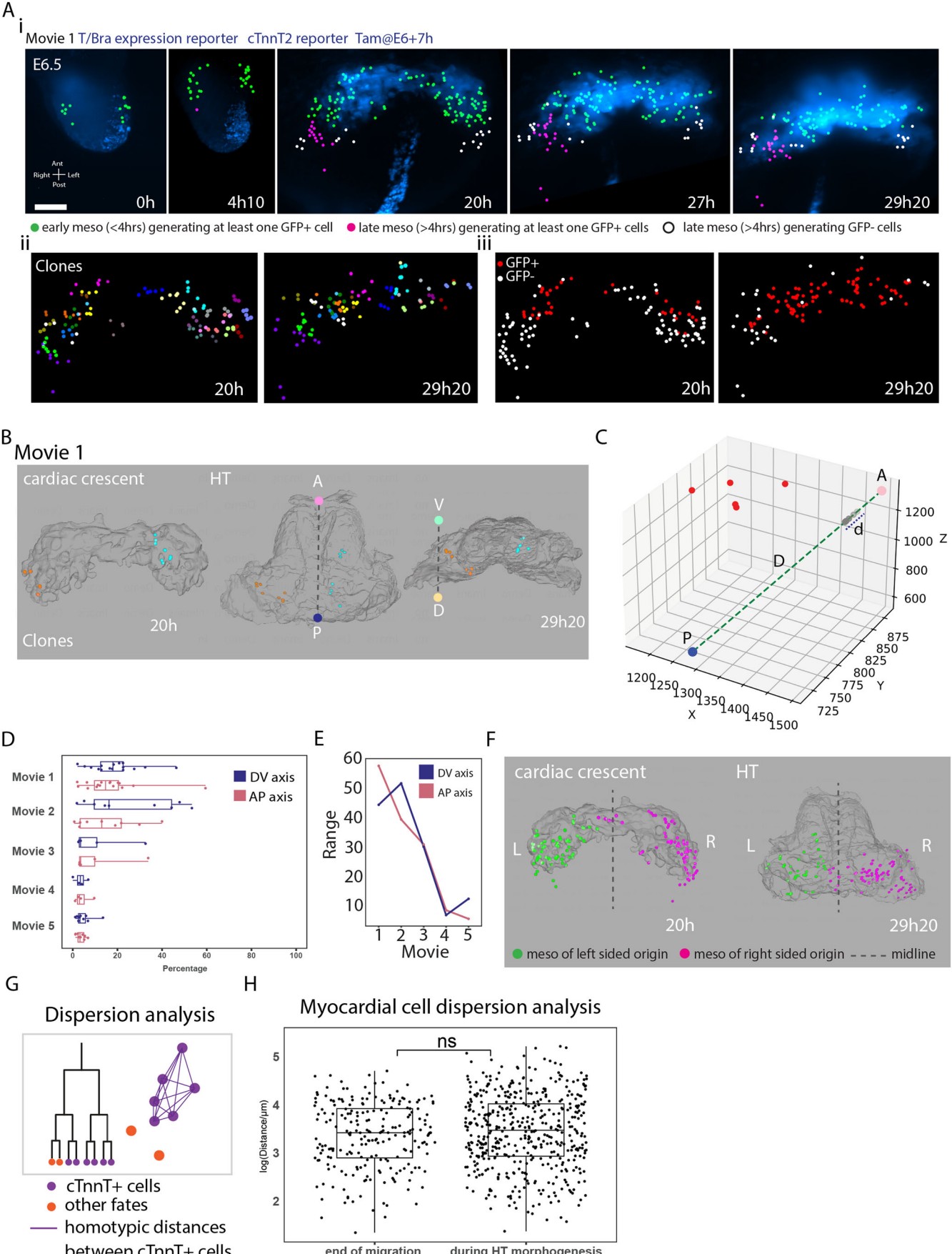

**A**

**i** Movie 1 T/Bra expression reporter  cTnnT2 reporter  Tam@E6+7h

E6.5 | 0h | 4h10 | 20h | 27h | 29h20

Ant / Right / Left / Post

● early meso (<4hrs) generating at least one GFP+ cell    ● late meso (>4hrs) generating at least one GFP+ cells    ○ late meso (>4hrs) generating GFP- cells

**ii** Clones | 20h | 29h20

**iii** ● GFP+  ● GFP-  | 20h | 29h20

**B** Movie 1

cardiac crescent    HT

A / P / V / D

Clones | 20h | 29h20

**C** 

**D**

Movie 1 / Movie 2 / Movie 3 / Movie 4 / Movie 5

■ DV axis  ■ AP axis

Percentage

**E** Range

■ DV axis  ■ AP axis

Movie

**F** cardiac crescent    HT

L / R    L / R

20h    29h20

● meso of left sided origin    ● meso of right sided origin    - - - midline

**G** Dispersion analysis

● cTnnT+ cells
● other fates
— homotypic distances between cTnnT+ cells

**H** Myocardial cell dispersion analysis

ns

log(Distance/µm)

end of migration    during HT morphogenesis

Figure 5. Dispersion analysis of myocardial clones.

(A) (i) Fate maps showing early (green) and late (magenta) mesoderm contributing to cTnnT-2a-GFP+ progenitors. Non-contributing, late progenitors are in white. (ii) Each colour represents a distinct clone. (iii) cTnnT-2a-GFP+ cells are in red, cTnnT-2a-GFP− cells in white. Scale bar: 100 μm. (B, C) Example of myocardial clones dispersed along anteroposterior (AP) and dorsoventral (DV) axes. cTnnT-2a-eGFP-positive cells' coordinates are mapped onto the AP and DV axes. The spread of each clone was quantified as the proportion of its range (d) along these axes (D). (D) Dispersion along AP and DV axes across movies. Each replicate is a Movie (one embryo). (E) Range of dispersion along the AP and DV axes for each movie. (F) Locations of mesodermal progenitors from left/right nascent mesoderm contributing to cTnnT-2a-GFP+ cells. (G) Schematic of dispersion analysis in lineages. (H) Comparison of cell-cell dispersion for cTnnT-2a-GFP+ myocytes at migration's end (defined by the mean LV/AVC differentiation time in each movie) and during heart tube formation (41.4 μm ± 31.9 (SD), $n = 293$ at the end of the migration period; 45.1 μm ± 34.7 (SD), $n = 597$, after heart tube formation ($P = 0.22$). HT heart tube, meso mesoderm, A anterior, P Posterior. All statistical analyses were performed using the Mann–Whitney $U$ test. For all boxplots, the box boundaries represent the 25th (lower quartile) and 75th (upper quartile) percentiles, with the centre line indicating the median. The whiskers extend to the minimum and maximum values in the dataset. Source data are available online for this figure.

LV/AVC-endocardial ($n = 4$), LV/AVC-ExEm ($n = 4$), LV/AVC-pericardial ($n = 2$), and LV/AVC-endothelial-like ($n = 1$), as well as endocardial-ExEm ($n = 3$) and pericardial-ExEm ($n = 1$) bipotent progenitors and one tripotent progenitor contributing to LV/AVC myocytes, endocardium, and extraembryonic mesoderm (Figs. 7A,C,D and EV3A–C; Appendix Fig. S10A,B; Table EV2). One additional bipotent progenitor generated highly migratory endothelial-like progeny located in both the embryonic and extraembryonic mesoderm (Fig. 7C').

Our lineage analysis led us to examine the initial locations of uni-fated and multipotent progenitors within the early proximal mesoderm (Movie EV10). We observed that LV/AVC, pericardial, and ExEm progenitors showed a tendency to localize in separate regions (Fig. 7A; Appendix Fig. S7). Notably, uni-fated pericardial progenitors were initially situated in anterior regions, consistent with a recent analysis in zebrafish indicating that pericardial cells originate from locations distinct from the heart fields (Moran et al, 2024) (preprint). However, we found that initial cell positions did not strictly correlate with their eventual fates. For instance, three neighbouring cells could migrate to different regions of the embryo and adopt diverse fates (Fig. 7C–C"). These results suggest that precise cellular resolution positional information dictating cell fate may not be present in the early proximal mesoderm. Moreover, bipotent and tripotent progenitors were intermingled with uni-fated LV/AVC progenitors, lacking any discernible spatial pattern (Fig. 7A).

Consistent with previous live-imaging analyses (Dominguez et al, 2023; Ivanovitch et al, 2021), the distal mesoderm, already present in the early stages of the gastrulating embryo (Fig. 7A, dark red cells at 1 h), migrated to more medial locations known to contribute to the right ventricle, outflow tract, and branchiomeric muscles (Kelly et al, 2001; Zaffran et al, 2004). No T/Bra lineage-positive medial mesoderm was identified in Movie 4 (Fig. 3C). Moreover, a subset of the cells displayed spindle-like shapes and were identified as endothelial-like cells ($n = 4$, Figs. 4 and 7C'; Appendix Fig. S10A).

In the late mesoderm, we identified 13 uni-fated atrial myocyte progenitors out of the 30 progenitors contributing to the cTnnT-2a-eGFP+ atrial myocytes. Longer tracks encompassing later stages will be necessary to determine whether the remaining progenitors contribute exclusively to cTnnT-2a-eGFP+ atrial myocytes or if they also give rise to additional lineages. Moreover, we found additional progenitors contributing exclusively to cTnnT-2a-eGFP-daughters, located in the inflow regions of the heart tube and the posterior lateral plate mesoderm ($n = 24$ lineages, identified as meso GFP- in Fig. 4). Longer imaging periods will be required to determine the identity of these cells.

Together, the live-imaging analysis of lineages reveals that early mesodermal cells harbour plasticity and diversity of fates during gastrulation (Zhang et al, 2021). However, their ability to alternate fates appears to diminish rapidly. In all cardiac lineage trees that displayed two or three fates ($n = 18$), progeny became lineage-restricted early, during migration, before the onset of cTnnT-2a-eGFP+ expression in the embryo (Fig. 7B). Of the 18 bipotent progenitors, 12 became uni-fated after the first generation, 5 after the second generation and 1 after the third generation (Figs. 4 and 7B). These findings are consistent with previous clonal analyses, suggesting that early mesodermal progenitors are rapidly specified into discrete fates shortly after the initiation of gastrulation (Devine et al, 2014; Lescroart et al, 2014; Zhang et al, 2021).

## Migration analysis in lineages reveals hidden patterns of mesodermal cell migration

Previous live analysis of cell trajectories during gastrulation revealed apparently chaotic individual cell movements during migration (Dominguez et al, 2023). Consistent with this analysis, we found that mesodermal cells dispersed extensively during migration (Fig. 8A). We analysed distances between the first two daughters in each lineage (coordinates were taken 10 min after the first cell division (Time point 0) and last time point before the daughters' subsequent cell division (Time point 1) and grand-daughters' (Time point 2) (or final time point at which all granddaughter cells exist. We only considered branches lasting at least 4 h into the cell cycle to allow sufficient cell migration). Distances between daughters and granddaughters gradually increased, reaching considerable distances within a single lineage (up to 331 μm) (Fig. 8A). We noted, however, that distances were highly heterogenous; a proportion of the progeny generated less dispersive daughters at T2, with separating distances of less than 25 μm between them (80 out of 268 calculated distances across 44/59 lineages). One possibility for the observed heterogeneity in these distances is that the daughters generated by uni-fated progenitors exhibit less dispersive migratory paths than those generated by bipotent progenitors (Fig. 8B). To test this hypothesis, we analysed cell movements in lineages, taking advantage of our lineage analysis from the live-imaging data.

Distances between sister cells can be large in non-cardiac lineages, sometimes exceeding 300 μm ($n = 3$). Daughters and granddaughters from bipotent progenitors generating both ExEm and either a myocyte, endocardial, or pericardial fate had greater distances than those from uni-fated ExEm progenitors (Fig. 8C at T2). In the cardiac crescent, bipotent progenitors contributing to

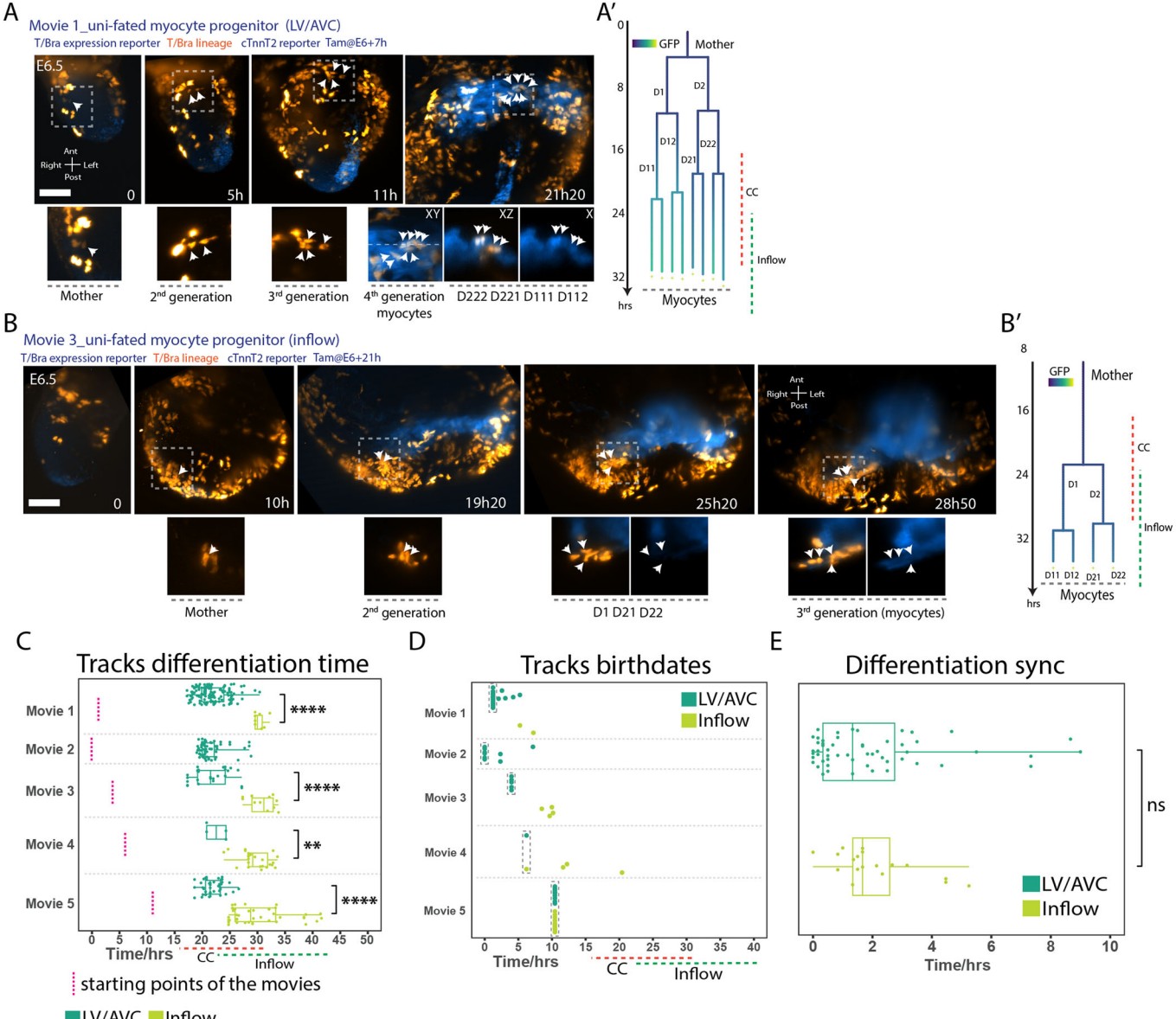

**Figure 6. LV/AVC and Inflow progenitors birthdate and timing of differentiation.**

(**A, B**) Time-lapse images and lineage trees of LV/AVC (**A–A'**) and inflow (**B–B'**) progenitors, coloured by normalized GFP intensity. White arrows (**A, B**) show the cells analysed in the lineage tree (**A'–B'**). Scale bars: 100 μm. (**C**) Time points at which LV/AVC and inflow progenitors exceed the GFP intensity threshold. Each replicate is a Movie (one embryo). Movie 1: LV vs Atria, P value = 2.8e-06. Movie 3: LV vs Atria, P value = 8.4e-07. Movie 4: LV vs Atria, P value = 0.0027. Movie 5: LV vs Atria, P value = 5e-14. (**D**) Birth dates of progenitors contributing to cTnnT-2a-GFP + LV/AVC and inflow myocytes. Each replicate is a Movie (one embryo). (**E**) Time between the first and last progenitor becoming cTnnT-2a-GFP+ in myocyte lineages. Each replicate is a Movie (one embryo). Statistical analyses: Mann–Whitney U test. *P ≤ 0.05, **P ≤ 0.01, ***P ≤ 0.001, ****P ≤ 0.0001. All statistical analyses were performed using the Mann–Whitney U test. For all boxplots, the box boundaries represent the 25th (lower quartile) and 75th (upper quartile) percentiles, with the centre line indicating the median. The whiskers extend to the minimum and maximum values in the dataset. Source data are available online for this figure.

myocyte, endocardial, or pericardial fates but not to the ExEm, produced daughters with greater distances than myocyte, endo-cardial, or pericardial uni-fated progenitors (Fig. 8D at T2). While no differences were seen immediately after division (Fig. 8C,D at T0), as development progressed, sisters with the same fate remained more closely positioned than those with distinct fates (Fig. 8C,D at T2). This finding shows uni-fated progenitors produce more closely positioned cell descendants than bipotent ones.

These differences in dispersion might be due to how similar or different cells' migration paths are. To explore this possibility, we analysed whether sisters sharing the same fate migrated closer together compared to those with divergent fates, using dynamic time warping (DTW) to account for temporal shifts in cell behaviour (Fig. 8E, "Methods"). We calculated the cumulative DTW distances, a distance measure that assigns small distances to similar paths that are temporally unaligned (Fig. 8E).

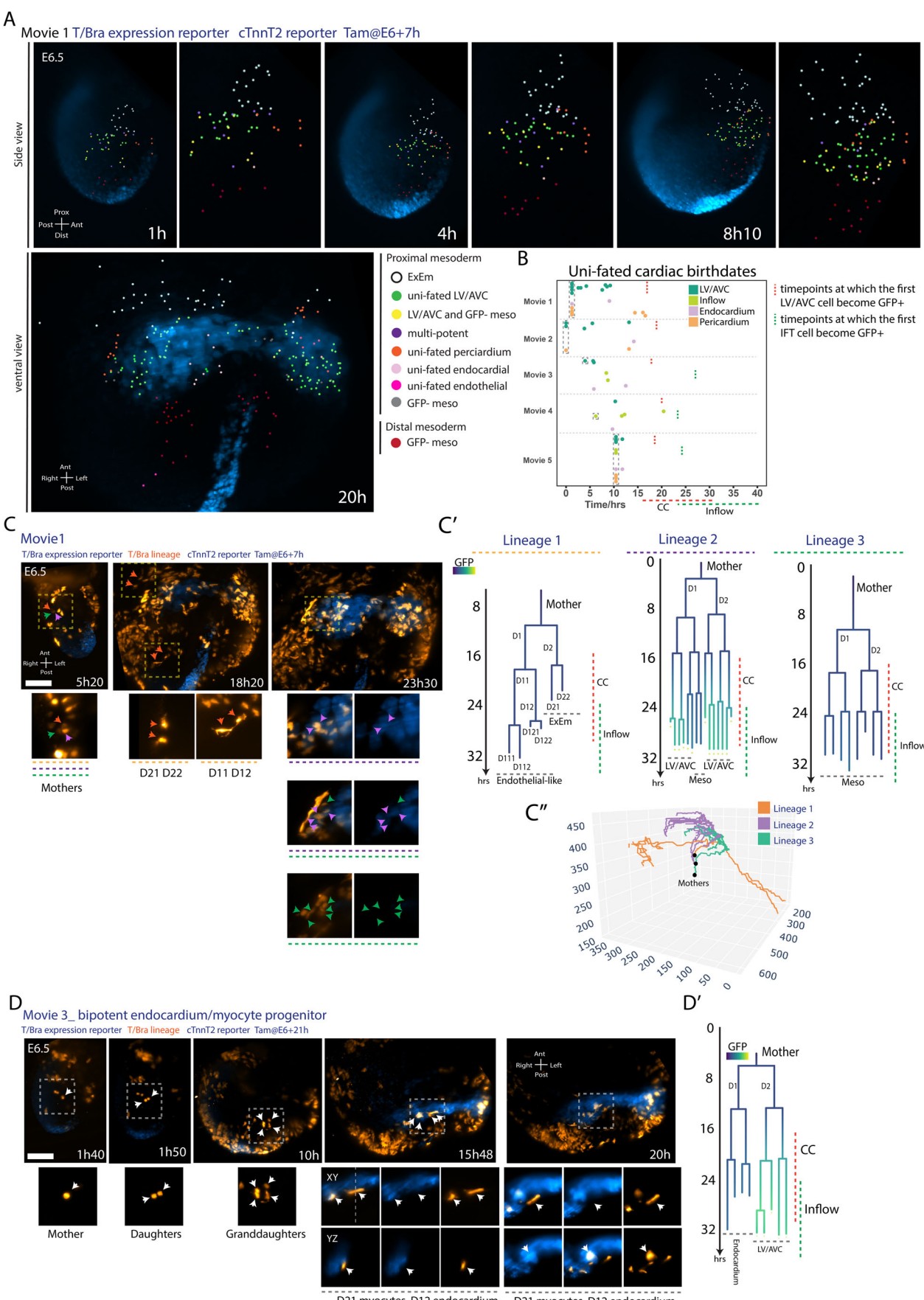

**Figure 7. Fate mapping of cardiac progenitors at the gastrula stage.**

(A) Cells are colour-coded by fate. (B) Birth dates of uni-fated progenitors for LV/AVC and inflow myocytes, endocardium, and pericardium. Cells present at the start of the movie are highlighted in a grey-dotted box. (C) Time-lapse sequences of $T^{nGPF-CreERT2/+}$;$R26R^{tdTomato/+}$; $cTnnT-2a-eGFP$ embryos showing three nearby progenitors contributing to extraembryonic mesoderm (ExEm), endothelial-like cells, LV/AVC, and undefined mesodermal fates. Corresponding lineage trees are colour-coded by normalized GFP intensity (C′), with arrows indicating tracked cells. (C″) 3D trajectories of the three lineages. Scale bar: 100 µm. (D) Time-lapse of $T^{nGPF-CreERT2/+}$;$R26R^{tdTomato/+}$; $cTnnT-2a-eGFP$ embryos showing a bipotent endocardial/myocardium progenitor. Corresponding lineage trees are colour-coded by normalized GFP intensity (D′), with arrows indicating tracked cells. Scale bar: 100 µm. Source data are available online for this figure.

As expected, uni-fated ExEm progenitors had lower DTW distances than bipotent progenitors, which produced cells contributing to both ExEm and one of the cardiac cell types (myocardium, endocardium, or pericardium). This divergence was evident from the first generation, indicating early divergence in the migratory paths of cells contributing to ExEm and cardiac fates (Figs. 8F–I and EV4A,B). For progenitors generating myocardium, endocardium, or pericardium fates but not ExEm, uni-fated progenitors also produced sisters with lower DTW distances compared to bipotent progenitors, becoming significant by the second generation suggesting a more gradual divergence in migratory paths (Figs. 8F,G and EV4C–E).

In 49 out of 168 cases, uni-fated progenitors generated daughter cells with distinct migratory trajectories (log DTW > 4.5 at the second generation, including 32 non-ExEm progenitors). Conversely, in 5 out of 45 cases, bipotent progenitors produced sister cells that followed similar paths but adopted different fates (log DTW < 4.5 at the second generation, including one ExEm-contributing progenitor). These findings suggest that sister cells can diverge in migratory paths while sharing the same fate or follow similar paths but adopt different fates (Fig. EV4E,F,G). However, in the majority of cases, sisters with the same fate exhibited notably similar migration patterns (Figs. 8J–H and EV4H–J).

A permutation test with 100,000 iterations confirmed that uni-fated progenitors produced sisters with more similar trajectories than bipotent progenitors ($P = 9.9e$-07) (Appendix Fig. S8A–D and "Methods"). Plotting DTW values over time showed that sisters with the same fate maintained similar migratory paths throughout, and any observed similarity is not attributed to systematic smaller DTW-contributions towards the end of the trajectory. By contrast, sister cells with distinct fates exhibited divergent migratory behaviours, with those contributing to ExEm showing a rapid and progressively accelerating divergence (increasing DTW values), while non-ExEm cardiac progenitors displayed a more gradual differences in paths (Fig. 9A–D).

The observed similarity in migratory paths of sister cells with shared fates led us to investigate whether these cells retain contact post-division. We quantified this by measuring the ratio of contact duration to total tracking duration for generation 1 (Fig. 9E; Appendix Fig. S9A–A' and "Methods"). Results revealed that daughter cells retain cell-cell contact immediately following cell division in 63% of cases ($n = 166$). Bipotent progenitors produced daughter cells with the shortest contact duration, displaying robust separating behaviours (Fig. 9F–H). By contrast, uni-fated pericardial progenitors maintained the longest contact duration, suggesting sustained interaction, while LV/AVC uni-fated progenitors showed a similar trend, albeit not statistically significant. These findings raise the possibility of correlation between progenitor fate and early contact behaviour.

## Emergence of distinct mesodermal cell behaviours during gastrulation

We next determine if distinct cell fates adopt different migratory behaviours and when these differences emerge in the mesoderm. Dispersion analysis during migration showed no significant differences between LV/AVC, pericardium, and endocardial fates, though atrial sister cells clustered more closely, while ExEm cells were more dispersed (Fig. 10A,B; Table EV3). During heart tube formation, endocardial cells displayed greater dispersion than LV/AVC and pericardial cells, similar to ExEm cells (Fig. 10C; Table EV3). This finding suggests that endocardial cells adopt distinct migratory behaviours at later stages, leading to their further dispersion during heart tube formation. Consistent with this analysis, we observed an endocardial clone spreading in both the ventricular and inflow component of the heart tube (Fig. 10D). Supporting this observation, a concurrent analysis revealed behavioural divergence in endothelial/endocardial precursors, with endothelial progenitors exhibiting higher speeds (Sendra et al, 2024, Preprint).

These findings led us to investigate cell speed patterns according to cell fate using our annotations. After aligning the movies to a unified timeline (Fig. 6C), we analysed cell speed in 5-hour intervals, revealing distinct trends (Table EV4). During the 0–5 h interval, ExEm progenitors exhibited the slowest speeds (Dominguez et al, 2023; Saykali et al, 2019), while bipotent, pericardial, and LV/AVC progenitors showed similar speed. Endocardial cells initially migrated slowly but showed a marked increase in speed during the 10–15 h interval, whereas pericardial progenitors exhibited a decrease in speed over the same period (Fig. 10E–G; Appendix Fig. S10). At this stage, endocardial cells began transmigrating through the mesoderm, preceding cTnnT-2a-eGFP + reporter activation and adopting a spindle-like morphology (Figs. 11A). Some were already positioned near the endoderm before cTnnT-2a-eGFP+ activation in both the LV and atrial regions (Fig. EV5A,D, $n = 22$). From around 25 h, other endocardial cells crossed the cTnnT-2a-eGFP+ myocardium, contributing to the left ventricle (LV) and atrial components of the heart tube (Figs. 11A and EV5B,C, $n = 13$). Similarly, endothelial-like cells exhibited spindle-like shapes and increased their speeds during the 10–15 h interval, aligning with the timing of endocardial cell acceleration and cell shape changes (Fig. 10E,G; Appendix Figs. S10 and S11A) consistent with those reported in Sendra et al (Sendra et al, 2024, Preprint).

Finally, ExEm progenitors showed a significant increase in speed and higher tortuosity during later periods of heart tube formation (25–40 h interval), indicating more meandering and chaotic migratory paths (Fig. 11B–D; Appendix Fig. S11B). In contrast, LV/AVC and pericardial cells displayed more directed movements,

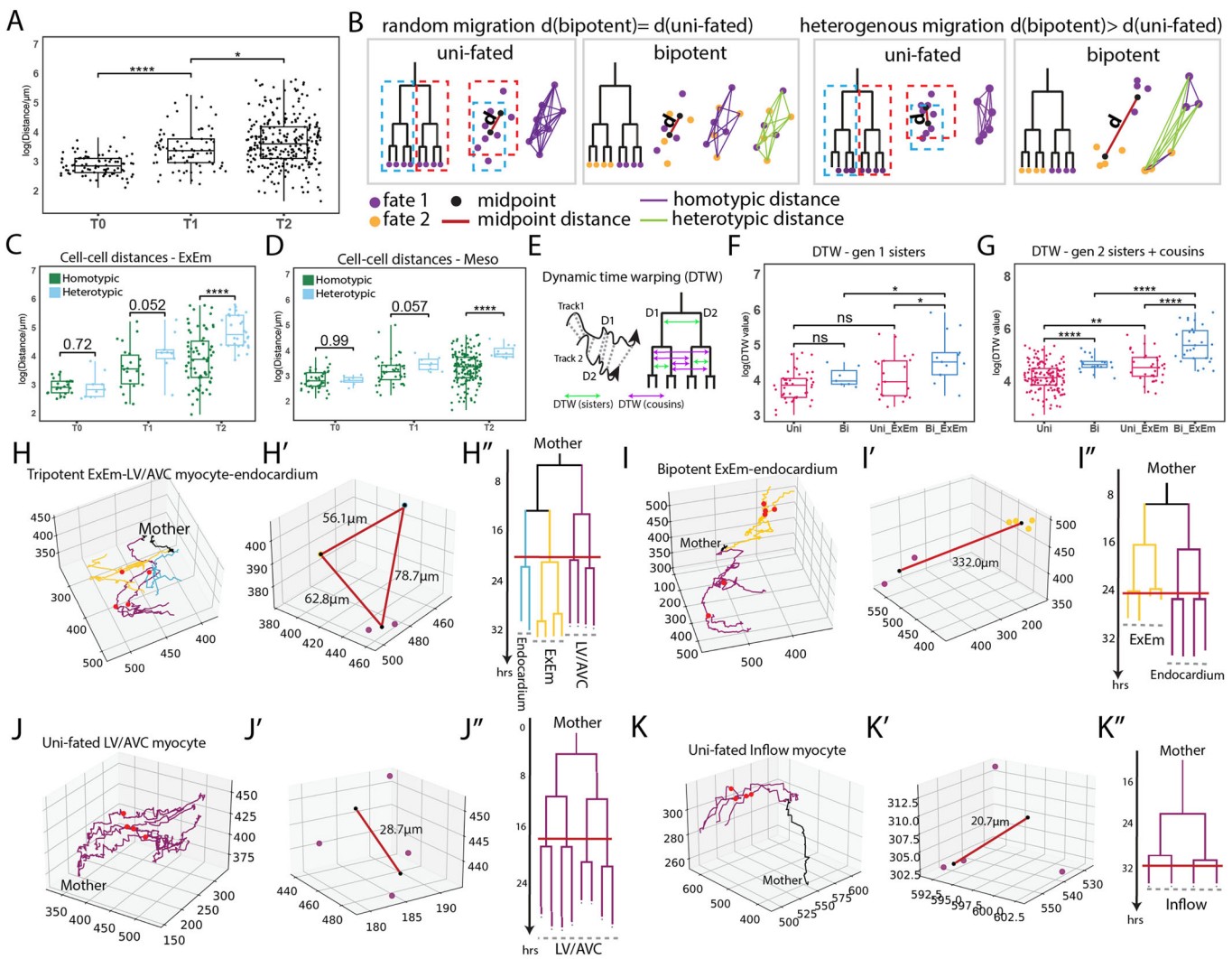

**Figure 8. Cell migration analysis in lineages.**

(A) Dispersion analysis over time shows the distances between all daughters and granddaughters in all the cardiac lineages at time points T0, T1, and T2. T0 vs T1, P value = 1.5e-05. T1 vs T2, P value = 0.071. Data are from the 5 datasets (Movies EV1–5). (B) Hypothesis schematic: random migration predicts equal dispersion for uni-fated and bipotent progenitors; heterogeneous trajectories predict less dispersion for uni-fated progenitors. "Midpoint distance" is the Euclidean distance between the midpoints of D1's and D2's daughters. Homotypic and heterotypic distances are between cells of the same or different fate. (C, D) Dispersion analysis comparing homotypic and heterotypic distances in lineages with (C) and without (D) ExEm cells at T0, T1, and T2. The scores are presented on a logarithmic scale for better visualization of differences across categories. (C) T0: Homo vs Het, P value = 0.72. T1: Homo vs Het, P value = 0.99. T2: Homo vs Het, P value = 0.99. (D) T0: Homo vs Het, P value = 0.72. T1: Homo vs Het, P value = 0.052. T2: Homo vs Het, P value = 1.9e-06. Data are from the five datasets (Movies EV1–5). (E) Dynamic time warping (DTW) quantifies similarities between two migration paths. (F, G) DTW scores comparing D1 and D2 sister tracks up to their next division (F) and for 2nd generation descendant tracks (sisters [D11–D12, D21–D22] and cousins [D11–D21, D11–D22]) (G) under the following conditions: Uni: Both daughters adopt the same cardiac fate. UniExEm: Both daughters adopt the same ExEm fate. Bi: Daughters adopt distinct fates, with none contributing to the ExEm. BiExEm: Daughters adopt distinct fates, with at least one contributing to the ExEm. The scores are presented on a logarithmic scale for better visualization of differences across categories. (F) Uni vs Bi, P value = 0.15. Uni vs UniExEm, P value = 0.2. BiExEm -Bi, P value = 0.024. UniExEm-BiExEm, P value = 0.032. (G) Uni vs Bi, P value = 0.0052. Uni vs UniExEm, P value = 0.021. BiExEm-Bi, P value = 2.1e-06. UniExEm-BiExEm, P value = 1e-04. Results were statistically analysed using a Mann–Whitney test. Data are from the five datasets (Movies EV1–5). (H–K) Examples of trajectories and corresponding midpoint distances (H'–K') and lineage trees (H''–K''). Red dots in H'–K' mark sampling points for midpoint analysis. All statistical analyses were performed using the Mann–Whitney U test. *P ≤ 0.05, **P ≤ 0.01, ***P ≤ 0.001, ****P ≤ 0.0001. For all boxplots, the box boundaries represent the 25th (lower quartile) and 75th (upper quartile) percentiles, with the centre line indicating the median. The whiskers extend to the minimum and maximum values in the dataset. Source data are available online for this figure.

aligning with the onset of epithelialization and tissue folding in the cardiac crescent to form the heart tube, with pericardial tissue largely following these directed movements taking place at these stages (Dominguez et al, 2023).

## Discussion

Our findings illustrate the migration of cardiac mesodermal lineages during gastrulation (summarized in Fig. 12A–C). Using

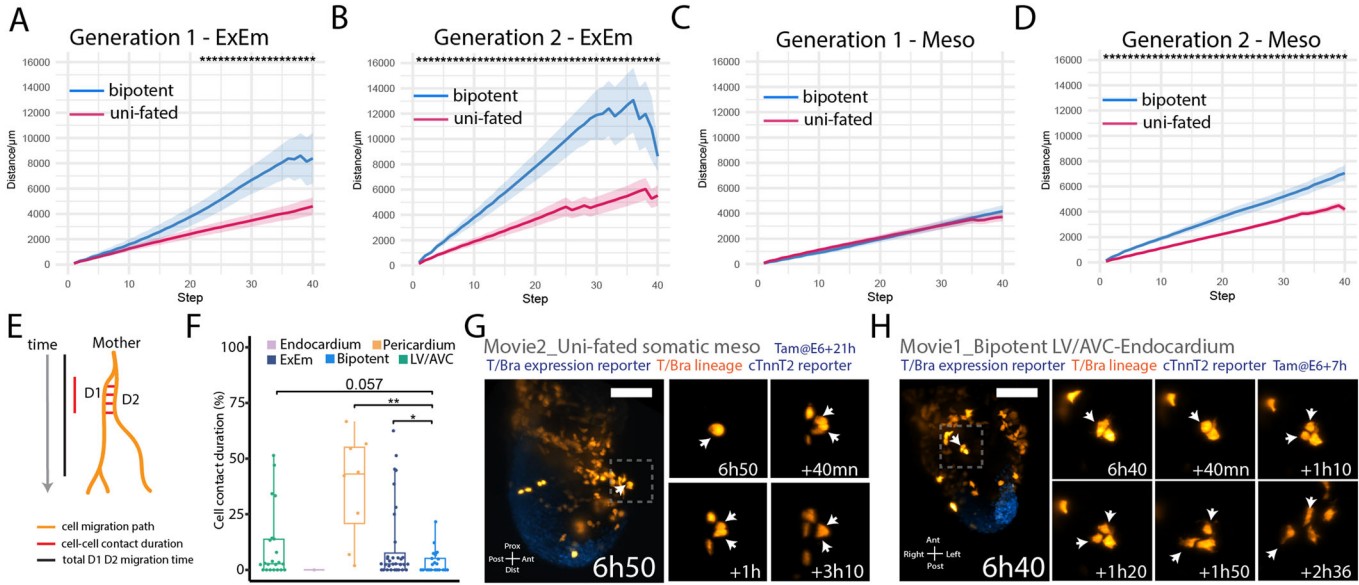

**Figure 9. Sister cells sharing the same cardiac fate sustain cell-cell interactions.**

(A–D) Mean DTW values over time for D1, D2 (generation 1) and D11, D12, D21, D22 (generation 2) daughters from uni-fated and bipotent progenitors, with ExEm cell contribution (A, B) or without (C, D). Shaded areas indicate SE; significant differences for each step are marked with a star. (E) Cell-cell contact duration analysis: ratio of contact time to track duration within the first 16 h. Short tracks (< 4 h) excluded. (F) Proportion of time sister cells with similar or distinct fates remain in contact. LV vs Bi, P value = 0.057. Pericardium vs Bi, P value = 0.0048. ExEm vs Bi, P value = 0.025. Data are from the five datasets (Movies EV1–5). (G, H) Time-lapse images of uni-fated pericardial (G) and bipotent LVAVC/Endocardium (H) progenitors. White arrows show daughter cells retaining cell-cell contact (G) or showing separating movements following division (H). Scale bar: 100 μm. *P ≤ 0.05, **P ≤ 0.01, ***P ≤ 0.001, ****P ≤ 0.0001. All statistical analyses were performed using the Mann–Whitney U test. For all boxplots, the box boundaries represent the 25th (lower quartile) and 75th (upper quartile) percentiles, with the centre line indicating the median. The whiskers extend to the minimum and maximum values in the dataset. Source data are available online for this figure.

live-imaging and single-cell tracking, we reconstructed cardiac mesodermal lineages and the migratory paths of cells over extended periods encompassing gastrulation and heart tube morphogenesis (~40 h). Culturing embryos in larger volumes of media culture using an open-top light-sheet microscope and in-house preparation of high-quality rat serum (Cockroft et al, 1992; Takahashi et al, 2014) were critical for these experiments. Moreover, while we utilised a novel cTnnT-2a-GFP transgenic line to trace myocytes up to the heart tube stage, incorporating additional reporters, for example, identifying the craniopharyngeal mesoderm generating the RV and OFT, or subpopulations within the intra-embryonic mesoderm, will provide further insights into cardiac and mesoderm development in future studies.

Our live imaging suggests that the cardiac crescent (or first heart field -FHF- and juxta-cardiac field partially overlapping the FHF) (Liang et al, 2013; Spater et al, 2013; Tyser et al, 2021; Watanabe et al, 2023) is predominantly destined to contribute to the LV/AVC. In contrast, the atria arise at different times during gastrulation (Bardot et al, 2017; Gonzalez et al, 2022; Ivanovitch et al, 2021; Lescroart et al, 2014; Zhang et al, 2021). These findings suggest that an early segregation of the ventricular and atrial cells has been conserved during evolution; an early segregation of these progenitor populations was previously shown at single-cell resolution in the zebrafish (Garcia-Martinez and Schoenwolf, 1993; Keegan et al, 2004; Stainier et al, 1993; Yutzey and Bader, 1995). They also align with in vitro differentiation experiments demonstrating that modulating pathways known to induce mesoderm can generate molecularly distinct mesoderm favouring

the generation of ventricular or atrial-like cardiomyocytes respectively (Dark et al, 2023; Lee et al, 2017; Mendjan et al, 2014; Schmidt et al, 2023; Yang et al, 2022).

The limited initial spread of T/Bra lineage-positive clones suggests that mesodermal cells quickly become confined to specific locations during gastrulation (Lescroart et al, 2014; Devine et al, 2014; Meilhac et al, 2004a), while large clones spanning multiple heart compartments may result from earlier induction events, possibly at the epiblast stage (Meilhac et al, 2004a). However, this restricted distribution at the heart tube stage does not necessarily imply early segmental regionalization within a strictly predetermined mesoderm. Additional clonal spreading (Meilhac et al, 2004b; Aanhaanen et al, 2009), may occur during later stages as the heart tube undergoes oriented growth, reshaping and looping. Our live imaging did not cover these later stages, leaving this possibility unconfirmed.

Our analysis showed that most early proximal mesoderm progenitors contributed to the LV/AVC, while fewer gave rise to endocardial and pericardial cells. This imbalance likely explains the higher proportion of myocytes compared to endocardial and pericardial cells in the cardiac crescent, suggesting that not all progenitors in the early proximal mesoderm are equivalent. Instead, specific biases in cell fate may be regulated to produce the correct cell types in the right proportions. Consistent with this idea, concurrent lineage tracing results proposed that the endothelial/endocardial fates undergo fast lineage divergence during gastrulation (Sendra et al, 2024, Preprint).

Our analysis revealed that the early proximal mesoderm contains bipotent progenitors capable of generating LV/AVC,

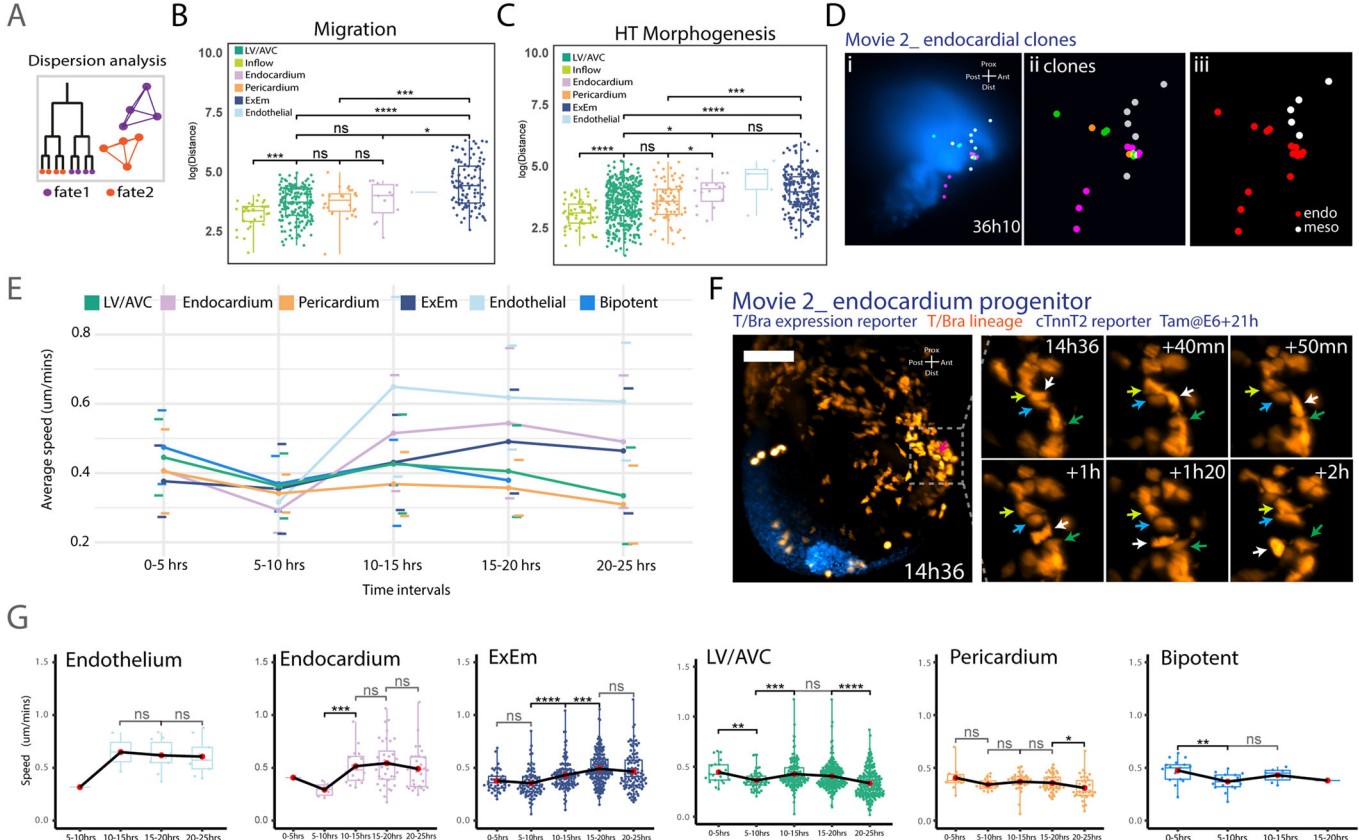

**Figure 10. Cardiac mesodermal cells speed and dispersion analysis.**

(A) Diagram of dispersion analysis for sister and cousin cell distances with similar fates. (B, C) Cell dispersion at migration end (B) and during heart tube formation (C). Data are from the five datasets (Movies EV1–5). (B) Atria vs LV, P value = 0.00016. LV vs Pericardium, P value = 0.98. LV vs Endocardium, P value = 0.44. LV vs ExEm, P value = 1.5e-11. Pericardium vs Endocardium, P value = 0.47. Pericardium vs ExEm, P value = 0.00011. Endocardium vs ExEm, P value = 0.035. (C) Atria vs LV, P value = 1.9e-05. LV vs Pericardium, P value = 0.54. LV vs Endocardium, P value = 0.013. LV vs ExEm, P value = 1.4e-11. Pericardium vs Endocardium, P value = 0.039. Pericardium vs ExEm, P value = 0.00017. Endocardium vs ExEm, P value = 0.94. Data are from the 5 datasets (Movies EV1–5). (D) (i–ii) Endocardial clones, each in a different colour. (iii) Endocardial cells highlighted in red. (E) Mean cell speeds per fate calculated across 5-h intervals. (F) Time-lapse images of endocardial progenitor (indicated by a white arrow) and non-endocardial progenitors (indicated by yellow, blue and green arrows). Scale bar: 100 μm. (G) Cell speeds per fate were calculated across 5-h time periods. Movies were temporally aligned as shown in Fig. 5B. Data are from the five datasets (Movies EV1–5). ExEm: 0–5 h vs 5–10 h, P value = 0.16. 5–10 h vs 10–15 h, P value = 2.9e-06. 10–15 h vs 15–20 h, P value = 8.4e-05. 15–20 h vs 20–25 h, P value = 0.045. LV/AVC: 0–5 h vs 5–10 h, P value = 0.0012. 5–10 h vs 10–15 h, P value = 0.00075. 10–15 h vs 15–20 h, P value = 0.13. 15–20 h vs 20–25 h, P value = 3e-08. Pericardium: 0–5 h vs 5–10 h, P value = 0.091. 5–10 h vs 10–15 h, P value = 0.079. 10–15 h vs 15–20 h, P value = 0.24. 15–20 h vs 20–25 h, P value = 0.014. Bi: 0–5 h vs 5–10 h, P value = 0.0017. 5–10 h vs 10–15 h, P value = 0.065. 10–15 h vs 15–20 h, N/A. *P ≤ 0.05, **P ≤ 0.01, ***P ≤ 0.001, ****P ≤ 0.0001. All statistical analyses were performed using the Mann–Whitney U test. For all boxplots, the box boundaries represent the 25th (lower quartile) and 75th (upper quartile) percentiles, with the centre line indicating the median. The whiskers extend to the minimum and maximum values in the dataset. Source data are available online for this figure.

pericardium, endocardium, and extraembryonic mesoderm. However, many mother cells produced at least one progeny whose fate could not be determined, making it difficult to confirm their potency and accurately estimate the incidence of bipotent cells due to challenges in reconstructing complete lineage histories. Addressing these limitations will require future studies to use more advanced microscopy techniques with higher contrast-to-noise ratio for improved visualization in deeper tissue layers.

Despite these challenges, our reconstruction of 227 early mesodermal lineage trees across up to five generations from five live-imaging datasets of gastrulating mouse embryos provides valuable insights into cell potency. Among the progenitors with confirmed fates, we identified 12 bipotent/tripotent and 29 unifated progenitors contributing to LV/AVC myocytes. These findings suggest that while many mesodermal progenitors are

committed to a myocyte fate, a substantial proportion of bipotent progenitors also exist. These bipotent progenitors became rapidly restricted into unique cardiac fates during migration - prior to the establishment of the cardiac crescent and onset of myocyte differentiation. The observation of short-lived multipotent cardiac progenitors is consistent with clonal analysis results of *Hand1+* and *Mesp1+* mesodermal progenitors (Devine et al, 2014; Lescroart et al, 2014; Zhang et al, 2021).

Previous migration analysis noted opposing cell density and motility gradients in the mesoderm (Dominguez et al, 2023). According to this model, cells continually exchange neighbours and disperse widely until their movements gradually diminish, eventually settling in positions and fates as gastrulation concludes. Our live-imaging analysis builds upon these findings, offering an extended evaluation of the migratory paths of cells in relation to

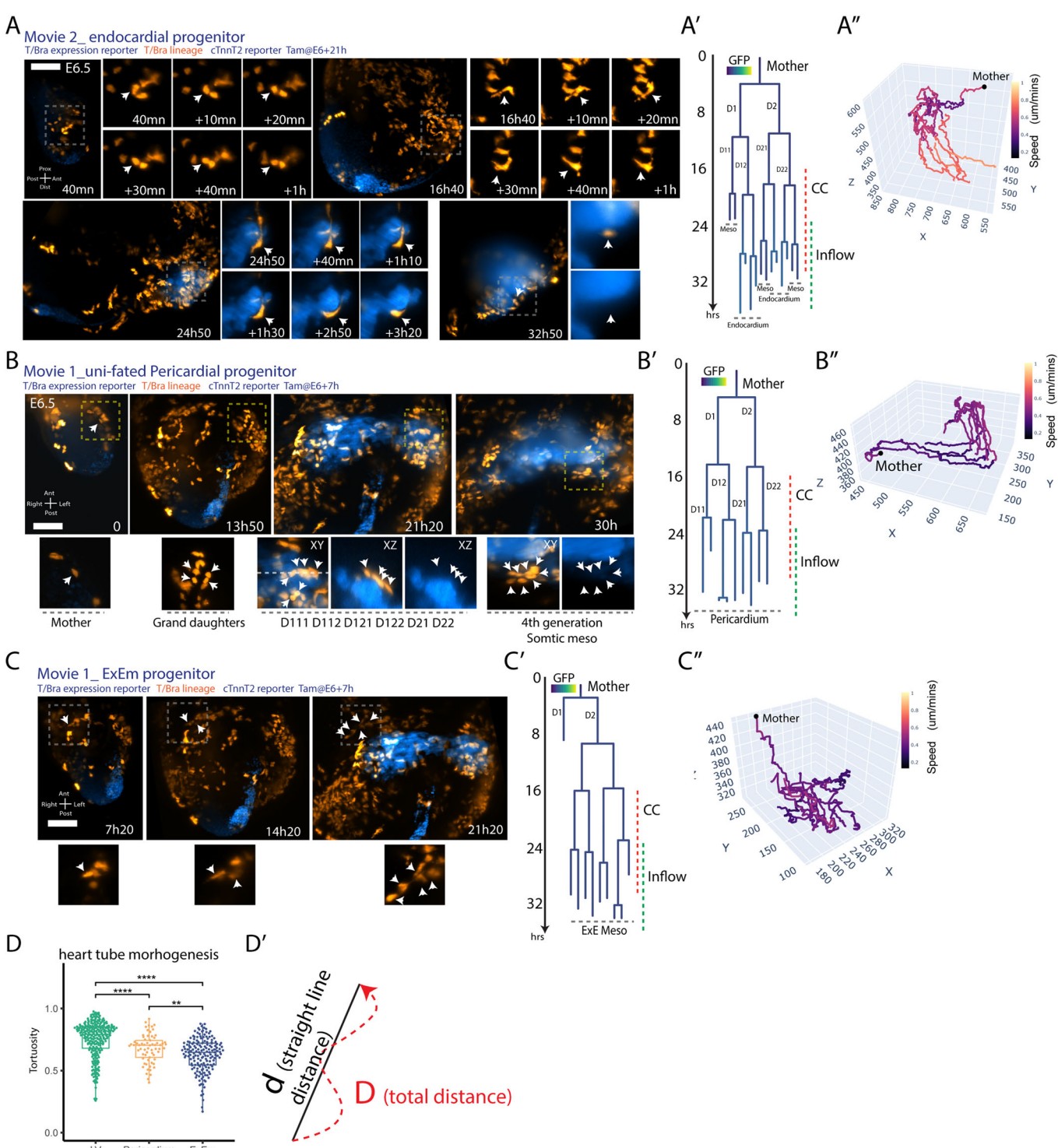

**Figure 11. Cardiac mesodermal cell behaviours analysis.**

(**A–C**) Time-lapse images of endocardial (**A**), uni-fated pericardial (**B**), and ExEm (**C**) progenitors. Corresponding lineage trees, coloured by GFP intensity, in (**A′**, **B′**, **C′**) and 3D speed plots in (**A″**, **B″**, **C″**). White arrows in (**A–C**) indicate cells in lineage trees (**A′**, **B′**, **C′**). Scale bar: 100 μm. (**D–D′**) Tortuosity analysis comparing ExEm and pericardial cells during heart tube formation stages (corresponding to the 25–42 h interval in Fig. 5B). Data are from the 5 datasets (Movies EV1–5). LV vs Pericardium = P value = 3.7e-05. LV vs ExEm, P value < 2.22e-16. Pericardium vs ExEm, P value = 0.0031. LV/AVC left ventricle/atrioventricular canal, HT heart tube. *P ≤ 0.05, **P ≤ 0.01, ***P ≤ 0.001, ****P ≤ 0.0001. All statistical analyses were performed using the Mann–Whitney U test. For all boxplots, the box boundaries represent the 25th (lower quartile) and 75th (upper quartile) percentiles, with the centre line indicating the median. The whiskers extend to the minimum and maximum values in the dataset. Source data are abuialable online for this figure.

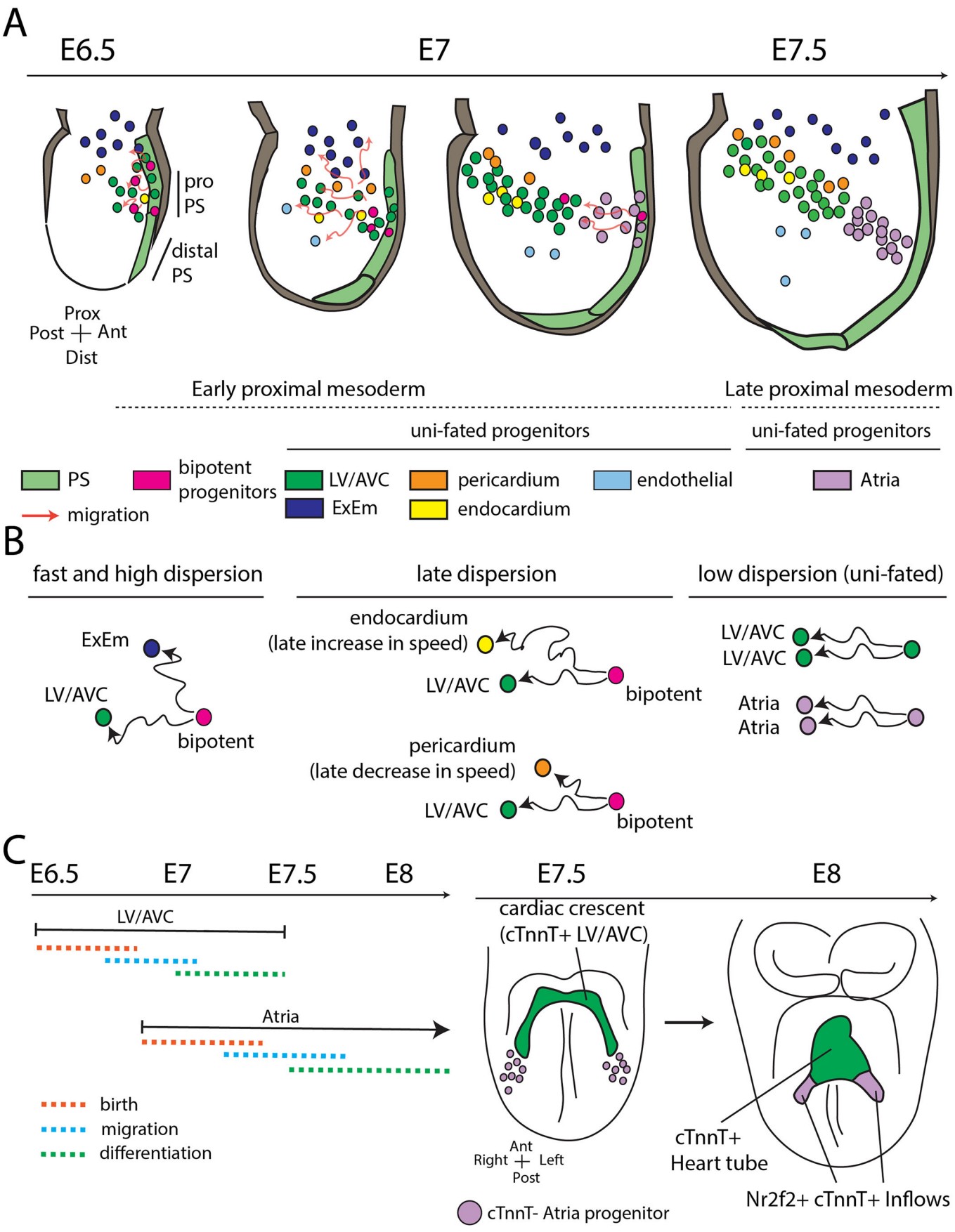

**Figure 12. Summary model of early heart development.**

(A) Cells located in the early proximal mesoderm contribute to the ExEm, LV/AVC myocardium, endocardium, and pericardium in unequal proportions, with the majority of progenitors giving rise to the LV/AVC myocardium. Bipotent progenitors contributing to all four cell types are initially present. The late proximal mesoderm contributes to the atrial/inflow myocardium. (B) Bipotent progenitors contributing to the ExEm exhibit the fastest and most dispersed migrations. In contrast, those giving rise to the endocardium, LV/AVC, and pericardium follow a more gradual divergence. Unipotent progenitors show more directed migration patterns. (C) LV/AVC progenitors differentiate early to form the cardiac crescent, while atrial progenitors later contribute to the formation of the heart tube's Nr2f2+ inflow tract during morphogenesis. ExEm Extraembryonic mesoderm, LV left ventricle, AVC atrioventricular canal, PS primitive streak, Prox proximal, Dist distal, Ant anterior, Post posterior.

their future fates in the heart tube. The analysis revealed that progenitors contributing to LV/AVC and atrial myocytes remain as separated cell populations throughout migration, establishing two distinct progenitor domains in the heart tube without mixing.

During development, the pericardial and myocardial layers, along with the underlying plexus of elongated endocardial cells, emerge in close proximity within the cardiac crescent (Dominguez et al, 2023; DeRuiter et al, 1992). The migratory paths of progeny contributing to these three distinct cardiac fates appear more predictable than previously understood, with sister cells sharing the same fate often following parallel trajectories over extended periods. Bipotent progenitors, particularly those giving rise to ExEm, displayed the fastest and most widely dispersed migration patterns among their daughter cells. In contrast, bipotent progenitors contributing to endocardial, LV/AVC and pericardial cells showed a more gradual divergence in migratory paths. This divergence was characterized by a late-stage increase in speed for endocardial cells, while pericardial cells exhibited a concurrent decrease in speed relative to LV/AVC progenitors. These gradual trajectory shifts likely reflect late-stage cellular behavioural changes associated with the process of fate commitment during gastrulation.

Future research will need to explore the potential mechanisms restricting migratory paths and determine whether early mesodermal progenitors display similar migratory behaviours due to shared initial internal states and/or a cell-specific response to environmental cues. Previous studies in zebrafish indicated that G-protein-coupled receptor signalling, a hallmark of chemokine signalling, regulates heart progenitor movements during gastrulation (Kupperman et al, 2000; Scott et al, 2007; Zeng et al, 2007). Moreover, BMP, Nodal, and FGF morphogen gradients regulate cell migration independently of cell fate (Ciruna and Rossant, 2001; Guo and Li, 2007; Nutt et al, 2001; von der Hardt et al, 2007; Yang et al, 2002). Thus, it seems that mesodermal cells respond to morphogen cues with precision, providing determinism to the morphogenetic cell behaviours. Simultaneously, they demonstrate plasticity regarding their final fate (Tam et al, 1997). While progenitors are seen giving rise to only one cardiac cell type, they could potentially generate additional cardiac fates when no longer constrain by positional cues. We propose that achieving a delicate balance between determinism and plasticity is essential to ensure robust morphogenesis. This balance enables cells to follow specific developmental pathways while also maintaining the flexibility needed to adapt to changing external cues.

Together, our live-imaging analysis of migration and cardiac lineages provides evidence that some regulation of directionality of cell movements and fate allocation may exist early within the mesoderm. The findings have broader implications for our understanding of organogenesis since they address how initial differences between progenitors and signalling cues may ultimately affect the fate and movements of cells.

# Methods

### Reagents and tools table

| Reagent/resource | Reference or source | Identifier or catalogue number |
|---|---|---|
| **Experimental Models** | | |
| Tnnt2em1(eGFP)Jcs (*M. musculus*) | This study | JAX Stock No. 040309 |
| TnEGFP-CreERT2/+ (*M. musculus*) | Genesis. 2013 Feb 25;51(3):210–218 | Acc. No. CDB0604K: http://www.cdb.riken.jp/arg/mutant%20mice%20list.html |
| (Gt(ROSA)26Sortm14(CAG-tdTomato)Hze /+ (*M. musculus*) | Nat Neurosci 13(1):133-40 | JAX Stock No. 007914 |
| BRE:H2B-Turquoise BMP reporter line /+ (*M. musculus*) | PLoS Biol. 2021 May 17;19(5):e3001200 | https://www.addgene.org/171499/ |
| C57BL/6 J /+ (*M. musculus*) | Charles River | na |
| **Antibodies** | | |
| mouse anti-cTNT2 | Thermo Fischer Scientific Systems | MS295P0 RRID:AB_61807 |
| Secondary Antibody Goat anti Mouse 568 | Thermo Fischer Scientific Systems | A11031 |
| Mouse anti-CD31/PECAM-1 | 553370 BD Pharmingen clone MEC 13.3 | Q08481 |
| **Chemicals, enzymes, and other reagents** | | |
| Rapiclear 1.52 | Sunjin lab | #RC152001:10 ml |
| Matrigel | Corning | 356255 |
| DMEM, low glucose, pyruvate, no glutamine, no phenol red | Thermo Fischer Scientific Systems | 11580406 |
| PVDF filter 0.22 μm | Millipore Millex | SLGV033RS |
| Donkey Serum | abcam | Ab7475 |
| Rat Serum | Made in-house (UCL) | |
| HEPES | GIBCO | 15630056 |
| Penicillin streptomycin | Biological Industries | 030311B |
| Sodium Pyruvate | GIBXO | 11360070 |
| GlutaMAX | GIBCO | 35050061 |

| Reagent/resource | Reference or source | Identifier or catalogue number |
|---|---|---|
| DAPI | Thermo Fisher | D1306 |
| RNaseZap | Thermo Fisher | AM9782 |
| Proteinase K | Thermo Fisher | EO0491 |
| Tween-20 | Merck | P1379 |
| Triton X-100 | Merck | T8787 |
| Tamoxifen | Sigma-Aldrich | T5648 |
| Paraformaldehyde | Thermo Fisher | 043368.9M |
| Corn oil | Merck | C8267-500ML |
| **Software** | | |
| Fiji | https://imagej.net/software/fiji/ Schindelin et al, 2012 | |
| BigStitcher | Horl et al, 2019 | |
| MaMuT | Wolff et al, 2018 | |
| Python | https://www.python.org/ | |
| ggplot2(R4.0) | https://ggplot2.tidyverse.org/ Wickham et al, 2016 | |
| Adobe Illustrator 2024 | | |
| Imaris | Oxford Instruments | |
| **HCR** | | |
| HCR Custom Probe Set | Molecular Instrument | Organism: Mouse (Mus musculus), Probe set: Custom: TnnT2, Probe set size: 20, HCR™ Amplifier: B2 |
| HCR Custom Probe Set | Molecular Instrument | Organism: Mouse (Mus musculus), Probe set: Custom: Nr2f2, Probe set size: 20, HCR™ Amplifier: B4 |
| HCR Custom Probe Set | Molecular Instrument | Organism: Mouse (Mus musculus), Probe set: Custom: HCN4, Probe set size: 20, HCR™ Amplifier: B1 |
| HCR Custom Probe Set | Molecular Instrument | Organism: Mouse (Mus musculus), Probe set: Custom: Hand1, Probe set size: 20, HCR™ Amplifier: B3 |
| HCR Custom Probe Set | Molecular Instrument | Organism: Mouse (Mus musculus), Probe set: Custom: Tal1, Probe set size: 20, HCR™ Amplifier: B1 |
| HCR Custom Probe Set | Molecular Instrument | Organism: Mouse (Mus musculus), Probe set: Custom: Hand1, Probe set size: 20, HCR™ Amplifier: B4 |
| HCR Custom Probe Set | Molecular Instrument | HCR™ Amplifier: B1, Amplifier fluorophore |
| HCR Custom Probe Set | Molecular Instrument | HCR™ Amplifier: B2, Amplifier fluorophore |

| Reagent/resource | Reference or source | Identifier or catalogue number |
|---|---|---|
| HCR Custom Probe Set | Molecular Instrument | HCR™ Amplifier: B3, Amplifier fluorophore |
| HCR Custom Probe Set | Molecular Instrument | HCR™ Amplifier: B4, Amplifier fluorophore: |
| **Other** | | |
| Viventis LS1 | Leica | |
| SP5 Leica confocal microscope | Leica | |
| Tokai Hit thermoplate | Tokai hit | |
| Olympus FVMPE-RS microscope | Olympus | |

## Experimental model and subject details

All animal procedures were performed in accordance with the Animal (Scientific Procedures) Act 1986 under the UK Home Office project licences PP8527846 (Crick) and PP3483414 (UCL) and PIL IA66C8062.

## Mouse strains

The $T^{nEGFP-CreERT2/+}$ (MGI:5490031) lines were obtained from Hiroshi Sasaki (Imuta et al, 2013). The $R26^{Tomato\ Ai14/\ Tomato\ Ai14}$ $(Gt(ROSA)26Sor^{tm14(CAGtdTomato)Hze}$ (MGI:3809524) (Madisen et al, 2010), were obtained from the Jackson Laboratory. The $(no\ gene)^{Tg(BRE:H2B;Turquoise)Jbri}$ BMP reporter line was generated by the Briscoe laboratory previously (Ivanovitch et al, 2021).

## Generation of the cTnnT-2a-eGFP line

The C-terminal tagging of cardiac troponin cTnnt2 with eGFP was generated in the Genetic Modification Service using a CRISPR-Cas9 strategy. This editing was performed by co-transfection of a Cas9-gRNA vector and a donor vector comprising the T2A self-cleaving peptide and eGFP into B6N 6.0 embryonic stem cells using Lipofectamine 2000. The donor vector contained a 786 bp insert of T2A-eGFP with 1 kb homology arms either side. The guide sequence used was 5'-TTTCATCTATTTCCAACGCC-3'. Two correctly targeted ESC clones were microinjected into blastocysts which were then transferred into the uterus of pseudo pregnant BRAL (C57BL/6 albino) females. Generated chimeras were crossed to BRAL and F0 chimera offspring were initially screened for the proper integration of T2A-eGFP by Sanger sequencing. The F0 were again crossed to BRAL to confirm germline transmission of the mutation and F1 mice were validated by Sanger sequencing. The line has been deposited to JAX (Stock No. 040309).

## Immunostaining

Embryos were dissected in 1× Phosphate buffered saline (PBS, Invitrogen), fixed for 4 h in 4% PFA at room temperature, washed with 1× PBS-0.1% Triton (0.1% PBS-T) and permeabilised in 0.5% PBS-T. Embryos were blocked with 1% donkey serum for 1 h,

incubated overnight at 4 °C with antibodies diluted in 0.1% PBS-T: mouse anti-cTnnT (1:250, Thermo Fischer Scientific Systems, MS295P0) and CD31 (553370 BD Pharmingen clone MEC 13.3). Embryos were washed in 0.1% PBS-T at room temperature and incubated overnight at 4 °C with secondary antibodies coupled to 556 fluorophores (1:200, Molecular Probes). After washing with 0.1% PBS-T, embryos were then incubated overnight at 4 °C with DAPI (1:1000). Embryos were washed in 0.1% PBS-T and mounted in Rapiclear. Confocal images were obtained on an inverted Sp5 confocal microscope with a ×20 oil objective (for early E7.5 embryos) or a 10× air objective (0.4 NA) (for E12.5 hearts) at a 2048 × 2048 pixels dimension with a z-step of 1–5 μm. Embryos were systematically imaged throughout from top to bottom. Images were processed using Fiji software (Schindelin et al, 2012).

## HCR

Whole-mount HCR fluorescent in situ hybridizations (ISH) were conducted using the HCR™ RNA-FISH Technology. Wild-type embryos were fixed in 4% PFA overnight and dehydrated in graded MeOH/PBS-Tween 0.1% washes (25%, 50%, 75%, 2 × 100%) on ice before storage at −20 °C. Embryos were rehydrated through a series of graded MeOH/PBS-Tween washes, followed by digestion with proteinase K (20 mg/mL) for 5 min (cardiac crescent stage) or 10 min (heart tube stage). Embryos were washed twice in PBS-Tween 0.1% and post-fixed with 4% PFA for 20 min at room temperature. Post-fixation, embryos were washed in PBS-Tween (3 × 5 min) and pre-incubated in a 50/50 hybridization buffer and PBS-Tween mix for 5 min. Embryos were then incubated with probe hybridization buffer preheated to 37 °C and hybridized overnight at 37 °C with specific probes (1 μM). Embryos were washed in preheated probe wash buffer at 37 °C (2 × 5 min, 2 × 30 min), followed by a 10 min wash in a 50/50 probe wash buffer and 5× SSCT at 45 °C. After three 5 min washes in 5× SSCT at room temperature, embryos were incubated in 1 mL amplification buffer for 30 min at room temperature. Fluorescent hairpin solutions (3 μM each) were snap-cooled and prepared in amplification buffer, and embryos were incubated overnight in the dark at room temperature. Excess hairpins were removed by washing with 5× SSCT (2 × 5 min, 2 × 30 min, 1 × 5 min) and stored at 4 °C protected from light. Embryos were stained with DAPI overnight, washed (3 × 10 min, 5× SSCT), and mounted on slides using Rapiclear solution. Confocal microscopy was performed using a ×20 oil immersion lens, with a step size of 1 μm for confocal imaging (Leica Sp5). The following probes were used: B4-Nr2f2 NM_183261.3; B5-HCN4 NM_001081192.2, B2-cTnnT2 NM_001276345.2 and Hand1-B4 NM_008213.2.

## Lineage tracing

To synchronize oestrous cycles, females were exposed to male-soiled bedding for 3 days prior to mating. On the fourth day, mating was conducted for 3 h (from 8 AM to 11 AM), and the presence of vaginal plugs was checked at 11 AM, designating the embryonic stage as E0. Tamoxifen (T5648, Sigma) was administered via oral gavage at a dose of 0.02 mg/g body weight, dissolved in corn oil, at specific developmental stages. tdTomato+ cells were counted across sections using the Cell Counter plugin in Fiji.

## Embryo harvesting and culture

Embryos were collected at E6.5, and variation in embryonic stages was due to the high degree of variability in embryonic stages within each litter and between different litters, as previously documented (Ivanovitch et al, 2021). Harvesting was performed in DMEM (D5921, Sigma-Aldrich) containing 10% FBS (A5256701, Thermo Fisher), 25 mM HEPES-NaOH (pH 7.2), penicillin, and streptomycin. Dissections were conducted under a stereoscope equipped with a Tokai Hit thermoplate set at 37 °C and completed within 5 min to preserve developmental potential. Embryos were then transferred to a culture medium consisting of 75% freshly prepared rat serum (filtered through a 0.2-μm filter) and 25% DMEM (supplemented with 1 mg/mL D-glucose and pyruvate, without phenol red and L-glutamine, D5921, Sigma-Aldrich), further enriched with Gluta-MAX, 100 U/mL penicillin, 100 μg/mL streptomycin, and 11 mM HEPES. Rat serum was prepared according to established protocols, heat-inactivated at 56 °C for 30 min, and stored at −80 °C until use. The culture medium was equilibrated in 5% O2, 5% CO2, and 90% N2 atmosphere and pre-warmed to 37 °C before embryo transfer.

## Embryo positioning and multiphoton imaging

Custom plastic holders were fabricated with varying hole diameters (0.3–0.5 mm) to secure the ectoplacental cone, following the method described by Nonaka et al (Ichikawa et al, 2014). Embryos were mounted with the anterior side facing up and overlaid with mineral oil (M8410, Sigma-Aldrich) to prevent medium evaporation. Before imaging, embryos were pre-cultured for 2 h in the microscopy setup. Time-lapse imaging was performed using a multiphoton Olympus FVMPE-RS microscope equipped with a 5% CO2 incubator and a heated chamber maintained at 37 °C. A 20x/ 1.00 W dipping objective with a 2-mm working distance was used for imaging. A SpectraPhysics MaiTai DeepSee pulsed laser set at 880 nm was used for single-channel two-photon imaging. Image acquisition settings included 250 mW output power, 7 μs pixel dwell time, and 610 × 610 μm image dimension (1024 × 1024 pixels). The z-step was set to 6 μm.

## Light-sheet microscopy

For all light-sheet acquisitions, we imaged *TnEGFP-CreERT2/+; R26 Tomato Ai14/ Tomato Ai14, cTnnT-2a-GFP + /−* embryos. Tamoxifen (T5648, Sigma) was administered via oral gavage at a dose of 0.02 mg/g body weight dissolved in corn oil at specified embryonic stages, and embryos were harvested at least 12 h post-gavage. Prior to imaging, embryos were pre-cultured for 2 h in the microscopy setup. To maintain positional stability during imaging, the ectoplacental cone was partially embedded in growth factor-reduced, phenol red-free Matrigel (356231, Corning) diluted 1:1 with culture medium within a dedicated open-top FEP sample chamber containing four wells. One embryo was mounted per well, accommodating up to four embryos per experiment. Imaging was conducted using the Viventis LS1 light-sheet microscope with a single-view, dual illumination configuration. Detection was performed using a Nikon 25× NA 1.1 water immersion objective (final magnification: 18.7×), yielding an 800 × 800 μm field of view. Sequential illumination was applied with 488 nm and 561 nm lasers. Image acquisition was performed every 2 min with z-stacks spanning 500 μm. The voxel size was 2 μm (z) × 0.347 μm (x, y),

and the light-sheet thickness was 3.3 μm. Exposure times for GFP and tdTomato channels were set to 50 ms and 50–100 ms, respectively. Embryos were maintained at 37 °C in an 8% $CO_2$ humidified environment throughout the imaging session.

## Cell tracking

The original images were converted into HDF5/XML file formats and registered using BigStitcher (Horl et al, 2019), following previously established methodologies (Dominguez et al, 2023; Dominguez et al, 2025). Processed images were then imported into the MaMuT Fiji plugin for manual cell tracking (Wolff et al, 2018). The MaMuT viewer allowed visualization of tracked cells with adjustable brightness, scale, and rotation settings. Cell identification was initiated at the first time point at which the cell's tdTomato fluorophore starts to be detectable in a movie, with manual tracking performed every five time points, except during mitosis, where tracking was conducted at every time point to ensure accurate measurements of daughter cell separation. The TrackScheme lineage browser was used to visualize reconstructed cell lineage trees, representing cells as nodes interconnected by edges, and cell divisions as split branches. For subsequent analysis, tracked cells' positions, tracks, division times, and GFP intensities were exported in CSV format. We were unable to fully track a large proportion of cells generating the presence of short lineage branches. This limitation hindered our ability to determine their final fates. One key reason for these shorter tracks was the occasional high density of labelling, which, coupled with the spatiotemporal resolution of our imaging setup ($0.347 \times 2$ μm, z-stacks acquired every 2 min), was insufficient to consistently and unambiguously curate some cell tracks. The difficulty in tracking was probably exacerbated by the high dispersion of cells during the earliest stages, which is particularly high in multipotent mother cells. Another contributing factor is related to cells migrating to deeper regions of the heart tube. Over extended timeframes, these cells often relocated towards the more dorsal regions of the forming heart tube, where they became dimmer due to their position along the z-axis. Consequently, many daughter cells did not meet the GFP intensity threshold required to classify them as myocytes and were thus labelled as mesodermal (see Fig. 7C, green arrows- for an example). Additionally, some cells could not be tracked for prolonged periods, especially as they moved dorsally during the transformation of the cardiac crescent into the heart tube. To avoid introducing erroneous lineage assumptions, we opted to stop tracking under such conditions. A limitation of light-sheet imaging is its reduced capacity to capture high-quality images in deeper tissues due to light scattering. Addressing this limitation and improving imaging depth will be critical in future studies.

## Threshold analysis

GFP values were normalised based on the highest GFP value within each movie's tracking data. To discern GFP-positive cells from the background, we implemented the following approach: GFP background intensities were measured at regular intervals throughout each movie and linear interpolation was applied between time points to establish a background value for every time point. The threshold for distinguishing GFP-positive and GFP-negative cells was determined by using a multiple of the mean background. The

threshold was set to the lowest multiplier, ensuring that all endocardial cells in each movie were classified as GFP-negative.

## Temporal alignment of the time-lapse movies

The movies were temporally aligned based on the differentiation timing of LV/AVC and inflow myocytes. Differentiation was defined as the point when the cTnnT-2a-eGFP signal intensity surpassed a predefined threshold. Time point 0 in Movie EV2 was used as the reference (T0). The time in the other movies was adjusted relative to this reference by adding 74 min (Movie EV1), 239 min (Movie EV3), 373 min (Movie EV4), and 626 min (Movie EV5) to synchronize their differentiation timings. All lineage trees were reconstructed using these adjusted time points. Birth date is the time point at which the cell's tdTomato fluorophore starts to be detectable in a movie.

## Lineage analysis

Each reconstructed lineage tree starts with a mother cell. The mother divides, giving rise to the two daughter cells D1 and D2. Then D1 divides giving rise to daughters D11 and D12. D11 divides giving rise to daughters D111 and D112. Lineage trees were generated in R version 4.3.2 using the 'ggtree' version 3.10.0 package with Newick format and imported into R through the 'read.tree' function from the 'ape' package version 5.7.1. The branch lengths of the trees are proportional to the cell cycle length. The trees were coloured using the 'scale_colour_gradient' function in the 'ggtree' package, which imparts a colour gradient based on the normalised GFP intensities, speeds or tortuosity of the cells or paintsubtree function in phytools version 2.03 package in R version 4.3.2 (Yu, 2020; Revell, 2012; Virtanen et al, 2020).

## Analysis of myocardial clone dispersion

To analyse the spread of myocardial clones along the anterior-posterior (AP) and dorsal–ventral (DV) axes, we first empirically established these axes in the heart tube. We then identified all cTnnT-2a-egfp-positive cells for each clone and mapped their coordinates onto the AP and DV axes. The spread of each clone was quantified as the proportion of its range (d) along these axes (D). To reduce bias from short-range movements immediately following cell division, we included only cells that were at least 4 h into their cell cycle in the analysis.

## Quantification of sister cell contact duration

We quantified the proportion of time that sister cells remained in contact following cell division. To determine the contact period, we measured the distances between 57 pairs of cells from Movies EV1 and 2, which were visually confirmed to be in contact. The mean contact distance was 13 μm. Distances between sister cells were then measured at each time point, and the duration for which the distance remained below 13 μm was recorded. The cell contact duration was expressed as a proportion of this contact period relative to the total cell cycle length. Only sister cells from generations 1 (D1, D2) and 2 (D11, D12, D21, D22) were included in the analysis. The study was restricted to the early migration period, within the first 16 h of the aligned movies, corresponding to the timeframe when the

first myocardial cell becomes cTnnT-2a-GFP + . Sister cells with a cell cycle duration of less than 4 h were excluded.

## Homotypic and heterotypic distances calculation

Trajectories were reconstructed in 3D using the matplotlib version 3.7.2 and plotly version 5.18.0 packages in Python version 3.11.53. Euclidean distances were computed to quantify the spatial relationships among cells belonging to the same family tree, categorizing them as 'homotypic distances' for cells of the same fate and 'heterotypic distances' for cells of different fates. Progenitors' coordinates were sampled at two distinct time points: at the end of migration and during the stages when the heart tube forms. We only considered branches lasting at least 4 h into the cell cycle to allow sufficient cell migration. The migration period for early proximal mesoderm cells, which include the LV/AVC, ExEm, endocardial, and pericardial cells, was defined as the time from the beginning of the movie until the completion of cell migration, marked by the mean differentiation time of the LV/AVC region. The mean differentiation times for LV/AVC cells were 20.7 h for Movie EV1, 20.8 h for Movie EV2, 17.3 h for Movie EV3, 16.3 h for Movie EV4, and 11.5 h for Movie EV5. The subsequent heart tube morphogenesis period extended from the end of the migration phase until the end of the movie, corresponding to the stages of heart tube formation. For atrial cells, the migration period concluded at the mean atrial differentiation time. The mean differentiation times for inflow cells were 29.1 h for Movie EV1, 26.8 h for Movie EV3, 23.3 h for Movie EV4, and 19.8 h for Movie EV5. No inflow cells were tracked in Movie EV2.

To analyse the evolution of homotypic and heterotypic distances over time, we focused on daughters' coordinates at three distinct time points: T0, T1, and T2. T0 was 20 min after the mother's initial cell division, T1 was the last time point before the daughters' subsequent cell division, and T2 was the final time point at which all granddaughter cells existed before the end of migration (as described above). If no third generation was present, we only considered branches lasting at least 4 h into the cell cycle to allow sufficient cell migration.

## Speed calculation

To compare the motility of different cell types, we calculated their speed using the CelltrackR package version 1.1.0 in R version 4.3.2 (Wortel et al, 2021). Speed was determined by dividing the total track length of each cell by its duration, resulting in a mean speed expressed in μm/minute. To evaluate changes in speed over time, the aligned movies were divided into 5-hour intervals, and the mean speed of cells in each interval was calculated. Cells with a cell cycle length of less than 40 min within any interval were excluded from the analysis.

## Cell tortuosity calculation during heart tube formation

Tortuosity quantifies the extent to which a cell's path deviates from a straight line. It was calculated using the ratio of the shortest straight-line distance between two points to the total path distance covered by the cell:

$$Tortuosity = \frac{Straight\ line\ distance}{Total\ distance}$$

A tortuosity value closer to one indicates a straighter migratory path, while values approaching zero signify a more curved trajectory. Local tortuosity was calculated at 50-min intervals and then averaged over the cell cycle duration computed for each cell. Cells with a cell cycle length of less than 3 h were excluded from the analysis.

## Dynamic time warping

Migratory trajectories were analysed by comparing the paths of sister cells from the point immediately after their mother cell division to the final time point before the first sister cell divided again. Trajectory similarity was assessed using dynamic time warping (DTW) distance, calculated using the dtw-python package (version 1.3.0). DTW was chosen over Euclidean distance because it assigns short distances to spatially similar but temporally asynchronous trajectories. The analysis used the 'symmetricP1' step pattern, a slope-constrained step pattern (Chiba, 1978; Shokoohi-Yekta et al, 2017). DTW values were calculated separately for sister cells with the same fate (uni-fated) and different fates (bipotent) across the first and second generations (D1, D2; D11, D12, D21, D22). Distinct analyses were performed for progenitors contributing exclusively to mesoderm (Uni_meso and Bi_meso) and progenitors contributing to at least one extraembryonic mesoderm (ExEm) daughter (Uni_ExEm and Bi_ExEm). To measure trajectory similarity during the migration period, DTW values were calculated up until the end of migration. To determine whether DTW values differed significantly between uni-fated and bipotent sister cells over time, mean DTW values for the first 40 steps were plotted, and statistical comparisons were made using the Mann–Whitney test.

## Permutation test

We generated 100,000 permutations by pooling all log DTW distances and randomly assigning them to either uni-fated or bipotent conditions. For each permutation, we calculated the difference in mean log DTW values between sister cells with shared fates and those with distinct fates. This iterative process created a null distribution, allowing us to test whether uni-fated progenitors produce sisters with more similar trajectories than bipotent progenitors (Appendix Fig. S8A–D).

# Data availability

The codes and source numerical data are available at: https://github.com/sabukar22/Supplementary-Data. The light-sheet microscopy datasets and cell tracking annotations are collected at: https://www.ebi.ac.uk/biostudies/bioimages/studies/S-BIAD1682. AccessionS-BIAD1682. https://doi.org/10.6019/S-BIAD1682. The cTnnT-2a-eGFP (*Tnnt2em1(eGFP)Jcs*) line is deposited at Jax (JAX Stock No. 040309).

The source data of this paper are collected in the following database record: biostudies:S-SCDT-10_1038-S44318-025-00441-0.

# Peer review information

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

## Acknowledgements

The authors thank Dr Martin Dominguez for his initial help with BigStitcher and Dr Rosie Marshall and Prof. Andrew Copp for their help with rat serum preparation. We also thank the Light Microscopy facilities and Biological Research Facilities at the Francis Crick Institute and UCL for their help with imaging and transgenic colonies. Research infrastructure within the Institute of Child Health, UCL is supported by the NIHR Great Ormond Street Hospital Biomedical Research Centre. The views expressed are those of the authors and not necessarily those of the NHS, the NIHR or the Department of Health. This work was funded by BHF (FS/IBSRF/21/25085) to KI. SA received a 4-year BHF PhD Studentships (FS/4yPhD/F/22/34181). This work was also supported by the Francis Crick Institute, which receives its core funding from Cancer Research UK, the UK Medical Research Council and Wellcome Trust (all under FC001051) to JB. PAE and JAD were supported by the Radiation Research Unit

at the Cancer Research UK City of London Centre Award (C7893/A28990); British Heart Foundation (BHF) FS/IBSRF/21/25085; British Heart Foundation (BHF) FS/4yPhD/F/22/34181; Cancer Research UK (CRUK) C7893/A28990; Francis Crick Institute (FCI) FC001051.

## Author contributions

**Shayma Abukar**: Conceptualization; Software; Formal analysis; Investigation; Visualization; Methodology; Writing—review and editing. **Peter A Embacher**: Methodology; Writing—review and editing. **Alessandro Ciccarelli**: Methodology. **Sunita Varsani-Brown**: Investigation. **Isabel G W North**: Investigation. **Jamie A Dean**: Supervision; Methodology; Writing—review and editing. **James Briscoe**: Resources; Writing—review and editing. **Kenzo Ivanovitch**: Conceptualization; Resources; Data curation; Formal analysis; Supervision; Funding acquisition; Validation; Investigation; Visualization; Methodology; Writing—original draft; Project administration; Writing—review and editing.

Source data underlying figure panels in this paper may have individual authorship assigned. Where available, figure panel/source data authorship is listed in the following database record: biostudies:S-SCDT-10_1038-S44318-025-00441-0.

## Disclosure and competing interests statement

The authors declare no competing interests.

# Expanded View Figures

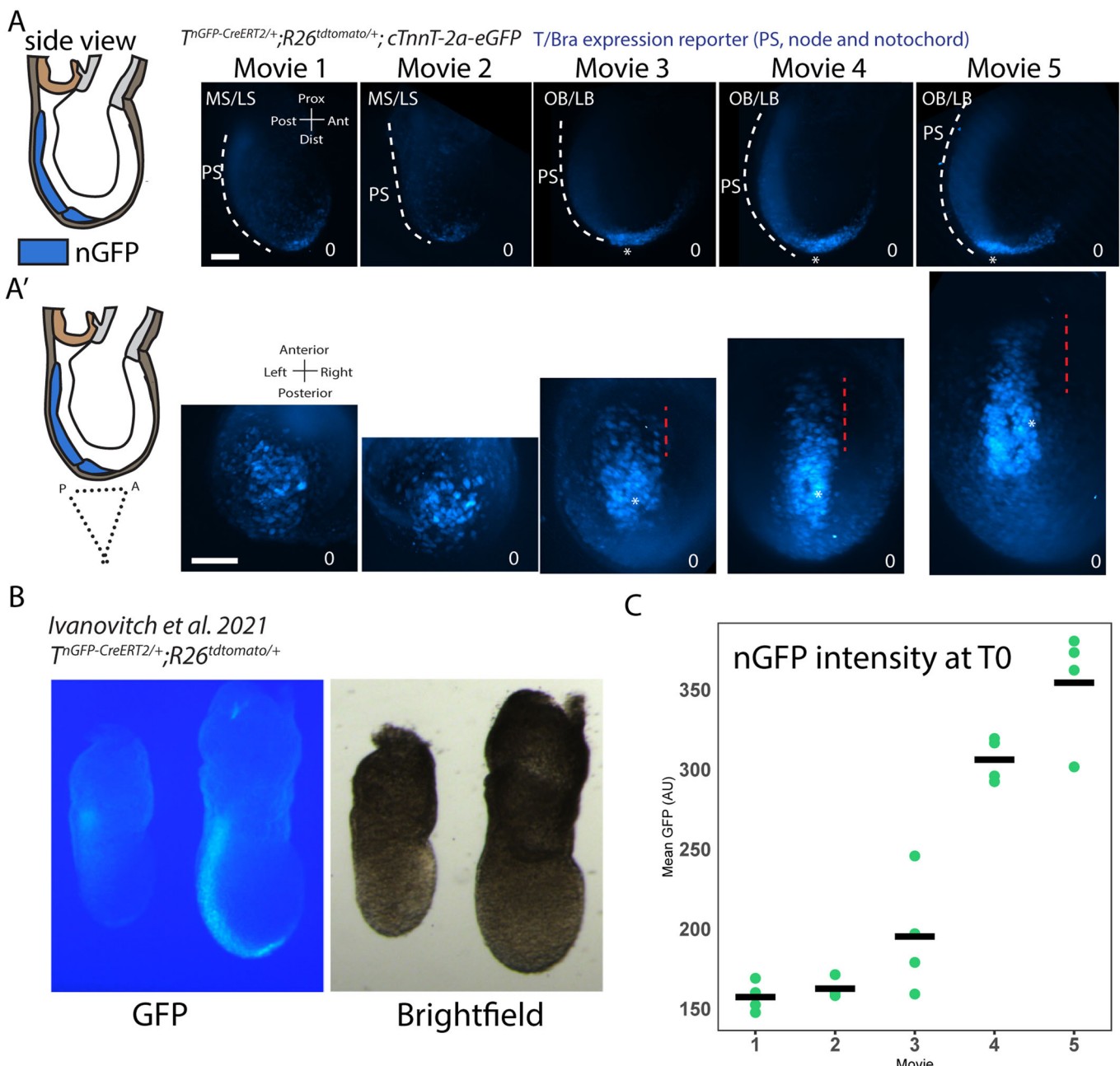

**Figure EV1.  Stage variation of the embryos imaged by light-sheet microscopy.**

(**A–A'**) At T0, the embryos in Movies EV1 and 2 are at earlier stages compared to those in Movies 3 to 5. Embryos in Movies EV3–5 at T0 are characterized by the presence of a node structure (highlighted by an asterisk in (**A**)) and the beginning of the notochord plate (highlighted by the red dotted line in (**A'**)). In contrast, the node and notochord are absent in embryos from Movies EV1 and 2 at T0. (**B**) The earlier MS-LS embryos have a lower nGFP primitive streak signal compared to the later-stage OB-LB embryos. Images adapted from Ivanovitch et al, 2021. (**C**) Quantification of nGFP primitive streak signal at T0 for embryos from Movies EV1–5. nGFP was quantified in four consecutive optical sections (see also Appendix Fig. S6). Each data point represents the mean GFP intensity for each z-slices. PS: primitive streak. Scale bar: 200 μm in (**A**) and 100 μm in (**A'**). MS-LS: mid and late streak stages. OB/LB: no bud to late bud stages. nGFP: nuclear GFP. Source data are available online for this figure.

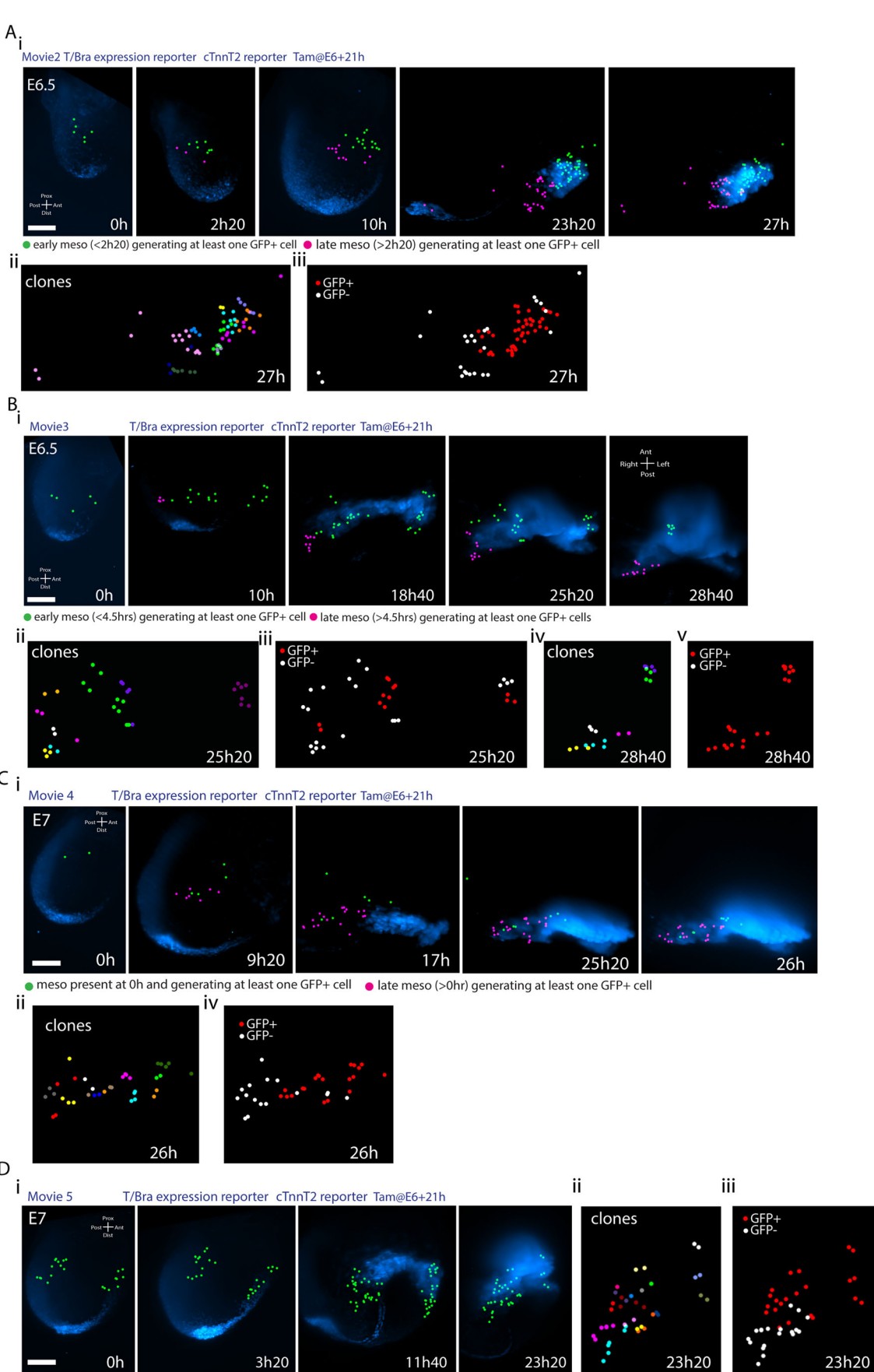

◀ **Figure EV2. Independent progenitors contribute to the LV/AVC and Atria.**

(A–D) Only progenitors contributing at least one cTnnT-2a-GFP+ cell are displayed. (i) Fate map showing early (green) and late (magenta) mesoderm contributions to heart tube regions (early/late classification is arbitrary). (ii) Each colour represents a unique clone. (iii) cTnnT-2a-GFP+ cells in red. Scale bar: 100 μm.

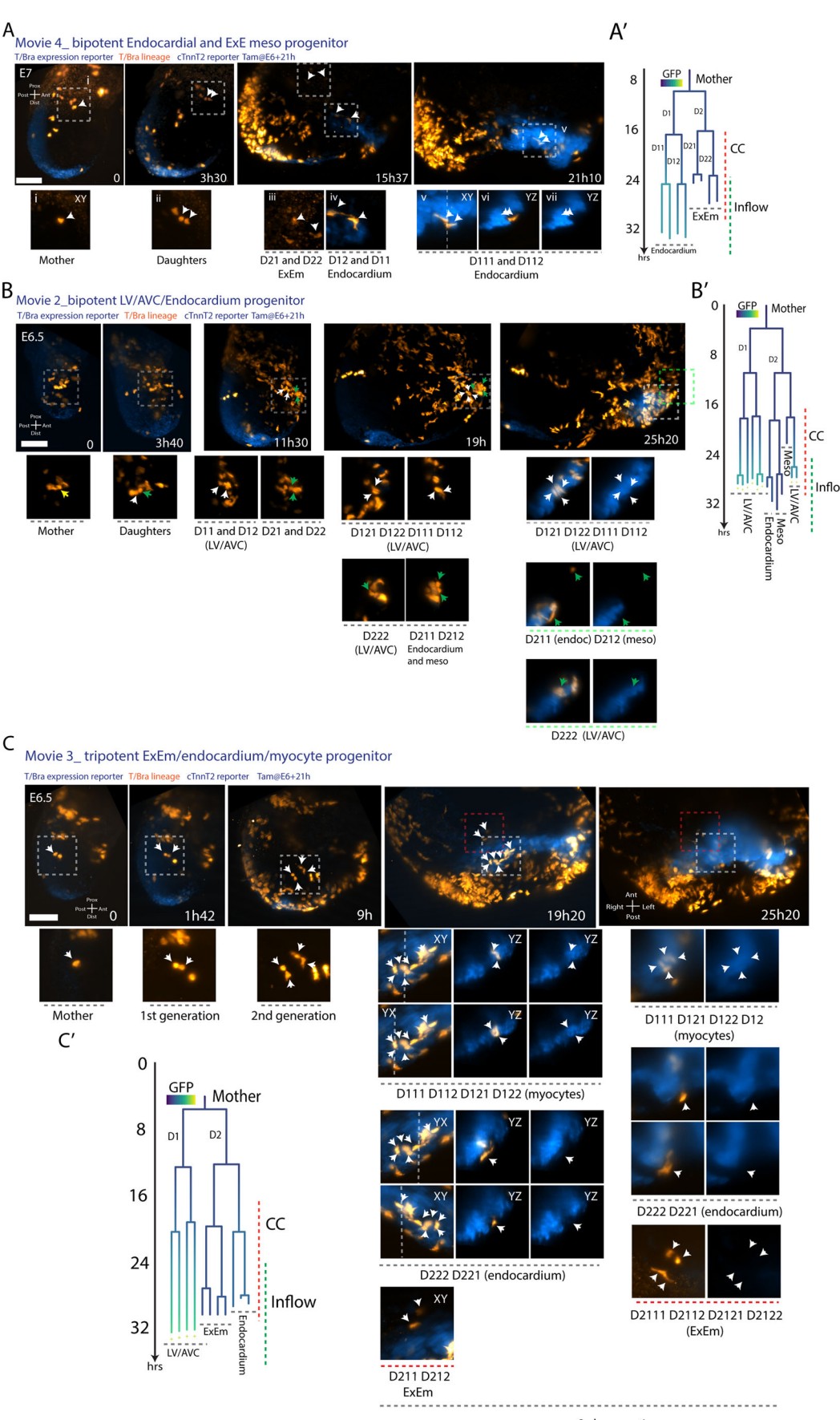

A  Movie 4_ bipotent Endocardial and ExE meso progenitor
T/Bra expression reporter  T/Bra lineage  cTnnT2 reporter  Tam@E6+21h

Mother | Daughters | D21 and D22 ExEm | D12 and D11 Endocardium | D111 and D112 Endocardium

B  Movie 2_bipotent LV/AVC/Endocardium progenitor
T/Bra expression reporter  T/Bra lineage  cTnnT2 reporter  Tam@E6+21h

Mother | Daughters | D11 and D12 D21 and D22 (LV/AVC) | D121 D122 D111 D112 (LV/AVC) | D121 D122 D111 D112 (LV/AVC)

D222 (LV/AVC) | D211 D212 Endocardium and meso

D211 (endoc) D212 (meso)

D222 (LV/AVC)

C  Movie 3_ tripotent ExEm/endocardium/myocyte progenitor
T/Bra expression reporter  T/Bra lineage  cTnnT2 reporter  Tam@E6+21h

Mother | 1st generation | 2nd generation

D111 D112 D121 D122 (myocytes)

D111 D121 D122 D12 (myocytes)

D222 D221 (endocardium)

D222 D221 (endocardium)

D211 D212 ExEm

D2111 D2112 D2121 D2122 (ExEm)

3rd generation

◀ **Figure EV3. Examples of bipotent and tripotent mesodermal progenitors.**

(**A–C**) Time-lapse images of $T^{nGPF-CreERT2/+}$;$R26R^{tdTomato/+}$; *cTnnT-2a-eGFP* embryos showing bipotent (**A**, **B**) and tripotent (**C**) progenitors. Corresponding lineage trees, coloured by normalized GFP intensity, are shown in (**A'–C'**), with arrows indicating the cells in (**A–C**). Scale bar: 100 μm. Source data are available online for this figure.

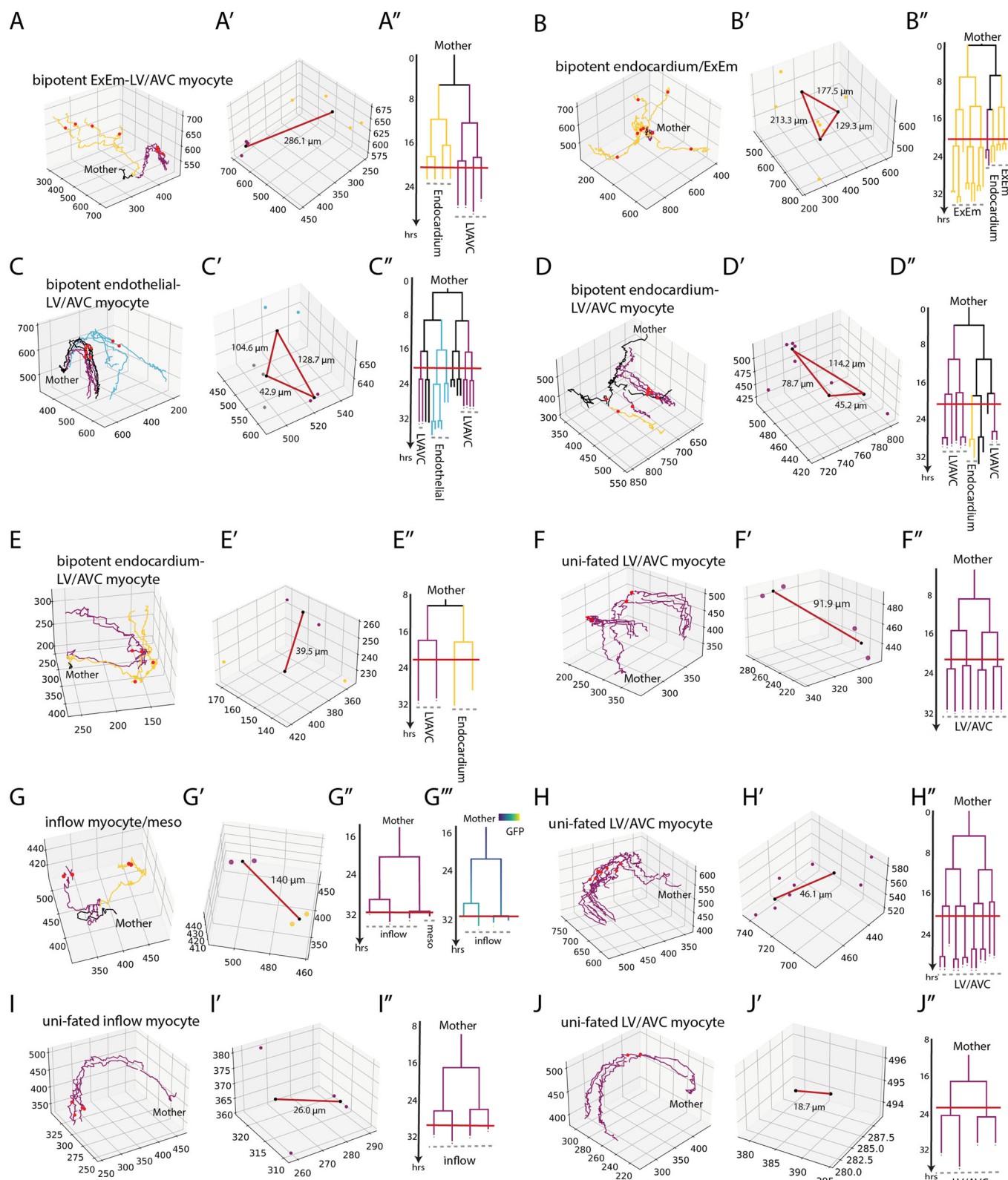

**Figure EV4. Mesodermal migration paths.**

Examples of trajectories (A–J) and corresponding midpoint distances (A'–J') and lineage trees (A"–J"). Red lines mark sampling points for midpoint analysis. LV/AVC left ventricle and atrioventricular canal, ExEm Extraembryonic mesoderm. Source data are available online for this figure.

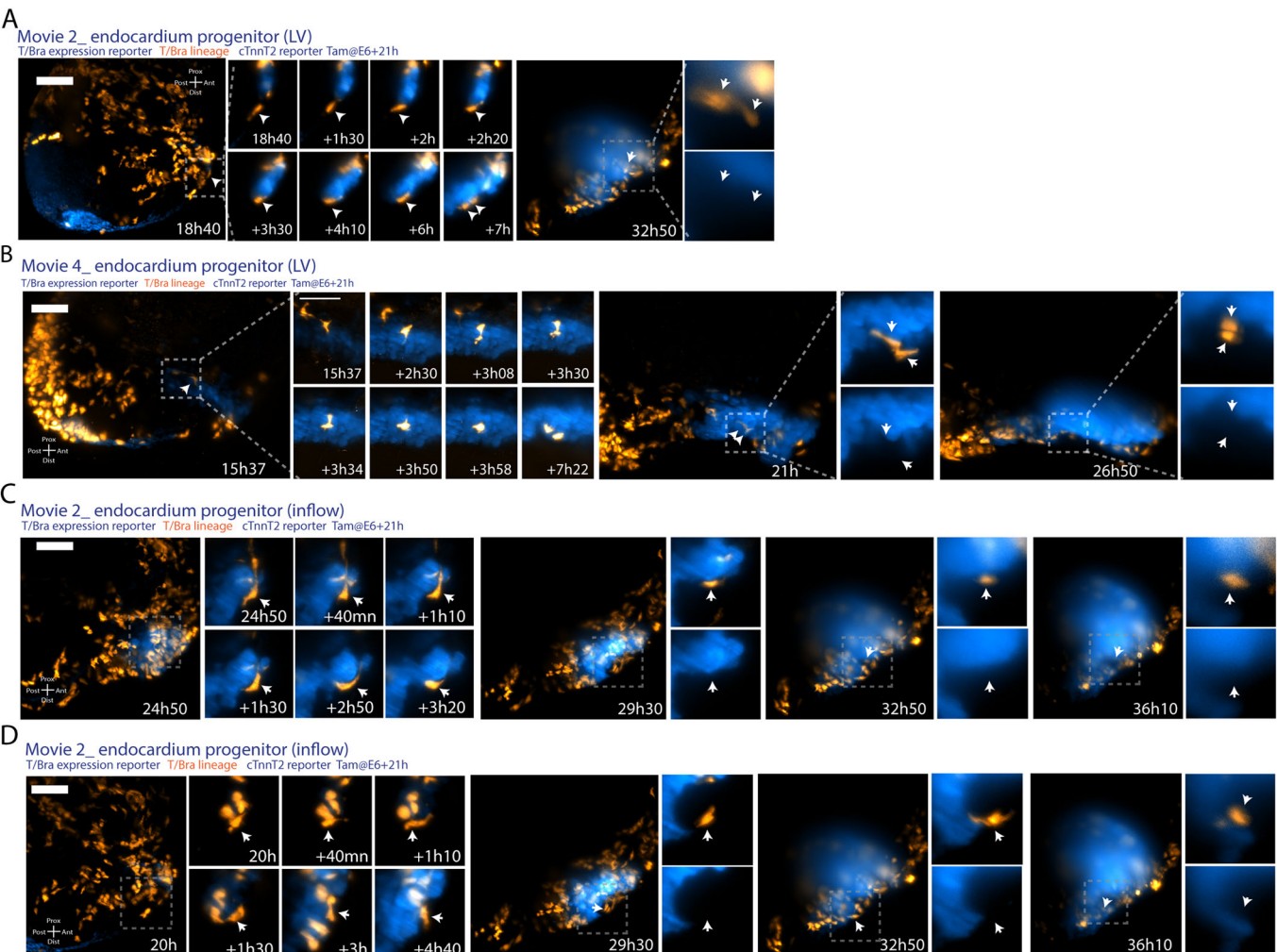

**Figure EV5. Endocardial cell behaviour.**

(A–D) Image sequences from time-lapse movies of $T^{nGPF\text{-}CreERT2/+}$;$R26R^{tdTomato/+}$; $cTnnT\text{-}2a\text{-}eGFP$ embryos showing endocardial progenitors in the LV (**A, B**) and in the inflows (**C, D**). White arrows in (**A–D**) show endocardial cells. LV left ventricle. Scale bar: 100 µm.

