## [Peer Review File · The EMBO Journal]

Early coordination of cell migration and cardiac fate determination during mammalian gastrulation.

Shayma Abukar, Peter Embacher, Alessandro Ciccarelli, Sunita Varsani-Brown, Isabel North, Jamie Dean, James Briscoe, and Kenzo Ivanovitch

Corresponding author(s): Kenzo Ivanovitch (k.ivanovitch@ucl.ac.uk)

Review Timeline:

Transfer from Review Commons:	26th Jan 25
Editorial Decision:	29th Jan 25
Revision Received:	25th Feb 25
Editorial Decision:	25th Mar 25
Revision Received:	29th Mar 25
Accepted:	8th Apr 25

Editor: Ieva Gailite

Transaction Report:

Review #1

1. Evidence, reproducibility and clarity:

Evidence, reproducibility and clarity (Required)

The manuscript describes the tracking of individual mesoderm cells through live imaging. Through a combination of reporters including a novel cardiomyocyte reporter and a combined nuclear GFP-inducible Cre reporter under the dependence of the Brachyury promoter, the authors label mesoderm cells at different stages of gastrulation then perform long term (>30h) live imaging of late gastrulation embryo up to the cardiac crescent and heart tube stages. They use elaborate analysis tools as well as manual tracking to reconstruct cells' trajectory, lineage trees, and various behavioral traits.

The study is well designed. Experiments are technically challenging, well executed, and carefully analysed.

Methods are clear and complete so that experiments should be faithfully reproduced provided availability of an appropriate microscope.

The description of the results of the live imaging experiments is not easy to read and understand, but I believe this is inherent to the complexity of the results themselves and due to the high diversity of behaviors observed. Similarly the figures are extremely dense and some graphs would benefit from a more didactic legend.

I realize the difficulty of being more concise due to the large amount of information and its diversity. If possible, I would suggest integrating tables within the results section that may help shorten the text, and may be easier to grasp.

The interpretation of the results is fair and in line with previous studies, which are adequately cited.

A discussion on the reasons why a large proportion of cells could not be determined as uni or multipotent might be useful. Instinctively I would imagine that a majority of those are multipotent and therefore harder to track, so if the authors do not agree with this interpretation it may be useful to detail technical reasons why those cells cannot be fully interpreted.

2. Significance:

Significance (Required)

Strengths: novel transgenic tools, powerful imaging technique, thorough quantified analysis.

Limitations: the development of embryos after E7.75-E8 is never completely normal *ex vivo*, particularly when there is no rotation. This is visible in the pictures of the embryos post culture (ballooned yolk sac, unattached allantois). It is probably not a limitation regarding cardiac development but may influence other mesoderm lineages notably ExE.

Advance: It is a unique study due to the labelling strategy, the length of imaging, and thereby the faithful tracking of cell lineages across several rounds of division. The information provided corroborates what previous hypothesis in the field based on less direct assessment, and is here very strong and unbiased.

The research is of great interest for developmental biologists (including but not limited to the heart field), cell biologists (notably those working on stem cells and organoids as it provides a ground truth), microscopy and image analysis experts.

3. How much time do you estimate the authors will need to complete the suggested revisions:

Estimated time to Complete Revisions (Required)

(Decision Recommendation)

Less than 1 month

Yes

Review #2

1. Evidence, reproducibility and clarity:

Evidence, reproducibility and clarity (Required)

The authors perform an elegant "tour de force" lineage relationships during mouse heart development. They perform long-term live imaging and single-cell tracking in mouse embryos from early gastrulation to stages of heart tube formation. They then track the progeny of individual cells and reconstruct the lineage tree of tracked cells. They analyze how their migratory paths of cells correlate with cell fate in the heart. Altogether, the manuscript presents a highly detailed live-imaging lineage tracing study of a subset of cells in the cardiac crescent in mouse. This presents a nice contribution to the literature, but would be improved by the suggestions below.

****Major comments:****

1. Can the authors be sure they can track all of the derivatives of labeled cells? They are claiming to be able to follow complete lineages, but I worry if they may lose progeny in their tracking or incorrectly conclude that cells are lineally related. wonder how you could show how accurate it really is. Perhaps if the authors could include a movie where they trace what they claim as an entire lineage of a single cell and show this with the mother and daughter cells labelled throughout the movie, that would at least provide an example for readers to make their own decisions about how reliable the lineage tracing is. Would it be feasible to include an interactive movie where the reader can move the embryo around in 3D at each time point?
2. The authors describe the lineage labeled cells as unipotent, bipotent, etc. But they cannot really say anything about developmental potential as they are only looking at normal fate which is less than their potential. Without manipulation of the cells through transplantation etc., the use of the term 'potential' or 'potent' is not appropriate except when they find cells that are multipotent. Rather than calling cells unipotent, I would suggest using the phrase 'assume a single fate'.
3. Lines 112-115, the authors state that variability in embryonic stages likely explains differences in labelling. Are there any morphological characteristics across the embryos that support this variability in stages? For example, any characteristics that suggest that the n=3 embryos are slightly older, and the n=7 embryos are slightly younger (line 111)?
4. Paragraph beginning on line 116: Please clarify how cells were counted, from the wholemount/across sections?
5. Line 165: Authors state that in the absence of tamoxifen, tdTomato-positive cells were identified in one embryo. Please state here the total number of embryos out of which this one embryo was counted.
6. Line 190: 'Figure 2-Supplementary Figure 3A-F' doesn't exist. Do they mean Fig.3

supplementary 3A-F?

7. Figure 1F-G: For cross sections in 'G' please show the level they were taken from in 'F'.

8. Figure 4I: There is a large disparity in cell dispersion across movies. Please comment on why this could be. Is there a difference in stage/morphology etc..

9. Figure 4K-L: The arrowhead color is too similar to the cell fluorescence color, making the visualization a little confusing. Changing the color of the arrowheads may be helpful. This is also true for some of the other figures (red arrowheads).

2. Significance:

Significance (Required)

This is a well-done study that will be useful to developmental biologists as well as cardiologists. The experiments seem very well done and beautifully executed. With the proposed modifications, it will make a very nice contribution to the literature.

3. How much time do you estimate the authors will need to complete the suggested revisions:

Estimated time to Complete Revisions (Required)

(Decision Recommendation)

Between 1 and 3 months

4. Review Commons values the work of reviewers and encourages them to get credit for their work. Select 'Yes' below to register your reviewing activity at Web of Science Reviewer Recognition Service (formerly Publons); note that the content of your review will not be visible on Web of Science.

No

Review #3

1. Evidence, reproducibility and clarity:

Evidence, reproducibility and clarity (Required)

In their manuscript, Abukar et al. investigate the origins and migratory behaviors of cardiac progenitor cells, in mice, from gastrulation to early heart tube formation. They use

sophisticated live imaging to track individual mesodermal cells, reconstructing their lineage and fate over several generations. The findings reveal distinct unipotent progenitors that contribute exclusively to specific cardiac regions, such as the left ventricle/atrioventricular canal (LV/AVC) or atrial cardiomyocytes. LV/AVC progenitors differentiate early, forming the cardiac crescent, while atrial progenitors differentiate later, contributing to the venous poles of the heart tube. Additionally, the study identifies multipotent mesodermal progenitors contributing to various mesodermal cell types, including the endocardium, pericardium and extraembryonic tissues.

****Major comments:****

1. Important conclusions of the manuscript rely on the expression of a reporter line (cTnnt2-2a-eGFP) as well as on the position of tdTomato⁺ cells in relation to the reporter. We feel that markers of non-myocardial lineages should have been used to better characterize these populations. We acknowledge the technical challenge of live imaging, which may not allow labeling of all lineages. We believe that a better description of the final stages of investigation with markers of endocardium, pericardium, extra-embryonic mesoderm together with the eGFP of the reporter will strengthen the conclusions drawn on the multipotency of the progenitors. If not addressed, some claims may appear more speculative and would benefit from being toned down.

2. Similarly, since all the results of the manuscript derive from five movies of five independent embryos, it would be important to provide a more detailed description (for example, in a table) of the experimental setup. This could include the timing of tamoxifen induction (+7h or +21h?), the stage of dissection (based on anatomical landmarks rather than dissection stage - see atlas of gastrulation), the duration of the movies, and the stage at the final time point. Providing this information would greatly enhance the ability to robustly compare each movie and ensure reproducibility. Of note, the methods section could benefit from additional clarity. For example, in line 594, the embryo from Movie1 is described as being dissected in the morning, while the next sentence states it was dissected in the afternoon, similar to the embryo in Movie5. To avoid confusion and ensure greater rigor, describing the developmental stage of the embryos rather than the time of dissection would be more precise and biologically meaningful.

3. This manuscript focuses primarily on LV/AVC progenitors and likely a subpopulation of atrial cardiomyocytes, leaving other cardiac progenitor populations unaddressed. While it is understandable that the study focuses on specific populations, the authors should further discuss the limitations of their approach and explain why not all cardiac progenitors were targeted. A discussion of how these limitations might impact the broader interpretation of their findings would also be valuable.

4. Since a recent preprint (Sendra et al.), already cited in the manuscript, used complementary approaches to investigate endothelial/endocardial cell fate during gastrulation, we feel that a more in-depth discussion is warranted. In particular, how the results presented here align with the early segregation between endocardial and myocardial lineages observed by Sendra et al. could be clarified. Additionally, it is unclear how these findings correlate with Foxa2 lineage tracing. Addressing these points could further strengthen the contextualization and impact of the manuscript.

****Minor comments:****

1. For all figures, annotations, axes and/or schematics would greatly help readers outside the field to locate the regions of interest within the embryo.
2. Interesting questions that could be easily addressed and added in the manuscript: are mother cells T-nGFP positives? If so, do they have different levels of GFP expression? From the different movies, is there a hot spot of cell division? What is the frequency of progenitors that adopt a sustained interaction with their sister cells?

2. Significance:

Significance (Required)

The manuscript presents a technically original study, offering one of the first prospective clonal analyses of cardiac progenitors during mouse gastrulation. While previous studies have addressed the fate of cardiac progenitors using retrospective clonal analysis or lineage tracing (e.g., Meilhac et al., 2004; Devine et al., 2014; Lescroart et al., 2014; Bardot et al., 2017; Ivanovitch et al., 2021; Tyser et al., 2021; Zhang et al., 2021), this work provides new insights into the temporal and spatial dynamics of cardiac progenitor migration and fate allocation. Notably, the study's investigation of the pericardium—a rarely studied cardiac mesodermal fate—adds significant novelty.

However, a limitation of the study is its focus on a relatively small region of the heart, primarily the left ventricle, atrioventricular canal, and atrium, which may not fully represent the broader diversity of cardiac progenitor behaviors across other regions of the developing heart. Additionally, the lack of markers for non-myocardial cell lineages leaves open questions regarding the full spectrum of progenitor fates. These aspects could be addressed in future studies to provide a more comprehensive understanding of cardiac development.

A complementary preprint by the Torres group (Sendra et al., 2024) combines retrospective

and prospective clonal analyses and highlights the multipotency of early mesodermal progenitors, particularly those contributing to non-cardiac fates. While both studies reveal the plasticity of early mesoderm, this manuscript by Abukar et al. focuses specifically on cardiac progenitors, offering unique insights into their behaviors and fate decisions.

The study is poised to have a broad impact on the fields of cardiac development and early mouse development. The tools and concepts developed here could also find applications in broader developmental biology studies. This review is written with expertise in cardiac development. I do not have sufficient expertise to evaluate computational modeling within the manuscript.

3. How much time do you estimate the authors will need to complete the suggested revisions:

Estimated time to Complete Revisions (Required)

(Decision Recommendation)

Between 1 and 3 months

Yes

Revision Plan

Manuscript number: RC-2024-02815

Corresponding author(s): Kenzo, Ivanovitch

[The “revision plan” should delineate the revisions that authors intend to carry out in response to the points raised by the referees. It also provides the authors with the opportunity to explain their view of the paper and of the referee reports.]

The document is important for the editors of affiliate journals when they make a first decision on the transferred manuscript. It will also be useful to readers of the reprint and help them to obtain a balanced view of the paper.

*If you wish to submit a full revision, please use our "Full Revision" template. **It is important to use the appropriate template to clearly inform the editors of your intentions.**]*

1. General Statements [optional]

This section is optional. Insert here any general statements you wish to make about the goal of the study or about the reviews.

We would like to thank you and the reviewers for the constructive comments. The reviewers raised several concerns, which we plan to address point by point.

2. Description of the planned revisions

Insert here a point-by-point reply that explains what revisions, additional experimentations and analyses are planned to address the points raised by the referees.

Reviewer #1

Evidence, reproducibility and clarity

The manuscript describes the tracking of individual mesoderm cells through live imaging. Through a combination of reporters including a novel cardiomyocyte reporter and a combined nuclear GFP-inducible Cre reporter under the dependence of the Brachyury promoter, the authors label mesoderm cells at different stages of gastrulation then perform long term (>30h) live imaging of late gastrulation embryo up to the cardiac crescent and heart tube stages. They use elaborate analysis tools as well as manual tracking to reconstruct cells' trajectory, lineage trees, and various behavioral traits.

The study is well designed. Experiments are technically challenging, well executed, and carefully analysed.

Revision Plan

Methods are clear and complete so that experiments should be faithfully reproduced provided availability of an appropriate microscope.

The description of the results of the live imaging experiments is not easy to read and understand, but I believe this is inherent to the complexity of the results themselves and due to the high diversity of behaviors observed. Similarly the figures are extremely dense and some graphs would benefit from a more didactic legend.

I realize the difficulty of being more concise due to the large amount of information and its diversity. If possible, I would suggest integrating tables within the results section that may help shorten the text, and may be easier to grasp.

We will add tables describing the numbers of uni-fated and multi-potent mothers, cell speeds, and dispersion. We will also split the figures to reduce the amount of information in each figure; and improve the legends by providing more detailed explanations.

The interpretation of the results is fair and in line with previous studies, which are adequately cited.

A discussion on the reasons why a large proportion of cells could not be determined as uni or multipotent might be useful. Instinctively I would imagine that a majority of those are multipotent and therefore harder to track, so if the authors do not agree with this interpretation it may be useful to detail technical reasons why those cells cannot be fully interpreted.

We have discussed further reasons why a large proportion of cells could not be classified as uni-fated or multipotent. Indeed, while our analysis revealed a predominance of uni-fated progenitors (n=98, generating 728 descendants) over bifated/trifated progenitors (n=18, generating 302 descendants), a significant number of mother cells (n=111) produced progeny whose fates could not be determined. This is due to multiple factors, as explained below.

First, we were unable to fully track a large proportion of cells that generate short tracks. This limitation hindered our ability to determine their final fates. One key reason for these shorter tracks was the occasional high density of labeling, which, coupled with the spatiotemporal resolution of our imaging setup (0.347 x 2 μ m z-stacks acquired every 2 minutes), was insufficient to consistently and unambiguously curate some cell tracks. We agree with the reviewer that the difficulty in tracking was probably exacerbated by the high dispersion of cells during the earliest stages, which is particularly high for multipotent mother cells. To avoid introducing erroneous lineage assumptions, we opted to stop tracking under such conditions.

Another contributing factor is related to cells migrating to deeper regions of the heart tube. Over extended timeframes, these cells often relocated towards the more dorsal regions of the forming heart tube, where they became dimmer due to their position along the z-axis. Consequently, many daughter cells did not meet the GFP intensity threshold required to classify them as

Revision Plan

myocytes and were thus labeled as mesodermal (line 194 and see Fig. 7C for an example). Additionally, some cells could not be tracked for prolonged periods, especially as they moved dorsally during the transformation of the cardiac crescent into the heart tube. A limitation of light-sheet imaging is its reduced capacity to capture high-quality images in deeper tissues due to light scattering. Addressing this limitation and improving imaging depth will be critical in future studies.

We also acknowledge the graded expression pattern of cTnnT2-GFP in the forming heart tube, with early and higher levels in LV/AVC myocytes and later, lower levels in inflow myocytes. To maintain consistency, we refrained from using different thresholds to account for these regional intensity differences. While this choice could have led to false negatives (e.g., inflow cells not meeting the GFP threshold), we believe this approach minimises the risk of false positives. Any daughter cells failing to meet the threshold were conservatively classified as mesodermal (meso GFP-), even though they may have been myocyte progenitors.

Additionally, some cells contributing to the inflow/atria regions may not have passed the GFP threshold during the imaging period but could have done so at later developmental stages. These cells were also classified as mesodermal, as their myocyte progenitor status could not be determined. This conservative approach prioritises accuracy over overestimation. We have included all these explanations in the main text and Materials and Methods.

Significance

Strengths: novel transgenic tools, powerful imaging technique, thorough quantified analysis.

Limitations: the development of embryos after E7.75-E8 is never completely normal *ex vivo*, particularly when there is no rotation. This is visible in the pictures of the embryos post culture (ballooned yolk sac, unattached allantois). It is probably not a limitation regarding cardiac development but may influence other mesoderm lineages notably ExE.

Advance: It is a unique study due to the labelling strategy, the length of imaging, and thereby the faithful tracking of cell lineages across several rounds of division. The information provided corroborates what previous hypothesis in the field based on less direct assessment, and is here very strong and unbiased.

The research is of great interest for developmental biologists (including but not limited to the heart field), cell biologists (notably those working on stem cells and organoids as it provides a ground truth), microscopy and image analysis experts.

Reviewer #2

Evidence, reproducibility and clarity

The authors perform an elegant "tour de force" lineage relationships during mouse heart development. They perform long-term live imaging and single-cell tracking in mouse embryos from early gastrulation to stages of heart tube formation. They then track the progeny of individual cells and reconstruct the lineage tree of tracked cells. They analyze how their migratory paths of cells correlate with cell fate in the heart. Altogether, the manuscript presents a highly detailed live-imaging lineage tracing study of a subset of cells in the cardiac crescent in

Revision Plan

mouse. This presents a nice contribution to the literature, but would be improved by the suggestions below.

Major comments:

1. Can the authors be sure they can track all of the derivatives of labeled cells? They are claiming to be able to follow complete lineages, but I worry if they may lose progeny in their tracking or incorrectly conclude that cells are lineally related. wonder how you could show how accurate it really is. Perhaps if the authors could include a movie where they trace what they claim as an entire lineage of a single cell and show this with the mother and daughter cells labelled throughout the movie, that would at least provide an example for readers to make their own decisions about how reliable the lineage tracing is. Would it be feasible to include an interactive movie where the reader can move the embryo around in 3D at each time point?

We have not tracked all the derivatives of labeled cells, as explained in our response to Reviewer 1. A number of mother cells (n=111) produced progeny whose fates could not be determined. Each cell track (up to 1,000 time points) required manual curation and verification, as even a single linkage error would compromise conclusions. When a track could not be unambiguously determined, we stopped tracking those cells. We have acknowledged this limitation in the manuscript.

We also agree with the reviewer that it is important to show the tracks, and we will therefore include supplementary movies displaying all the cells tracks. Furthermore, we are submitting all our datasets to the Image Data Resource (IDR) (<https://idr.openmicroscopy.org/>). Our datasets have been accepted, and the IDR team is currently assessing our track data, cell annotations, and metadata. This will enable users to download the data and fully assess them interactively in 3D using MaMuT or Mastodon (<https://mastodon.readthedocs.io/en/latest/index.html>) for cell tracking, as well as to generate their own tracking data. The availability of our data through this resource will significantly enhance its value to the community.

2. The authors describe the lineage labeled cells as unipotent, bipotent, etc. But they cannot really say anything about developmental potential as they are only looking at normal fate which is less than their potential. Without manipulation of the cells through transplantation etc., the use of the term 'potential' or 'potent' is not appropriate except when they find cells that are multipotent. Rather than calling cells unipotent, I would suggest using the phrase 'assume a single fate'.

We have replaced all instances of unipotent with uni-fated.

3. Lines 112-115, the authors state that variability in embryonic stages likely explains differences in labelling. Are there any morphological characteristics across the embryos that support this variability in stages? For example, any characteristics that suggest that the n=3 embryos are slightly older, and the n=7 embryos are slightly younger (line 111)?

Revision Plan

We thank the reviewer for this excellent suggestion. Unfortunately, as the embryos were collected at different times, it is not possible to directly compare embryos from different litters. To address this, we would need to repeat the lineage tracing experiments by collecting embryos at fixed time points. This approach would allow us to compare variability in developmental stages at the time of collection while accounting for differences in labeling. Our live analysis shows that the early and late mesoderm contribute to the cardiac crescent and heart tube inflows, respectively, supporting our interpretation of the lineage tracing results.

4. Paragraph beginning on line 116: Please clarify how cells were counted, from the wholemount/across sections?

We counted the tdTomato+ cells across sections in wholemount embryos using the Cell Counter plugin in Fiji. We added this information to the Methods section.

5. Line 165: Authors state that in the absence of tamoxifen, tdTomato-positive cells were identified in one embryo. Please state here the total number of embryos out of which this one embryo was counted.

Done.

6. Line 190: 'Figure 2-Supplementary Figure 3A-F' doesn't exist. Do they mean Fig.3 supplementary 3A-F?

Yes, thank you, we corrected. Fig.3 supplementary 3A-F is now Fig.4 supplementary 3A-F.

7. Figure 1F-G: For cross sections in 'G' please show the level they were taken from in 'F'.

The cross-section shown in panel G (now Figure 2B) was not taken from the same embryo depicted in panel F (now Figure 2C). We apologize for the confusion and have clarified this point in the text.

8. Figure 4I: There is a large disparity in cell dispersion across movies. Please comment on why this could be. Is there a difference in stage/morphology etc..

Movies 1 and 2 depict embryos cultured at earlier stages, while Movies 3 to 5 show embryos cultured at later stages. The later the embryonic stage at the start of culture, the less dispersed along the anterior-posterior (AP) and dorsal-ventral (DV) axes of the heart tube the clones were. This is consistent with the idea that cell dispersion was more prominent during the earliest phases of migration taking place in the earlier embryos, consistent with the results from Dominguez et al. 2022. We will add a graph comparing the stages at which the cells were tracked (based on the alignment of the movies shown in Figure 5B) to cell dispersion to illustrate this point and have clarified in the manuscript.

Revision Plan

9. Figure 4K-L: The arrowhead color is too similar to the cell fluorescence color, making the visualization a little confusing. Changing the color of the arrowheads may be helpful. This is also true for some of the other figures (red arrowheads).

We have changed all the red arrows to white arrows.

Significance

This is a well-done study that will be useful to developmental biologists as well as cardiologists. The experiments seem very well done and beautifully executed. With the proposed modifications, it will make a very nice contribution to the literature.

Reviewer #3 (Evidence, reproducibility and clarity (Required)):

In their manuscript, Abukar et al. investigate the origins and migratory behaviors of cardiac progenitor cells, in mice, from gastrulation to early heart tube formation. They use sophisticated live imaging to track individual mesodermal cells, reconstructing their lineage and fate over several generations. The findings reveal distinct unipotent progenitors that contribute exclusively to specific cardiac regions, such as the left ventricle/atrioventricular canal (LV/AVC) or atrial cardiomyocytes. LV/AVC progenitors differentiate early, forming the cardiac crescent, while atrial progenitors differentiate later, contributing to the venous poles of the heart tube. Additionally, the study identifies multipotent mesodermal progenitors contributing to various mesodermal cell types, including the endocardium, pericardium and extraembryonic tissues.

Major comments:

1. Important conclusions of the manuscript rely on the expression of a reporter line (cTnnt2-2a-eGFP) as well as on the position of tdTomato+ cells in relation to the reporter. We feel that markers of non-myocardial lineages should have been used to better characterize these populations. We acknowledge the technical challenge of live imaging, which may not allow labeling of all lineages. We believe that a better description of the final stages of investigation with markers of endocardium, pericardium, extra-embryonic mesoderm together with the eGFP of the reporter will strengthen the conclusions drawn on the multipotency of the progenitors. If not addressed, some claims may appear more speculative and would benefit from being toned down.

We agree that the use of additional specific reporters and endogenous marker gene expression data would provide further insights and have now acknowledged this point in the Discussion. For example, the extra-embryonic mesoderm is situated in the extra-embryonic space, and additional markers would help identify which cell types within the ExEm compartment were

Revision Plan

traced. Similarly, many cells were classified as meso but could not be defined further in the absence of suitable markers in our live imaging experiments.

However, we stand by our assertion that the spatial distribution of progenitors in the heart tube regions, as observed in our live-imaging data—particularly within the somatic and inner endocardial layers surrounding the cTnnT-2a-GFP+ myocardial layer—provides the most compelling evidence.

Gene expression is not always a perfect proxy for assigning cell fates without carefully documented spatial context, as transcription factors (TFs) are often expressed in multiple cell types. For example, Hand1 is expressed in the pericardium, ExEm, and left ventricle myocardium, while Nr2f2 is expressed throughout the posterior mesoderm and not exclusively in myocytes (as shown in Fig. 1H). Similarly, Tal1 is expressed in hemogenic endothelial/blood progenitors located in the ExEm and endocardial lineages.

Therefore, we stand by our cell annotations. This approach, based on cell location, aligns with well-established lineage mapping studies that have long demonstrated the predictive power of spatial and morphological information in early development. For instance, Wei et al. (2000) successfully predicted early segregation between myocardial and endocardial lineages solely based on cell location within these layers of the heart tube. Decades-old research has provided clear evidence that the pericardial (somatic), myocardial (splanchnic), and endocardial layers are distinguishable in E7.5 mouse embryos (see DeRuiter et al., 1992, PMID: 1567022, Figure 2A-F). In fact, cell types were often defined through morphological observation long before gene expression techniques became available. Such approaches remain relevant for elucidating cell fates, particularly in early embryogenesis, when spatial information plays a crucial role in defining progenitors.

2. Similarly, since all the results of the manuscript derive from five movies of five independent embryos, it would be important to provide a more detailed description (for example, in a table) of the experimental setup. This could include the timing of tamoxifen induction (+7h or +21h?), the stage of dissection (based on anatomical landmarks rather than dissection stage - see atlas of gastrulation), the duration of the movies, and the stage at the final time point. Providing this information would greatly enhance the ability to robustly compare each movie and ensure reproducibility. Of note, the methods section could benefit from additional clarity. For example, in line 594, the embryo from Movie1 is described as being dissected in the morning, while the next sentence states it was dissected in the afternoon, similar to the embryo in Movie5. To avoid confusion and ensure greater rigor, describing the developmental stage of the embryos rather than the time of dissection would be more precise and biologically meaningful.

We thank the reviewer for this suggestion. While we have already temporally aligned our movies based on the timing of the first LV/AVC progenitors and atrial progenitors passing the threshold to be considered as myocytes (Fig. 5B), we will provide additional staging of the embryos based on morphological landmarks at T0. This will include the extent of the nGFP+ primitive streak and

Revision Plan

the normalized intensity of the nGFP signal. Additionally, the duration of the movies and the timing of tamoxifen induction will be indicated in the table, as suggested by the reviewer. We removed the statement on the dissection in the morning and afternoon since it was clumsy.

3. This manuscript focuses primarily on LV/AVC progenitors and likely a subpopulation of atrial cardiomyocytes, leaving other cardiac progenitor populations unaddressed. While it is understandable that the study focuses on specific populations, the authors should further discuss the limitations of their approach and explain why not all cardiac progenitors were targeted. A discussion of how these limitations might impact the broader interpretation of their findings would also be valuable.

We agree with the reviewer that our analysis focuses mainly on the LV/AVC and atrial progenitors and have now mentioned these limitations in our Discussion. However, the HCN4+ inflow structures of the heart tube we are analysing likely contribute to most (if not all) of the atria later in development, rather than constituting a subpopulation. Published lineage tracing of HCN4+ cells using a tamoxifen inducible system suggests that these cells contribute to most of E19.5 atria (Fig. 2b in Später et al., 2013), raising the question of the extent of the contribution from an additional HCN4- population to the atria. However, we agree that this question warrants further investigation.

Regarding the progenitors contributing to the RV and OFT, we agree with the reviewer that our analysis does not fully address these progenitors. While we did analyse a subset of distal mesodermal cells contributing to the pharyngeal mesoderm (labeled in red in Fig.), the absence of a live marker prevented us from determining whether these cells localized in this part of the embryo were part of the cardiopharyngeal mesoderm. Consequently, we labeled these cells as meso GFP- in our results.

We suspect that mesodermal cells contributing to the pharyngeal mesoderm may arise earlier than atrial progenitors and are currently investigating their origin using a new Tbx1-2a-tdTomato reporter line (Figure 1). However, as these findings are still preliminary and require further work, which is beyond the scope of this manuscript, we prefer not to include these data at this stage.

More broadly, we fully agree with the reviewer that the inclusion of additional markers in future studies will provide a more comprehensive understanding of cardiac development, and we are excited to pursue this work in the coming years.

Revision Plan

A

Figure 1 Image sequences from a time-lapse video of $Tn^{GFP-CreERT2/+}; R26^{Zgreen/+}; Tbx1-2a-tdTomato$ embryos, following tamoxifen administration (0.02 mg/body weight) at E6+21h.

4. Since a recent preprint (Sendra et al.), already cited in the manuscript, used complementary approaches to investigate endothelial/endocardial cell fate during gastrulation, we feel that a more in-depth discussion is warranted. In particular, how the results presented here align with the early segregation between endocardial and myocardial lineages observed by Sendra et al. could be clarified. Additionally, it is unclear how these findings correlate with *Foxa2* lineage tracing. Addressing these points could further strengthen the contextualization and impact of the manuscript.

We agree with the reviewer and have highlighted in our Discussion how our findings align with the Sendra et al. study. Specifically, our observation of short-lived multipotent progenitors supports the hypothesis that mesodermal lineages, including endocardial lineage, are rapidly established during gastrulation. Our observation of rare endo-myocardial bipotent progenitors is consistent with these findings and aligns with clonal analyses by Devine et al., which identified a shared mesodermal progenitor between these two lineages (Figure 1J in Devine et al., 2014).

However, we believe that the scATAC-seq evidence for an earlier lineage bias specifically toward the endocardial lineage warrants further investigation. In our opinion, it remains unclear whether the nuclei analyzed in their study represent prospective endocardium equivalent to the cells we observed in the live-imaging experiments. Notably, both *Nfatc1* and *Notch1* exhibit broader expression patterns beyond the endocardium, including in yolk sac endothelial cells and the allantois (see *J Cell Biol* (2022) 221 (6): e202108093, and doi.org/10.1002/dvdy.21246). Thus, it is plausible that the first mesodermal lineage decision observed in the Sendra et al. scATAC-seq analysis corresponds to the establishment of ExEm hemato/endothelial cells, which are the first mesoderm to ingress in the primitive streak at E6.5 (*Development* (1999) 126 (21): 4691–4701). Moreover, the scATAC-seq analysis does not demonstrate that the cells analysed are irreversibly excluded from a myocardial fate at these early stages. Instead, their data likely reflect chromatin reconfiguration within a subset of posterior epiblast cells in response to signaling.

We have clarified our mention of *Foxa2* lineage tracing. In a previous manuscript (Ivanovitch et al.), we identified a *Foxa2*+/*T*+ primitive streak (PS) region that contributes to the LV

Revision Plan

myocardium but not to the endocardial lineage at the midstreak stage, further supporting the finding that a population of uni-LV/AVC-fated progenitors exists.

Minor comments:

1. For all figures, annotations, axes and/or schematics would greatly help readers outside the field to locate the regions of interest within the embryo.

We have added axes on all our figures and added annotated.

2. Interesting questions that could be easily addressed and added in the manuscript: are mother cells T-nGFP positives? If so, do they have different levels of GFP expression? From the different movies, is there a hot spot of cell division? What is the frequency of progenitors that adopt a sustained interaction with their sister cells?

We thank the reviewer for these great suggestions. We will analyse the nGFP signals in mother cells and test whether those that are nGFP+ exhibit different levels of GFP expression. We are particularly interested on this question since we hypothesised in our previous manuscript (Ivanovitch et al., 2021, Figure 1J-K and S4 Fig) that LV progenitors express lower levels of T/Bra and, consequently, lower levels of nGFP expression compared to Atria progenitors. Furthermore, we will analyse the frequency of progenitors that adopt sustained interactions with their sister cells.

We also explored the reviewer's suggestion to analyse whether there is a hotspot of cell division. However, we found this analysis to be complex and will require spatial and temporal registration of the embryos. We feel this falls outside the scope of the present manuscript. That said, we fully agree with the reviewer that this is an intriguing question.

Reviewer #3 (Significance (Required)):

The manuscript presents a technically original study, offering one of the first prospective clonal analyses of cardiac progenitors during mouse gastrulation. While previous studies have addressed the fate of cardiac progenitors using retrospective clonal analysis or lineage tracing (e.g., Meilhac et al., 2004; Devine et al., 2014; Lescroart et al., 2014; Bardot et al., 2017; Ivanovitch et al., 2021; Tyser et al., 2021; Zhang et al., 2021), this work provides new insights into the temporal and spatial dynamics of cardiac progenitor migration and fate allocation. Notably, the study's investigation of the pericardium—a rarely studied cardiac mesodermal fate—adds significant novelty.

However, a limitation of the study is its focus on a relatively small region of the heart, primarily the left ventricle, atrioventricular canal, and atrium, which may not fully represent the broader diversity of cardiac progenitor behaviors across other regions of the developing heart.

Additionally, the lack of markers for non-myocardial cell lineages leaves open questions regarding the full spectrum of progenitor fates. These aspects could be addressed in future studies to provide a more comprehensive understanding of cardiac development.

A complementary preprint by the Torres group (Sendra et al., 2024) combines retrospective and prospective clonal analyses and highlights the multipotency of early mesodermal progenitors, particularly those contributing to non-cardiac fates. While both studies reveal the plasticity of early mesoderm, this manuscript by Abukar et al. focuses specifically on cardiac progenitors, offering unique insights into their behaviors and fate decisions.

The study is poised to have a broad impact on the fields of cardiac development and early mouse development. The tools and concepts developed here could also find applications in broader developmental biology studies. This review is written with expertise in cardiac development. I do not have sufficient expertise to evaluate computational modeling within the manuscript.

3. Description of the revisions that have already been incorporated in the transferred manuscript

Please insert a point-by-point reply describing the revisions that were already carried out and included in the transferred manuscript. If no revisions have been carried out yet, please leave this section empty.

We have improved and split the figures to reduce the amount of information in each figure; and altered the legends to provide more detailed explanations. We have implemented textual changes to clarify points raised by the reviewer. We have also improved the summary schematics.

4. Description of analyses that authors prefer not to carry out

Please include a point-by-point response explaining why some of the requested data or additional analyses might not be necessary or cannot be provided within the scope of a revision. This can be due to time or resource limitations or in case of disagreement about the necessity of such additional data given the scope of the study. Please leave empty if not applicable.

Dear Dr. Ivanovitch,

Thank you for submitting your manuscript for consideration by the EMBO Journal. I have now read your manuscript, the reviewer comments and your response to them. Based on our editorial assessment and the referees' positive evaluations, I would like to invite you to submit a revised version of the manuscript along the lines indicated in your revision plan. From the editorial side, I find that adding the immunofluorescence-based marker validation as requested by reviewer #3 in their point 1 would strengthen the manuscript, and I strongly encourage you to add such data to the study.

We generally allow three months as standard revision time. Should you foresee a problem in meeting this three-month deadline, please let us know in advance in order to arrange an extension. As a matter of policy, competing manuscripts published during this period will not negatively impact on our assessment of the conceptual advance presented by your study. However, please contact me as soon as possible upon publication of any related work to discuss the appropriate course of action.

When preparing your letter of response to the referees' comments, please bear in mind that this will form part of the Review Process File and will therefore be available online to the community. For more details on our Transparent Editorial Process, please visit our website: <https://www.embopress.org/page/journal/14602075/authorguide#transparentprocess>. Please also see the attached instructions for further guidelines on preparation of the revised manuscript.

Please feel free to contact me if you have any further questions regarding the revision. Thank you for the opportunity to consider your work for publication. I look forward to receiving your revised manuscript.

With best regards,

Ieva

We realize that it is difficult to revise to a specific deadline. In the interest of protecting the conceptual advance provided by the work, we recommend a revision within 3 months (29th Apr 2025). Please discuss the revision progress ahead of this time with the editor if you require more time to complete the revisions. Use the link below to submit your revision:

Link Not Available

Rev_Com_number: RC-2024-02815

New_manu_number: EMBOJ-2025-120297-T

Corr_author: ivanovitch

Title: Live-imaging reveals Coordinated Cell Migration and Cardiac Fate Determination during Mammalian Gastrulation.

Response to Reviewers.

We would like to thank the reviewers for the constructive comments. The reviewers raised several concerns, which we are addressing point by point.

Reviewer #1

Evidence, reproducibility and clarity

The manuscript describes the tracking of individual mesoderm cells through live imaging. Through a combination of reporters including a novel cardiomyocyte reporter and a combined nuclear GFP-inducible Cre reporter under the dependence of the Brachyury promoter, the authors label mesoderm cells at different stages of gastrulation then perform long term (>30h) live imaging of late gastrulation embryo up to the cardiac crescent and heart tube stages. They use elaborate analysis tools as well as manual tracking to reconstruct cells' trajectory, lineage trees, and various behavioral traits.

The study is well designed. Experiments are technically challenging, well executed, and carefully analysed.

Methods are clear and complete so that experiments should be faithfully reproduced provided availability of an appropriate microscope.

The description of the results of the live imaging experiments is not easy to read and understand, but I believe this is inherent to the complexity of the results themselves and due to the high diversity of behaviors observed. Similarly the figures are extremely dense and some graphs would benefit from a more didactic legend.

I realize the difficulty of being more concise due to the large amount of information and its diversity. If possible, I would suggest integrating tables within the results section that may help shorten the text, and may be easier to grasp.

We have added 4 tables describing the staging of the embryos in movies 1-5 (Table 1) the numbers of uni-fated and multi-potent mothers (Table 2), cell dispersion (Table 3) and speeds (Table 4). We have also split the figures to reduce the amount of information in each figure (we have now 11 figures instead of 7); and improve the legends by providing more detailed explanations, particularly regarding the dispersion analysis (Fig. 8).

The interpretation of the results is fair and in line with previous studies, which are adequately cited.

A discussion on the reasons why a large proportion of cells could not be determined as uni or multipotent might be useful. Instinctively I would imagine that a majority of those are multipotent and therefore harder to track, so if the authors do not agree with this interpretation it may be useful to detail technical reasons why those cells cannot be fully interpreted.

We have discussed further reasons why a large proportion of cells could not be classified as uni-fated or multipotent. Indeed, while our analysis revealed a predominance of uni-fated progenitors (n=98, generating 728 descendants) over bifated/trifated progenitors (n=18, generating 302 descendants), a significant number of mother cells (n=111) produced progeny whose fates could not be determined. This is due to multiple factors, as explained below.

First, we were unable to fully track a large proportion of cells that generate short tracks. This limitation hindered our ability to determine their final fates. One key reason for these shorter tracks was the occasional high density of labeling, which, coupled with the spatiotemporal resolution of our imaging setup (0.347 x 2 μ m z-stacks acquired every 2 minutes), was insufficient to consistently and unambiguously curate some cell tracks. We agree with the reviewer that the difficulty in tracking was probably exacerbated by the high dispersion of cells during the earliest stages, which is particularly high for multipotent mother cells. To avoid introducing erroneous lineage assumptions, we opted to stop tracking under such conditions.

Another contributing factor is related to cells migrating to deeper regions of the heart tube. Over extended timeframes, these cells often relocated towards the more dorsal regions of the forming heart tube, where they became dimmer due to their position along the z-axis. Consequently, many daughter cells did not meet the GFP intensity threshold required to classify them as myocytes and were thus labeled as mesodermal (see Fig. 7C for an example). Additionally, some cells could not be tracked for prolonged periods, especially as they moved dorsally during the transformation of the cardiac crescent into the heart tube. A limitation of light-sheet imaging is its reduced capacity to capture high-quality images in deeper tissues due to light scattering. Addressing this limitation and improving imaging depth will be critical in future studies.

We also acknowledge the graded expression pattern of cTnnT2-GFP in the forming heart tube, with early and higher levels in LV/AVC myocytes and later, lower levels in inflow myocytes. To maintain consistency, we refrained from using different thresholds to account for these regional intensity differences. While this choice could have led to false negatives (e.g., inflow cells not meeting the GFP threshold), we believe this approach minimises the risk of false positives. Any daughter cells failing to meet the threshold were conservatively classified as mesodermal (meso GFP-), even though they may have been myocyte progenitors.

Additionally, some cells contributing to the inflow/atria regions may not have passed the GFP threshold during the imaging period but could have done so at later

developmental stages. These cells were also classified as mesodermal, as their myocyte progenitor status could not be determined. This conservative approach prioritises accuracy over overestimation. We have included all these explanations in the main text and Material and Methods.

Significance

Strengths: novel transgenic tools, powerful imaging technique, thorough quantified analysis.

Limitations: the development of embryos after E7.75-E8 is never completely normal *ex vivo*, particularly when there is no rotation. This is visible in the pictures of the embryos post culture (ballooned yolk sac, unattached allantois). It is probably not a limitation regarding cardiac development but may influence other mesoderm lineages notably ExE.

Advance: It is a unique study due to the labelling strategy, the length of imaging, and thereby the faithful tracking of cell lineages across several rounds of division. The information provided corroborates what previous hypothesis in the field based on less direct assessment, and is here very strong and unbiased.

The research is of great interest for developmental biologists (including but not limited to the heart field), cell biologists (notably those working on stem cells and organoids as it provides a ground truth), microscopy and image analysis experts.

Reviewer #2

Evidence, reproducibility and clarity

The authors perform an elegant "tour de force" lineage relationships during mouse heart development. They perform long-term live imaging and single-cell tracking in mouse embryos from early gastrulation to stages of heart tube formation. They then track the progeny of individual cells and reconstruct the lineage tree of tracked cells. They analyze how their migratory paths of cells correlate with cell fate in the heart. Altogether, the manuscript presents a highly detailed live-imaging lineage tracing study of a subset of cells in the cardiac crescent in mouse. This presents a nice contribution to the literature, but would be improved by the suggestions below.

Major comments:

1. Can the authors be sure they can track all of the derivatives of labeled cells? They are claiming to be able to follow complete lineages, but I worry if they may lose progeny in their tracking or incorrectly conclude that cells are lineally related. wonder how you could show how accurate it really is. Perhaps if the authors could include a movie where they trace what they claim as an entire lineage of a single cell and show this with the mother and daughter cells labelled throughout the movie, that would at least provide an example for readers to make their own decisions about

how reliable the lineage tracing is. Would it be feasible to include an interactive movie where the reader can move the embryo around in 3D at each time point?

We have not tracked all the derivatives of labeled cells, as explained in our response to Reviewer 1. Each cell track (up to 1,000 time points) required manual curation and verification, as even a single linkage error would compromise conclusions. When a track could not be unambiguously determined, we stopped tracking those cells. We have acknowledged this limitation in the manuscript.

We also agree with the reviewer that it is important to show the tracks, and we have therefore included supplementary movies displaying all the cells track as Maximum Projection (Movie EV13 and Movie EV4). Furthermore, our movies and track annotations are available at the BiImage archives (Accession: S-BIAD1682 DOI: 10.6019/S-BIAD1682). This will enable users to download the data and fully assess them interactively in 3D+t using MaMuT or Mastodon (<https://mastodon.readthedocs.io/en/latest/index.html>) for cell tracking, as well as to generate their own tracking data. The availability of our data through this resource will significantly enhance its value to the community.

2. The authors describe the lineage labeled cells as unipotent, bipotent, etc. But they cannot really say anything about developmental potential as they are only looking at normal fate which is less than their potential. Without manipulation of the cells through transplantation etc., the use of the term 'potential' or 'potent' is not appropriate except when they find cells that are multipotent. Rather than calling cells unipotent, I would suggest using the phrase 'assume a single fate'.

We have replaced all instances of unipotent with uni-fated.

3. Lines 112-115, the authors state that variability in embryonic stages likely explains differences in labelling. Are there any morphological characteristics across the embryos that support this variability in stages? For example, any characteristics that suggest that the n=3 embryos are slightly older, and the n=7 embryos are slightly younger (line 111)?

We thank the reviewer for this excellent suggestion. Unfortunately, as the embryos were collected at different times, it is not possible to directly compare embryos from different litters. To address this, we would need to repeat the lineage tracing experiments by collecting embryos at fixed time points. This approach would allow us to compare variability in developmental stages at the time of collection while accounting for differences in labeling. Our live analysis shows that the early and late mesoderm contribute to the cardiac crescent and heart tube inflows, respectively, supporting our interpretation of the lineage tracing results from fixed embryos.

4. Paragraph beginning on line 116: Please clarify how cells were counted, from the whole mount/across sections?

We counted the tdTomato+ cells across sections in wholemount embryos using the Cell Counter plugin in Fiji. We added this information to the Methods section.

5. Line 165: Authors state that in the absence of tamoxifen, tdTomato-positive cells were identified in one embryo. Please state here the total number of embryos out of which this one embryo was counted.

Done.

6. Line 190: 'Figure 2-Supplementary Figure 3A-F' doesn't exist. Do they mean Fig.3 supplementary 3A-F?

Yes, thank you, we corrected. Fig.3 supplementary 3A-F is now Fig. EV4A-F.

7. Figure 1F-G: For cross sections in 'G' please show the level they were taken from in 'F'.

The cross-section shown in panel G (now Figure 2B) was not taken from the same embryo depicted in panel F (now Figure 2C). We apologize for the confusion and have clarified this point in the Figure legends.

8. Figure 4I: There is a large disparity in cell dispersion across movies. Please comment on why this could be. Is there a difference in stage/morphology etc..

Movies 1 and 2 depict embryos cultured from earlier stages, while Movies 3 to 5 show embryos cultured from later stages. We have now provided further evidence for these differences in stages (Fig. EV1 and Table 1 and line 192) which we now also based on anatomical landmarks -namely the node, primitive streak and notochord and not just on the timing of differentiation of the LV/AVC and Atria cardiomyocytes (see also our response to Reviewer 3 point 2). The later the embryonic stage at the start of culture, the less dispersed along the anterior-posterior (AP) and dorsal-ventral (DV) axes of the heart tube the clones were. This is consistent with the idea that cell dispersion was more prominent during the earliest phases of migration taking place in the earlier embryos, consistent with the results from Dominguez et al. 2022. We have added a graph further comparing the range of cell dispersion per stages tracked (Fig. 6F and based on the alignment of the movies shown in Fig. 5B) for each movie to illustrate this point and have clarified in the manuscript (line 275).

9. Figure 4K-L: The arrowhead color is too similar to the cell fluorescence color, making the visualization a little confusing. Changing the color of the arrowheads may be helpful. This is also true for some of the other figures (red arrowheads).

We have changed all the red arrows to white arrows.

Significance

This is a well-done study that will be useful to developmental biologists as well as cardiologists. The experiments seem very well done and beautifully executed. With the proposed modifications, it will make a very nice contribution to the literature.

Reviewer #3 (Evidence, reproducibility and clarity (Required)):

In their manuscript, Abukar et al. investigate the origins and migratory behaviors of cardiac progenitor cells, in mice, from gastrulation to early heart tube formation. They use sophisticated live imaging to track individual mesodermal cells, reconstructing their lineage and fate over several generations. The findings reveal distinct unipotent progenitors that contribute exclusively to specific cardiac regions, such as the left ventricle/atrioventricular canal (LV/AVC) or atrial cardiomyocytes. LV/AVC progenitors differentiate early, forming the cardiac crescent, while atrial progenitors differentiate later, contributing to the venous poles of the heart tube. Additionally, the study identifies multipotent mesodermal progenitors contributing to various mesodermal cell types, including the endocardium, pericardium and extraembryonic tissues.

Major comments:

1. Important conclusions of the manuscript rely on the expression of a reporter line (cTnnt2-2a-eGFP) as well as on the position of tdTomato+ cells in relation to the reporter. We feel that markers of non-myocardial lineages should have been used to better characterize these populations. We acknowledge the technical challenge of live imaging, which may not allow labeling of all lineages. We believe that a better description of the final stages of investigation with markers of endocardium, pericardium, extra-embryonic mesoderm together with the eGFP of the reporter will strengthen the conclusions drawn on the multipotency of the progenitors. If not addressed, some claims may appear more speculative and would benefit from being toned down.

We agree that the use of additional specific reporters and endogenous marker gene expression data would provide further insights into the cell populations traced, and we have addressed this point in the Discussion (line 511). For example, the ExEm is situated in the extra-embryonic space, and additional markers would help further delineate specific cell types within this compartment (e.g., endothelium, blood, mesothelium, etc...). Similarly, many cells were classified as mesoderm since they were located in the intra mesodermal compartment but could not be further defined due to the absence of suitable markers in our live imaging experiments.

However, we would like to stand by our cell annotations in the heart tube region. To further support this, we have incorporated additional HCR and immunostainings to

identify the pericardium and endocardium at the heart tube stage using marker genes (Appendix Fig. S3A-B). These data confirm that at E8.5, the heart tube consists of a cTnnT-2a-GFP+ myocardial layer surrounding by a cTnnT-2a-GFP-CD31+ endocardial inner layer and a cTnnT-2a-GFP- Hand1+ pericardial outer layer (or somatic mesoderm) (see Zhang et al., 2021; Dominguez et al., 2023; Ivanovitch et al., 2017 for similar stainings).

Gene expression is, however, not always a definitive proxy for assigning cell fates without considering spatial context. Transcription factors (TFs) are often expressed in multiple cell types, making their interpretation complex. For example, Hand1 is expressed in both the pericardium and the ExEm and later in the left ventricle myocardium during heart looping stages. Similarly, CD31 is expressed in both endothelial and endocardial lineages.

Therefore, we stand by our cell annotations, which are based on cell location and align with previous lineage-mapping studies (DeRuiter et al., 1992, PMID: 1567022, Figure 2A-F). For example, Wei et al. (2000) (PMID: 11084650) successfully predicted the early segregation of myocardial and endocardial lineages solely based on cell location within these layers of the heart tube.

2. Similarly, since all the results of the manuscript derive from five movies of five independent embryos, it would be important to provide a more detailed description (for example, in a table) of the experimental setup. This could include the timing of tamoxifen induction (+7h or +21h?), the stage of dissection (based on anatomical landmarks rather than dissection stage - see atlas of gastrulation), the duration of the movies, and the stage at the final time point. Providing this information would greatly enhance the ability to robustly compare each movie and ensure reproducibility. Of note, the methods section could benefit from additional clarity. For example, in line 594, the embryo from Movie1 is described as being dissected in the morning, while the next sentence states it was dissected in the afternoon, similar to the embryo in Movie5. To avoid confusion and ensure greater rigor, describing the developmental stage of the embryos rather than the time of dissection would be more precise and biologically meaningful.

We thank the reviewer for this suggestion. While we had already temporally aligned our movies based on the timing of the first LV/AVC progenitors and atrial progenitors passing the threshold to be considered myocytes (Fig. 5B), we now provide additional staging of the embryos at T0 in a new figure and table (Fig. EV1, Table 1 and Appendix Fig. S4 and line 192).

The staging is based on anatomical landmarks, including:

1. The presence or absence of a visible node,
2. The presence or absence of axial mesoderm
3. The length of the T-2a-nGFP+ primitive streak
4. The average fluorescence intensity of T-2a-nGFP+ primitive streak cells. We previously found that primitive streak cells at the mid to late streak stages express lower levels of T protein compared to the OB and EB stages. The

transgenic $T^{nEGP-CreERT2/+}$ embryo showed a similar pattern, with lowest nGFP signal at the mid-late streak stage (see S4Ai and S4Aii Fig in Ivanovitch et al. 2021).

Thus, embryos in movies 3, 4, and 5 are culture from later stages than those in movies 1 and 2. Additionally, the duration of the movies and the timing of tamoxifen induction are indicated in Table 1, as suggested by the reviewer. We also removed the statement regarding dissection in the morning and afternoon, as it was unnecessary.

3. This manuscript focuses primarily on LV/AVC progenitors and likely a subpopulation of atrial cardiomyocytes, leaving other cardiac progenitor populations unaddressed. While it is understandable that the study focuses on specific populations, the authors should further discuss the limitations of their approach and explain why not all cardiac progenitors were targeted. A discussion of how these limitations might impact the broader interpretation of their findings would also be valuable.

We agree with the reviewer that our analysis focuses mainly on the LV/AVC and atrial progenitors. However, we think the HCN4+ inflow structures of the heart tube we are analysing may contribute to most (if not all) of the atria later in development, rather than constituting a subpopulation of atrial cells. Published lineage tracing of HCN4+ cells using a tamoxifen inducible system suggests that these cells contribute to most of E19.5 atria (Fig. S4A -tam @E8 -in Später et al., 2013), raising the question of the extent of the contribution from an additional HCN4- population to the atria. However, we agree that this question warrants further investigation.

Regarding the progenitors contributing to the RV and OFT, we agree with the reviewer that our analysis does not fully address these progenitors. While we did analyse the distal mesodermal cells (labeled in dark red in Fig. 7A and Movie EV10 and line 362), the absence of a live marker prevented us from determining whether these cells localised in this part of the embryo were part of the cardiopharyngeal mesoderm. Consequently, we labeled these cells as meso GFP- in our lineage trees (Fig 4A).

We have now mentioned these limitations in our Discussion (line 492).

We suspect that mesodermal cells contributing to the pharyngeal mesoderm may arise earlier than atrial progenitors and are currently investigating their origin using a new *Tbx1-2a*-tdTomato reporter line (Figure 1 below). However, as these findings are still preliminary and require further work, which is beyond the scope of this manuscript, we prefer not to include these data at this stage.

More broadly, we fully agree with the reviewer that the inclusion of additional markers in future studies will provide a more comprehensive understanding of cardiac development, and we are excited to pursue this work in the coming years.

A

Figure 1 Image sequences from a time-lapse video of $TnGFP-CreERT2^{+/-}; R26ZGreen^{+/-}; Tbx1-2a-tdTomato$ embryos, following tamoxifen administration (0.02 mg/body weight) at E6+21h.

4. Since a recent preprint (Sendra et al.), already cited in the manuscript, used complementary approaches to investigate endothelial/endocardial cell fate during gastrulation, we feel that a more in-depth discussion is warranted. In particular, how the results presented here align with the early segregation between endocardial and myocardial lineages observed by Sendra et al. could be clarified. Additionally, it is unclear how these findings correlate with *Foxa2* lineage tracing. Addressing these points could further strengthen the contextualization and impact of the manuscript.

We agree with the reviewer and have highlighted in our Discussion how our findings align with the Sendra et al. study (line 543). Specifically, our observation of short-lived multipotent progenitors supports the hypothesis that mesodermal lineages, including the endocardial lineage, are rapidly established during gastrulation. Our observation of rare endo-myocardial bipotent progenitors is consistent with these findings and aligns with clonal analyses by Devine et al., which identified a shared mesodermal progenitor between these two lineages (Figure 1J in Devine et al., 2014).

However, we believe that the scATAC-seq evidence for an earlier lineage bias specifically toward the endocardial lineage warrants further investigation. In our opinion, it remains unclear whether the nuclei analysed in their study represent prospective endocardium equivalent to the cells we observed in the live-imaging experiments. Notably, both *Nfatc1* and *Notch1* exhibit broader expression patterns beyond the endocardium, including in yolk sac endothelial cells and the allantois (see *J Cell Biol* (2022) 221 (6): e202108093, and doi.org/10.1002/dvdy.21246). Thus, it is plausible that the first mesodermal lineage decision observed in the Sendra et al. scATAC-seq analysis corresponds to the establishment of ExEm hemato/endothelial cells, which are the first mesoderm to ingress in the primitive streak at E6.5 (*Development* (1999) 126 (21): 4691–4701). Moreover, the scATAC-seq analysis does not demonstrate that the cells analysed are irreversibly excluded from a myocardial fate at these early stages. Instead, their data likely reflect chromatin reconfiguration within a subset of posterior epiblast cells in response to signalling.

We have clarified our mention of Foxa2 lineage tracing (line 138). In a previous manuscript (Ivanovitch et al.), we identified a Foxa2+/T+ primitive streak (PS) region that contributes to the LV myocardium but not to the endocardial lineage at the midstreak stage, further supporting the finding that a population of uni-LV/AVC-fated progenitors exists.

Minor comments:

1. For all figures, annotations, axes and/or schematics would greatly help readers outside the field to locate the regions of interest within the embryo.

We have added axes on all our figures and added annotations and schematics in Fig. EV1A and A'.

2. Interesting questions that could be easily addressed and added in the manuscript: are mother cells T-nGFP positives? If so, do they have different levels of GFP expression? From the different movies, is there a hot spot of cell division? What is the frequency of progenitors that adopt a sustained interaction with their sister cells?

We thank the reviewer for these great suggestions. We have analysed the nGFP signals in mother cells and tested whether they expressed different levels of GFP. We were particularly interested in this question since we hypothesised in our previous publication that LV progenitors express lower levels of T/Bra and, consequently, lower levels of nGFP expression compared to atria progenitors (Ivanovitch et al., 2021, Figure 1J-K and S4 Fig). Unfortunately, this analysis was not feasible since, by the time the tdTomato fluorophore was detectable by microscopy, the progenies were no longer expressing the T-2a-nGFP-cre transgene (see below, Figure 2A). In all the movies except for Movie 2, all the mother cells had nGFP intensity levels below the background intensity values (black bars in Figure 2A).

A

Figure 2A. *nGFP* intensity analysis in mother cells per fate and per movie. Black lines show the threshold background value for each fate (based on the background value at the stages when early and late mesodermal progenitors are born).

We have analysed the frequency of progenitors that adopt sustained interactions with their sister cells and mentioned (line 453).

We also explored the reviewer's suggestion to analyse whether there is a hotspot of cell division. However, we found this analysis to be complex and will require spatial and temporal registration of the embryos. We feel this falls outside the scope of the present manuscript. That said, we fully agree with the reviewer that this is an interesting question.

Reviewer #3 (Significance (Required)):

The manuscript presents a technically original study, offering one of the first prospective clonal analyses of cardiac progenitors during mouse gastrulation. While previous studies have addressed the fate of cardiac progenitors using retrospective clonal analysis or lineage tracing (e.g., Meilhac et al., 2004; Devine et al., 2014; Lescroart et al., 2014; Bardot et al., 2017; Ivanovitch et al., 2021; Tyser et al., 2021; Zhang et al., 2021), this work provides new insights into the temporal and spatial dynamics of cardiac progenitor migration and fate allocation. Notably, the study's investigation of the pericardium—a rarely studied cardiac mesodermal fate—adds significant novelty.

However, a limitation of the study is its focus on a relatively small region of the heart, primarily the left ventricle, atrioventricular canal, and atrium, which may not fully represent the broader diversity of cardiac progenitor behaviors across other regions of the developing heart. Additionally, the lack of markers for non-myocardial cell lineages leaves open questions regarding the full spectrum of progenitor fates. These aspects could be addressed in future studies to provide a more comprehensive understanding of cardiac development.

A complementary preprint by the Torres group (Sendra et al., 2024) combines retrospective and prospective clonal analyses and highlights the multipotency of early mesodermal progenitors, particularly those contributing to non-cardiac fates. While both studies reveal the plasticity of early mesoderm, this manuscript by Abukar et al. focuses specifically on cardiac progenitors, offering unique insights into their behaviors and fate decisions.

The study is poised to have a broad impact on the fields of cardiac development and early mouse development. The tools and concepts developed here could also find applications in broader developmental biology studies. This review is written with expertise in cardiac development. I do not have sufficient expertise to evaluate computational modeling within the manuscript.

Dear Dr. Ivanovitch,

Thank you for submitting a revised version of your manuscript. We have now received input from all original reviewers, who find that their main concerns have been addressed satisfactorily and recommends acceptance of the manuscript.

Additionally, there remain a few editorial points that need addressing before I can extend official acceptance of the manuscript:

1. Please submit up to five keywords.
2. Please make sure that the order of the sections in the manuscript is as follows: abstract, introduction, results, discussion, materials & methods, data availability section, acknowledgments, disclosure statement and competing interests, references, main figure legends, tables, expanded figure legends.
3. Please rename "Competing interests" section into "Disclosure and competing interests statement" (further info: <https://www.embopress.org/page/journal/14602075/authorguide#conflictsofinterest>).
4. CRedit has replaced the traditional author contributions section because it offers a systematic, machine-readable author contributions format that allows for more effective research assessment. Please remove the Authors Contributions from the manuscript and use the free text boxes beneath each contributing author's name in our online submission system to add specific details on the author's contribution. More information is available in our guide to authors.
5. Please check the order of the figure callouts, as they should be listed sequentially.
6. Figure 4 has only one panel, so the label A is not necessary.
7. Figure panel 10F is not mentioned in the manuscript text - please add the corresponding callout.
8. There is a reference to "data not shown" on page 9. According to our policy, which does not permit references to "data not shown", please include this information in the Appendix. Please see also <https://www.embopress.org/page/journal/14602075/authorguide#unpublisheddata>.
9. Please rename the movies into Movie EV1-EV10 and update the callouts accordingly. The legends should be removed from the manuscript text file and zipped with each movie file. Further information is available here: <https://www.embopress.org/page/journal/14602075/authorguide#expandedview>
10. Please rename Tables 1-4 into Table EV1-EV4, also throughout the manuscript text. Please remove their legends from the manuscript text file and include above the tables in each Excel file.
11. In the Appendix, please add the manuscript title in the front page, e.g., "Appendix for... 'Early coordination of cell migration and cardiac fate determination during mammalian gastrulation'".
12. The provided synopsis image is difficult to read when resized to the required maximal dimensions of 550x600 pixels (please see in the attachment). Please increase the font size or select a part of the image for the synopsis.
13. Source data files need to be saved as one folder per figure and then uploaded as .zip files. E.g. all the Source data files for figure 1 need to be saved in a single folder and this needs to be zipped and then uploaded as "SD figure 1.zip" file. For EV and/or appendix figures, please ZIP together all source data in a single folder. Completed Source Data checklist should be uploaded as Related Manuscript File.
14. In our standard image integrity check, we noted that the following figure panels are reused in the manuscript while being labelled as derived from different movies:
 - between Figure 3C (labelled as from Movie 4) and Figure EV3A (Movie 3).
 - Between Figure EV3A (Movie 3) and Figure EV5B (Movie 4).Please check and correct as necessary.
15. Our data editors have flagged the following issues in figure legends that need correcting:
 - Please indicate what */ **/ ***/ **** represents; if this represents p value(s), please indicate the statistical test used and where appropriate, the exact p value in the legend(s) of figure(s) 8A, C, D, F, G.
 - Please indicate what */ **/ ***/ **** represents; if this represents p value(s), please indicate the exact p value in the legend(s) of figure(s) 2D, 5B, 9F, 10B, C, G; 11D.
 - Please indicate the statistical test used for data analysis in the legend of figure 6B.
 - Please define the box plots in terms of minima, maxima, centre, bounds of box and whiskers, and percentile in the legends of figures 2D, 5B-D; 6B, 8A, C, D, F, G; 9F, 10B, C, G; 11D.
 - Please define the centre of the box plot in the legend of figure 6E.
 - Please add information on the nature and number of replicates in the legends of figures 5B, C, D; 6E; 8A, C, D, F, G; 9F, 10B, C, G; 11D
 - Please define the scale bar for figures 5A, 9G, H; 10F, 11A-C; EV3 A-C; EV5 A-D.
 - Please define the white arrow heads in the legend of figures 5E, F.
 - Please define the white arrows in the legend of figures 9G, H, EV5 A-D.
 - Please define the yellow, green, blue arrows in the legend of figure 10F.

With best wishes,

Ieva

Revision to The EMBO Journal should be submitted online within 90 days, unless an extension has been requested and approved by the editor; please click on the link below to submit the revision online before 23rd Jun 2025:

Link Not Available

Referee #1:

This is a highly valuable paper that has been significantly improved since the original version. The authors have added new HCR/immunofluorescence experiments that provide deeper insights into the different lineages at the final time points of their analysis. They have also included new tables that precisely describe the staging of the embryos, the various fates of mother cells, their dispersion, and speed, which are crucial for fully understanding the impressive dataset generated in this study. Several points of concern raised during previous reviews have been thoroughly addressed, particularly regarding the cell population traced. The overall clarity of the manuscript has been greatly enhanced through the splitting of complex figures and the addition of detailed annotations.

Overall, this paper will be a valuable resource for the field of developmental biology, especially heart development.

Referee #2:

The authors have done an excellent job of addressing the reviewers' concerns. The paper is now acceptable for publication.

Referee #3:

The authors have adequately responded to my comments, as well as (in my opinion) those of the other reviewers. I do not have additional concerns.

Rev_Com_number: RC-2024-02815
New_manu_number: EMBOJ-2025-120297R
Corr_author: ivanovitch
Title: Early coordination of Cell migration and Cardiac fate Determination during Mammalian Gastrulation.

We would like to reviewers for their constructive comments. We hope we have addressed all the remaining editorial points.

1. Please submit up to five keywords.

Cardiac Progenitors, Gastrulation, Light Sheet microscopy, Heart fields, Live-imaging

2. Please make sure that the order of the sections in the manuscript is as follows: abstract, introduction, results, discussion, materials & methods, data availability section, acknowledgments, disclosure statement and competing interests, references, main figure legends, tables, expanded figure legends.

Done

3. Please rename "Competing interests" section into "Disclosure and competing interests statement" (further

info: <https://www.embopress.org/page/journal/14602075/authorguide#conflictofinterest>).

Done

4. CRediT has replaced the traditional author contributions section because it offers a systematic, machine-readable author contributions format that allows for more effective research assessment. Please remove the Authors Contributions from the manuscript and use the free text boxes beneath each contributing author's name in our online submission system to add specific details on the author's contribution. More information is available in our guide to authors.

Done

5. Please check the order of the figure callouts, as they should be listed sequentially.

Done. We had to reorder the figures 5 and 6 and 8 and 9 and EV4.

6. Figure 4 has only one panel, so the label A is not necessary.

Done

7. Figure panel 10F is not mentioned in the manuscript text - please add the corresponding callout.

Done

8. There is a reference to "data not shown" on page 9. According to our policy, which does not permit references to "data not shown", please include this information in the Appendix. Please see

also <https://www.embopress.org/page/journal/14602075/authorguide#unpublisheddata>.

Done

9. Please rename the movies into Movie EV1-EV10 and update the callouts accordingly. The legends should be removed from the manuscript text file and zipped with each movie file. Further information is available here: <https://www.embopress.org/page/journal/14602075/authorguide#expandedview>

Done.

10. Please rename Tables 1-4 into Table EV1-EV4, also throughout the manuscript text. Please remove their legends from the manuscript text file and include above the tables in each Excel file.

Done

11. In the Appendix, please add the manuscript title in the front page, e.g., "Appendix for... 'Early coordination of cell migration and cardiac fate determination during mammalian gastrulation'".

Done

12. The provided synopsis image is difficult to read when resized to the required maximal dimensions of 55x600 pixels (please see in the attachment). Please increase the font size or select a part of the image for the synopsis.

Done and we have added a Figure 12 to show the entire synopsis.

13. Source data files need to be saved as one folder per figure and then uploaded as .zip files. E.g. all the Source data files for figure 1 need to be saved in a single folder and this needs to be zipped and then uploaded as "SD figure 1.zip" file. For EV and/or appendix figures, please ZIP together all source data in a single folder. Completed Source Data checklist should be uploaded as Related Manuscript File.

Done

14. In our standard image integrity check, we noted that the following figure panels are reused in the manuscript while being labelled as derived from different movies:

- between Figure 3C (labelled as from Movie 4) and Figure EV3A (Movie 3).

Thank you for noticing– Movie 3 in Figure EV3A was Movie 4.

- Between Figure EV3A (Movie 3) and Figure EV5B (Movie 4). Please check and correct as necessary.

Figure EV5B shows Movie 4. This was correct.

15. Our data editors have flagged the following issues in figure legends that need correcting:

- Please indicate what */ **/ ***/ **** represents; if this represents p value(s), please indicate the statistical test used and where appropriate, the exact p value in the legend(s) of figure(s) 8A, C, D, F, G.

Done

- Please indicate what */ **/ ***/ **** represents; if this represents p value(s), please indicate the exact p value in the legend(s) of figure(s) 2D, 5B, 9F, 10B, C, G; 11D.

Done

- Please indicate the statistical test used for data analysis in the legend of figure 6B.

Done

- Please define the box plots in terms of minima, maxima, centre, bounds of box and whiskers, and percentile in the legends of figures 2D, 5B-D; 6B, 8A, C, D, F, G; 9F, 10B, C, G; 11D.

Done

- Please define the centre of the box plot in the legend of figure 6E.

Done

- Please add information on the nature and number of replicates in the legends of figures 5B, C, D; 6E; 8A, C, D, F, G; 9F, 10B, C, G; 11D

Done

- Please define the scale bar for figures 5A, 9G, H; 10F, 11A-C; EV3 A-C; EV5 A-D.

Done

- Please define the white arrow heads in the legend of figures 5E, F.

Done

- Please define the white arrows in the legend of figures 9G, H, EV5 A-D.

Done

- Please define the yellow, green, blue arrows in the legend of figure 10F.

Done

With best wishes,

leva

Dear Kenzo,

Thank you for addressing the final editorial points. I sincerely apologise for the slow process from our side due to the high number of submissions that we experience at the moment. I am now pleased to inform you that your manuscript has been accepted for publication - congratulations on a nice study!

Before we forward your manuscript to our publishers, I would need your input on the two points below:

1. We are still lacking the information on box plot features for figure panel 2D. Based on the other figures in the study, I extrapolate that it should read as follows: "The box boundaries represent the 25th (lower quartile) and 75th (Upper quartile) percentiles, with the centre line indicating the median. The whiskers extend to the minimum and maximum values in the dataset." Please let me know if this is correct, and I can then include this information in the figure legend for you.
2. We would like to propose some edits in the manuscript abstract and synopsis (please see below and in the attached text file). I have also written a short blurb that will accompany the title of your manuscript in our online table of contents. Please let me know if any corrections or adjustments are needed:

Blurb:

Tracking of individual mesodermal cells using light sheet microscopy reveals patterns of directionality and cardiac fate allocation in the mouse embryo.

Synopsis

Progenitor cells that will contribute to formation of specific heart tissues are derived from distinct regions of the primitive streak in the developing mouse embryo. This study utilises light-sheet microscopy in mouse embryos to track individual mesodermal cells destined to the heart for up to 40 hours, revealing dynamic behaviours and lineages from gastrulation to heart tube formation.

1. The left ventricle progenitors originate from early proximal mesoderm, while atrial progenitors are derived from late proximal mesoderm.
2. Left ventricle progenitors differentiate early into myocytes, forming the cardiac crescent, while atrial progenitors differentiate later to establish the heart tube's inflow regions.
3. Most early proximal mesodermal cells are uni-fated and contribute predominantly to myocytes.
4. Multi-potent progenitors are present and rapidly commit to specific cardiac fates as they migrate toward defined embryonic regions.
5. Pairs of sister cells generated by uni-fated progenitors exhibit more coordinated migration paths compared to descendants of multi-potent progenitors.

If you have any questions, please do not hesitate to contact the Editorial Office. Thank you for this contribution to The EMBO Journal and congratulations on a great paper!

With best wishes,

Ieva

Rev_Com_number: RC-2024-02815

New_manu_number: EMBOJ-2025-120297R1

Corr_author: ivanovitch

Title: Early coordination of Cell migration and Cardiac fate Determination during Mammalian Gastrulation.